# Evaluating LLM Uncertainty in Long-Form Generation Using Deterministic Ground Truth

Ido Amit [* 1]   Ido Galil [* 2]   Ran El-Yaniv [1 2]

## Abstract

As LLMs generate increasingly long outputs, effective uncertainty estimation must identify errors at fine-grained levels rather than discard entire responses. While such methods exist, evaluating uncertainty at any resolution (token to an entire generation) is challenging and highly sensitive to label imperfections, making zero-noise benchmarks essential; yet, long-form generation benchmarks tend to rely on fallible labels rather than deterministic ground truth. We introduce Single-answer Atomic Long-form Target (SALT), a benchmark of six procedurally generated tasks with single deterministic long textual ground truths, enabling unit-level evaluation of correctness, calibration, and ranking without external judges. Equipped with SALT, our analysis of 50+ LLMs reveals key insights: We identify which confidence functions dominate each uncertainty aspect and show that confidence ranking largely breaks at atomic resolution, even when clearer separability emerges at coarser line-level units. SALT further enables controlled atom-level interventions throughout generation, revealing two separable drivers of future errors: propagation from corrupted prefixes, dominated by global context correctness, and bounded degradation from increasing answer-context length. Finally, we demonstrate that reasoning, via Chain-of-Thought prompting or internalized through training, introduces a trade-off, improving accuracy while degrading confidence ranking. These findings directly impact risk-critical applications requiring reliable error identification and mitigation. We release the Code at github.com/IdoAmit198/SALT.

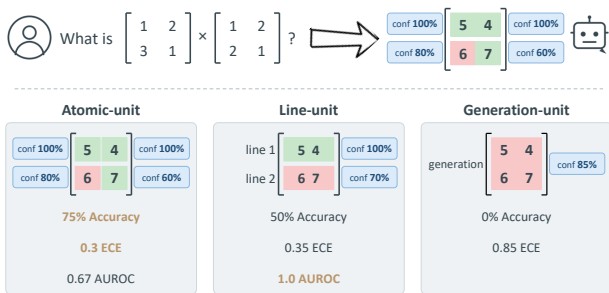

*Figure 1.* Granular uncertainty evaluation with SALT. Decomposing long-form generation into high-resolution units isolates localized errors, revealing precise calibration (ECE) and ranking (AUROC) signals obscured by coarse, generation-level evaluation.

## 1. Introduction

Large Language Models (LLMs) have demonstrated remarkable capabilities across a diverse array of Natural Language Processing (NLP) tasks, increasingly driving their adoption in real-world applications. However, deployment in high-stakes environments is fraught with risk due to their propensity for generating factually incorrect content. In domains such as healthcare, where LLMs are explored for initial patient assessment and decision-making aid, mistakes can have severe consequences (Agarwal et al., 2024; Kim et al., 2025). To mitigate these risks, robust uncertainty estimation is a critical component for trustworthy integration, enabling mechanisms like *selective generation* (Zablotskaia et al., 2023; Lee et al., 2024), where the model signals when to abstain, and *reliable decision support* (Valente et al., 2021; Lindenmeyer et al., 2024), where calibrated probabilities inform human experts of the likelihood of error.

As modern LLMs increasingly generate long-form outputs, such as software repositories spanning hundreds of thousands of tokens, their generations naturally comprise many distinct semantic components. Within a single output, some components may be correct while others contain localized errors or hallucinations. In such settings, evaluating uncertainty for the entire generation is insufficient. For instance, a diagnostic AI system providing multiple possible conditions must be calibrated at the component level to help clinicians prioritize testing. Furthermore, it is more efficient to iden-

---

[*]Equal contribution   [1]Technion   [2]Nvidia.   Correspondence to: Ido Amit <dodoamit198@gmail.com>, Ido Galil <idogalil.ig@gmail.com, igalil@nvidia.com>, Ran El-Yaniv <rani@cs.technion.ac.il, relyaniv@nvidia.com>.

*Proceedings of the 43rd International Conference on Machine Learning*, Seoul, South Korea. PMLR 306, 2026. Copyright 2026 by the author(s).

tify and revise specific faulty units than to discard a long, largely correct output. Reliable deployment thus requires fine-grained uncertainty estimation, covering both ranking and calibration, to assess individual components.

However, evaluating fine-grained uncertainty estimation in long-form generation poses inherent challenges. First, establishing correctness itself is non-trivial: many claims have multiple valid realizations, while others may be unverifiable without external knowledge. Second, fine-grained evaluation requires decomposing a free-form generation into smaller units, a process that is typically ambiguous and highly sensitive to modeling choices. Together, these issues make high-resolution uncertainty evaluation especially vulnerable to label noise and subjective judgments.

Prior work often simplifies evaluation by focusing on short-form generation or multiple-choice Q&A, where correctness is unambiguous (Wang et al., 2024b; Zellers et al., 2019; Bisk et al., 2020; Sakaguchi et al., 2021; Clark et al., 2018). However, these settings assess confidence over predefined choices rather than the generated content itself, or short outputs (few tokens, or a word) rather than the nuanced content characteristic of real-world generation (Bakman et al., 2025). Recent long-form benchmarks decompose outputs into "atomic facts" verified by LLM-based judges or entailment models (Manakul et al., 2023; Min et al., 2023; Wang et al., 2024a; Zhang et al., 2024b; Vashurin et al., 2025). This paradigm introduces noise during decomposition and verification, which can severely skew uncertainty metrics, as analyzed in Appendix I. Moreover, correctness-label errors that share biases with uncertainty scores can systematically distort AUROC rankings (Santilli et al., 2025). Furthermore, extracting claims out of context decouples confidence signals from the original text.

To address these limitations, we introduce Single-answer Atomic Long-form Target (SALT), a benchmark for zero-noise uncertainty evaluation in long-form generation. SALT comprises six procedurally generated tasks spanning logic, mathematics, code, translation, and multiple-needles-in-a-haystack (Multi-Needle). Each task is constructed such that the complete long-form output has a single deterministic ground truth, known a priori. This enables precise unit-level labeling without human or model-based judges. Modular and extensible, SALT supports effectively unbounded instance generation with controllable difficulty, ensuring contamination-free evaluation as future models evolve.

We summarize our main contributions as follows:

(1) We introduce **SALT**, a fine-grained zero-noise benchmark for long-form generations.

(2) **Fine-Grained Evaluation of Uncertainty Throughout Generation**: SALT supports evaluation at multiple granularities, allowing uncertainty to be assessed continuously throughout the generation process rather than only at the level of complete outputs.

(3) **Large-Scale Empirical Study Across 50+ LLMs** - Using SALT, we conduct a comprehensive evaluation of over 50 contemporary LLMs, including GPT-5.1, Gemini-3-Pro, Qwen-3, GLM-4.7, and MiniMax M2.5, providing the first large-scale, contamination-free analysis of uncertainty estimation behavior in long-form generation.

(4) **Key Insights into Uncertainty and Error Dynamics**: SALT enables a high-resolution analysis throughout long-form generation, revealing that (i) reasoning mechanisms (CoT/Reasoning models) introduce a "reasoning trade-off," boosting precision while systematically damaging confidence ranking; (ii) confidence ranking exhibits a high-resolution failure mode: raw confidence signals often fail to separate correct from incorrect atoms, even when coarser levels ranking remains informative; and (iii) future correctness has two separable drivers: propagation from corrupted prefixes, dominated by global rather than local context correctness, and bounded degradation from increasing answer-context length.

## 2. Problem Setup

Let $\mathcal{X}$ and $\mathcal{Y}$ denote the input and output spaces, respectively, where both consist of token sequences, i.e., $\mathcal{X}, \mathcal{Y} \subseteq \Sigma^*$ for a finite token vocabulary $\Sigma$. Given an input sequence $\mathbf{x} = (x_1, \ldots, x_m)$, a language model $f : \mathcal{X} \to \mathcal{Y}$ produces an output sequence $\hat{\mathbf{y}} = (\hat{y}_1, \ldots, \hat{y}_{T_{\text{gen}}})$, where $T_{\text{gen}}$ denotes the length of the **generated** sequence. We restrict $\hat{\mathbf{y}}$ to the model's *final answer*, using explicit output fencing; intermediate reasoning tokens are not considered.

### 2.1. Granularity and Correctness

A generation can be evaluated at various levels of granularities. While characters and tokens are small units, having little semantic meaning on their own, sentences and documents are coarse, encapsulating a lot of information. The need for such an intermediate level of granularity can be illustrated by a matrix multiplication task. Suppose an element in the resulting matrix is 135. Evaluating this output at the character or token level may assign partial credit to a value such as $-130$, despite it being entirely incorrect in the context of the task. However, evaluating correctness only at the level of the full matrix would ignore the fact that many individual elements may have been computed correctly, with the errors arising from a small number of localized mistakes or hallucinations.

We resolve this trade-off by evaluating correctness at an intermediate granularity using *evaluation units*—the smallest meaningful subcomponents that can be assessed independently. This approach enables precise error attribution with-

out over- or under-penalizing the generation. While SALT supports any arbitrary resolution, including character or token levels (see Appendix J), we focus on two task-dependent resolutions: the *atomic-unit* (the smallest standalone meaning, e.g., a matrix element) and the *line-unit* (a larger logical grouping, e.g., a matrix row).

To formalize this intermediate granularity, we decompose the token sequence $\hat{y}$ into evaluation units, which are contiguous token spans that represent atomic semantic components. Formally, let $\hat{\mathbf{u}} = (\hat{u}_1, \ldots, \hat{u}_{U_{\text{gen}}})$ denote the sequence of $U_{\text{gen}}$ generated evaluation units, where each unit $\hat{u}_i$ comprises multiple consecutive tokens. This decomposition defines a partition of $\hat{\mathbf{y}}$ via boundary indices $0 = j_0 < j_1 < \ldots < j_{U_{\text{gen}}} = T_{\text{gen}}$, where the $i$-th unit is the contiguous subsequence $\hat{u}_i = (\hat{y}_{j_{i-1}+1}, \ldots, \hat{y}_{j_i})$. For example, if $T_{\text{gen}} = 7$ and $j_1 = 3, j_2 = 7$, then $\hat{u}_1 = (\hat{y}_1, \hat{y}_2, \hat{y}_3)$ and $\hat{u}_2 = (\hat{y}_4, \hat{y}_5, \hat{y}_6, \hat{y}_7)$.

As discussed in Section 1, relying on proxy signals (such as single-token answers in multiple-choice tasks and the Negative Log-Likelihood of candidate answers) or noisy model-based verification fails to capture uncertainty throughout the generation process. To reliably evaluate the model's output, SALT requires a setting where the ground truth is completely unambiguous, a unique and deterministic sequence. Therefore, we define the *ground-truth sequence* as $\mathbf{y} = (y_1, \ldots, y_{T_{\text{gt}}})$. Note that $T_{\text{gt}}$ may differ from the generated length $T_{\text{gen}}$. Similarly, we define $\mathbf{u} = (u_1, \ldots, u_{U_{\text{gt}}})$ as the set of ground-truth evaluation units.

With generated units $\hat{\mathbf{u}}$ and ground-truth units $\mathbf{u}$ established, we define correctness via strict string equality. A unit $\hat{u}_i$ is correct if and only if it exactly matches $u_i$. For instance, in Matrix Multiplication, if the ground truth is 135, a partial sequence like 13 or an incorrect value like $-135$ both receive a binary score of 0.

Prior work distinguishes between intrinsic hallucinations (verifiably incorrect information) and extrinsic hallucinations (unverifiable claims) (Ji et al., 2023; Zhang et al., 2023; Huang et al., 2025a). Our deterministic ground-truth setting allows us to identify a restricted but important subtype, which we term *redundant units*: generated units that do not appear in the ground-truth sequence. These constitute a task-specific form of intrinsic hallucination, where the model generates unnecessary content beyond what the task requires. For example, in the matrix multiplication task, a redundant unit would be an extra element that violates the matrix's dimensions, which would be mathematically invalid. This unit-level definition enables precise localization of hallucinations throughout generation.

## 2.2. Confidence Functions

To quantify the model's confidence in its generated output, we define a confidence score function $\kappa : \mathcal{X} \times \mathcal{Y} \to \mathbb{R}$ parameterized by the model $f$. We write $\kappa(x, \hat{y}; f)$ to denote the confidence assigned to generation $\hat{y}$ given input $x$ under model $f$. When the model is clear from context, we abbreviate this as $\kappa(x, \hat{y})$. Unlike classification tasks, where confidence is typically associated with a single predicted label, generation tasks produce output sequences consisting of multiple tokens, and confidence can therefore be defined at different granularities. In particular, confidence estimates may be computed at the token level (e.g., using log-probabilities from the language model head), at the level of spans or words, or directly at the level of the entire generated sequence.

Given the definition of evaluation units above, we compute the confidence score of a specific unit $\hat{u}_i$ via $\kappa(\mathbf{x}, \hat{u}_i \mid f)$. Since units typically consist of multiple tokens, token-level signals are often aggregated to obtain unit-level confidence score. For such units, we aggregate token probabilities using functions like mean probability, perplexity, or entropy (Table 7, Appendix F). This applies regardless of the evaluation granularity, whether at the atomic, line, or generation level.

In tasks such as ranking or selective prediction (Chow, 1957; El-Yaniv & Wiener, 2010; Geifman & El-Yaniv, 2017; Cortés-Ciriano & Bender, 2018), $\kappa$ can take arbitrary real values without loss of interpretability. However, calibration metrics require $\kappa \in [0, 1]$ with a probabilistic interpretation (Dawid, 1982; Zadrozny & Elkan, 2002; Niculescu-Mizil & Caruana, 2005). To ensure every confidence function can be evaluated on all metrics, we compare several calibration methods (Section 5.2) and apply the best-performing post-hoc calibration (details in Appendix E.1).

## 2.3. Metrics

Throughout this work, we assess several metrics of LLMs' generations. To assess accuracy, we measure precision and recall. Precision and recall are defined as

$$\text{Precision} = \frac{|\mathbf{u} \cap \hat{\mathbf{u}}|}{U_{\text{gen}}}, \qquad \text{Recall} = \frac{|\mathbf{u} \cap \hat{\mathbf{u}}|}{U_{\text{gt}}}. \quad (1)$$

Recall reflects the proportion of ground-truth units recovered but ignores hallucinations, whereas precision considers the presence of redundant units. For example, if a generation contains all ground-truth units but comprises 99% incorrect content, recall remains 100% while precision drops to 1%. This sensitivity makes precision the natural metric for assessing correctness. Given their very high average Pearson correlation (0.98, see Table 2) we report only precision as our measure of accuracy (see Appendix A).

We adopt the expected calibration error (ECE) metric

(Naeini et al., 2015), which is widely used to assess confidence calibration. For a finite test set of size $N$, ECE is calculated by grouping all instances into $m$ interval bins (such that $m \ll N$), each of size $\frac{1}{m}$ (the confidence interval of bin $B_j$ is $(\frac{j-1}{m}, \frac{j}{m}]$). With $\text{acc}(B_j)$ being the mean accuracy in bin $B_j$ and $\text{conf}(B_j)$ being its mean confidence, ECE is defined as

$$ECE = \sum_{j=1}^{m} \frac{|B_j|}{N} \sum_{i \in B_j} \left| \frac{\mathbb{1}[\hat{y}_f(x_i) = y_i]}{|B_j|} - \frac{\kappa(x, \hat{y}_f(x_i)|f)}{|B_j|} \right|$$

$$= \sum_{j=1}^{m} \frac{|B_j|}{N} |\text{acc}(B_j) - \text{conf}(B_j)|$$

We additionally evaluate the adaptive calibration error (ACE) (Nixon et al., 2019), with results reported in the Appendix. Beyond calibration, a desired confidence function should induce an accurate partial order by assigning higher scores to correct units than to incorrect ones. Specifically, for any two random input context-generated unit pairs where $\hat{u}_i$ is incorrect and $\hat{u}_j$ is correct, the *ranking* performance of $\kappa$ is defined, as a binary-specialized instance of the general ranking risk (Galil et al., 2023), as the probability that $\kappa$ assigns a higher confidence score to the correct unit:

$$\mathbf{Pr}\left[ \kappa(\mathbf{x}^{(1)}\hat{\mathbf{u}}_{1:i-1}^{(1)}, \hat{u}_i^{(1)}) < \kappa(\mathbf{x}^{(2)}\hat{\mathbf{u}}_{1:j-1}^{(2)}, \hat{u}_j^{(2)}) \right.$$
$$\left. \Big| \; \hat{u}_i^{(1)} \neq u_i^{(1)} \wedge \hat{u}_j^{(2)} = u_j^{(2)} \right], \quad (2)$$

where $\hat{\mathbf{u}}_{1:i-1}$ denotes the units generated prior to $\hat{u}_i$, and $\mathbf{x}\hat{\mathbf{u}}_{1:i-1}$ denotes the context on which the confidence of $\hat{u}_i$ is conditioned. The AUROC metric is often used to evaluate confidence ranking. Specifically, under a 0/1 loss, AUROC is equivalent to the probability defined in Equation 2 (Tortorella, 2006), making it a suitable metric for assessing the model's ability to rank generated units by correctness. While the Prediction Rejection Ratio (PRR) (Malinin & Gales, 2021) is also a standard ranking metric, we find it to be nearly perfectly correlated with AUROC (see Appendix G.4) and thus focus our analysis on the latter.

We follow the terminology of Wang et al. (2024a) and distinguish between *macro-AUROC* and *micro-AUROC*, which differ in how ranking comparisons are formed. Macro-AUROC pools evaluation units across all generations, measuring whether correct units receive higher confidence than incorrect ones across the dataset. micro-AUROC, in contrast, evaluates ranking *within individual generations* by restricting comparisons to units from the same generation by enforcing $\mathbf{x}^{(1)} = \mathbf{x}^{(2)}$ in Equation 2. For example, a model may achieve high macro-AUROC by ranking correct units from easy generations above incorrect units from harder ones, while still failing to distinguish correct and incorrect units within any single long-form output, a failure

captured by micro-AUROC. As a result, macro-AUROC reflects global confidence ordering across tasks and prompts, whereas micro-AUROC reflects the model's ability to rank reliability *within a single long-form response*.

# 3. Related Work

textbfLLM uncertainty estimation methods can be categorized by the signals they utilize. *Information-based* methods aggregate token-level log-probabilities to derive unit-level scores, such as perplexity or entropy (Min et al., 2023; Geng et al., 2024; Huang et al., 2025b). We prioritize this approach for its theoretical grounding and computational efficiency, while also evaluating *Verbalization* methods (Kadavath et al., 2022; Tian et al., 2023) as a complementary signal. Other methods include *Internal-state* and *Sampling* methods, which we discuss further in Appendix D.

**Uncertainty Estimation Benchmarks for LLMs:** Existing evaluation frameworks are largely split between constrained deterministic tasks and open-ended generation with proxy evaluation. Short-form benchmarks provide an unambiguous ground truth for precise metric calculation, e.g., multiple-choice or arithmetic Q&A (Ye et al., 2024; Xiong et al., 2024; Yona et al., 2024b; Wang et al., 2024b; Rein et al., 2024; White et al., 2025) but fail to capture the structural complexity of long-form outputs. Recent efforts have moved toward evaluating free-form text (Fadeeva et al., 2023; Vashurin et al., 2025), often decomposing generations into fine-grained units (Wang et al., 2024a; Zhang et al., 2024c; Min et al., 2023; Yang et al., 2025; Yona et al., 2024a; Zhang et al., 2024a). However, these methods typically trade rigor for complexity: they rely on noisy model-based judges or proxy metrics rather than deterministic ground truth. Closest to our motivation, Santilli et al. (2025) show that the choice of correctness function alters how confidence functions are ranked. SALT takes a complementary route by removing correctness estimation from the evaluation pipeline, combining structural depth of long-form generation with deterministic unit-level ground truth.

# 4. Constructing the SALT Benchmark

We introduce SALT, a dynamic benchmark comprising six procedurally generated tasks designed with a single deterministic ground-truth, to evaluate uncertainty estimation in open-ended, long-form generation at unit-level resolutions. To address the limitations of existing static evaluations, which often rely on short-form answers or external judges, we construct our tasks to ensure deterministic computability of correctness at the unit level. In this section, we first outline the five core design objectives governing SALT (Section 4.1). We then detail the task domains (Section 4.2) and describe our pipeline for response processing

and alignment (Section 4.3).

## 4.1. Design Objectives

SALT is designed to satisfy five key objectives essential for zero-noise evaluation of uncertainty estimation in long-form generations.

(1) **Evaluation of Long-Form Generation**. Tasks require non-trivial, multi-atom outputs rather than single tokens. This elicits the localized failure modes—such as hallucinations or "context rot"—characteristic of extended generations. By providing the necessary length and complexity, this objective establishes the substrate for fine-grained uncertainty analysis across the entire output.

(2) **Evaluation on Model-Generated Outputs**. We assess uncertainty on *actual model generations* rather than predefined candidate sets or post-processed representations of an output. Scores are computed from token probabilities and aggregated as defined in Section 2.2, ensuring assessment of the exact generated text without auxiliary post-processing.

(3) **Single Deterministic Ground Truth**. Instances are algorithmically composed so the complete ground truth is known *a priori*. This eliminates reliance on human or LLM-based judges and provides unambiguous correctness signals at the atom and line levels, which is critical for metrics sensitive to label noise (see Appendix I).

(4) **Unit decomposition**. Generations naturally decompose into units aligned with task semantics (e.g., arithmetic entries). This structure enables direct evaluation of accuracy, calibration, and ranking at atomic resolution throughout the generation process, bypassing the auxiliary parsing, heuristic matching, or LLM-based alignment common in prior work (Min et al., 2023; Yan et al., 2025).

(5) **Unbounded Instance Generation**. All tasks are procedurally generated with tunable parameters, enabling the creation of an effectively infinite evaluation set. This prevents test-set memorization and ensures that uncertainty estimates are measured on genuinely unseen long-form generations.

## 4.2. Benchmark Tasks

SALT comprises eight tasks spanning mathematics, logic, code, scientific translation, Multi-Needle, rule induction, and spatial planning. These tasks are designed to elicit long, structured generations that decompose into task-aligned atomic units. The six tasks with complete model coverage total over 50k units and form the aggregate analysis in the main results, providing high-resolution insights throughout long-form generation. A detailed breakdown of tasks and examples is provided in Appendix A. Crucially, the tasks are selected to span a spectrum of semantic dependence structures. This range allows us to study correctness and uncertainty across tasks that differ in the information required to determine each target unit: from *local, output-independent mappings* (e.g., DNA Translation), through *global input-dependent computations* (e.g., Matrix Multiplication), to tasks with stronger ordering or logical dependencies (e.g., Multi-Needle and First-Order Logic). A detailed breakdown of task roles, examples, and a formal analysis of these dependencies is provided in Appendices A and K.

**Math: Matrix Multiplication** and **Kronecker Product** test precision across long contexts and high-dimensional dependencies, but represent distinct reasoning challenges. The former requires global aggregation, where each output element depends on an entire row and column, while the latter tests local, repetitive mapping. SALT generates matrices with configurable dimensions, constrained according to the problem size $L$ as summarized in Table 3, but allows for arbitrary scaling to test the limits of context windows. Each numerical entry in the resulting matrix constitutes an individual atom for evaluation.

**Coding:** The **Code** task requires models to predict the output of a Python function for a given input. We use validated solutions to LeetCode problems as the function and devise inputs that adhere to the problem constraints. Ground-truth outputs are obtained by executing the source function with these devised inputs in a deterministic Python environment. This task evaluates the model's ability to trace code logic and simulate execution over complex data structures.

**Logic: First-Order Logic** probes deductive and procedural reasoning over extended outputs. In the First-Order Logic task a model is prompted with formulas and known variables' truth values, and is prompted to find all deducible variables and their truth values.

**Translation:** The **DNA Translation** task evaluates deterministic sequence-to-sequence over long generations. Each nucleotide is mapped to its complement via a fixed rule (A $\leftrightarrow$ T, C $\leftrightarrow$ G), requiring no biological expertise. Despite its simplicity, maintaining precision across long sequences is difficult; models achieve an average precision of only 37.8%, comparable to the complex First-Order Logic task (39.5%) and far below Code (68.0%). This underscores the challenge of sustained attention in long, repetitive mappings.

**Multi-Needle:** The **Words Collection** task requires identifying and ordering all items of a specific category (e.g., animals) from a passage (Bird & Loper, 2004). As a structured sequence of individual atoms, it enables precise diagnosis of both omissions (recall errors) and redundant units (precision errors) during long-form generation.

**ARC-AGI:** The task tests generalization on highly abstract tasks from a few examples. Given input-output grid examples and a test input, the model must produce the corresponding output grid. SALT treats each cell as an atom.

**Maze: Maze** requires models to find the unique shortest path in a grid with obstacles, evaluating spatial planning under global constraints. Each step along the path is an atomic unit, allowing for precise localization of navigation and optimality errors.

ARC-AGI and Maze are evaluated separately in Section A.1 due to incomplete model coverage under compute constraints and are excluded from the main-body analysis.

### 4.3. Response Processing and Alignment

To transition from open-ended generation to the formal definition of atoms in Section 2, we employ a two-stage parsing pipeline. First, to isolate the target generation from conversational fillers or reasoning preambles, we require models to produce a *fenced final answer sequence* (e.g., enclosing the solution in specific delimiters). This allows deterministic extraction via regular expressions, removing formatting variability without altering the generated content.

Second, the extracted sequence is segmented into units according to task-specific delimiters (e.g., commas for matrices, spaces for Multi-Needle). Given the deterministic structure and unit-decomposition properties of SALT, both the ground truth $\mathbf{u}$ and the generated sequence $\hat{\mathbf{u}}$ are ordered sequences of units, aligned strictly by index. This allows for binary correctness labeling of each unit $\hat{u}_i$ via exact string matching, ignoring stylistic whitespace while strictly penalizing content errors. We additionally explore an alternative evaluation setting using dynamic sequence alignment in Appendix H. We find that alignment impacts certain tasks, and discuss why the non-alignment setting remains the most realistic for evaluating long-form reliability.

**Consistency with Established Benchmarks.** We examine how model performance on SALT aligns with well-established benchmarks by comparing the precision-based rankings of over 50 LLMs against *MMLU-Pro*, *GPQA Diamond*, and *LLM-Arena*. We observe strong Spearman correlations: 0.73 with LLM-Arena, 0.84 with MMLU-Pro, and 0.85 with GPQA-Diamond. These correlations indicate that SALT induces model orderings broadly consistent with existing evaluation standards.

## 5. Results

In our figures, model markers use the developer's company logo. Marker size scales with parameter count. Models without a parent family are shown as grey circles. For closed models with undisclosed parameters, we assume a large scale and set marker sizes accordingly.

The results proceed in two parts. First, Section 5.1 study correctness dynamics in long-form generation, showing how prior errors and answer-context length shape future correct-

ness. Second, Section 5.2 evaluate uncertainty estimation, separating calibration from ranking and analyzing how this distinction interacts with post-hoc calibration, confidence functions, granularity, transferability, and reasoning.

### 5.1. Correctness Dynamics in Long-Form Generation

Prior benchmarks often rely on short-form answers or coarse evaluations, lacking the granularity to track how model correctness evolves throughout a long generation. A key advantage of SALT is its ability to decompose outputs into sequential, verifiable units, enabling high-resolution monitoring of correctness as the generation unfolds.

Leveraging this capability, we observe that models are more accurate early in the generation than later (Figure 2). Figure 3 further shows a strong association between future correctness and the correctness history of the generated prefix: generations with perfect prior records usually remain correct, while those dominated by prior errors consistently fail. Even a small fraction of prior mistakes correlates with disproportionate degradation; for example, 15% incorrect prior atoms reduces subsequent correctness to approximately 50%. Additional observational analyses are provided in Appendix B.1.

These observational trends are consistent with error accumulation in long-form generation, connecting to prior observations of hallucination snowballing (Zhang et al., 2024d) and long-context degradation, often referred to as context rot (Shi et al., 2023; Hong et al., 2025). However, because answer-context length, and prefix correctness naturally covary, we next use controlled interventions to disentangle their roles.

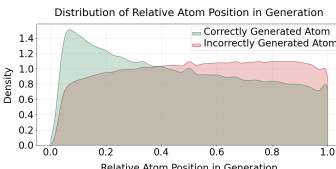

*Figure 2.* Probability density of atom correctness relative to its position within the generation. The x-axis represents normalized length (e.g., $x = 0.6$ is the 60th atom in a 100-atom sequence).

**How Context Correctness and Answer Length Shape Future Correctness**. To test whether the correctness of the generated prefix itself affects future correctness, we perform a controlled counterfactual intervention on the model's own answer context. For a target atom, we keep the original prompt fixed, and modify preceding atoms, either "corrupting" correct ones or "fixing" incorrect ones, and regenerate the target. Each modified prefix is compared against the original-prefix baseline for the same model, sample, and target position. We analyze two intervention treatments: changes in global prefix correctness and changes in local

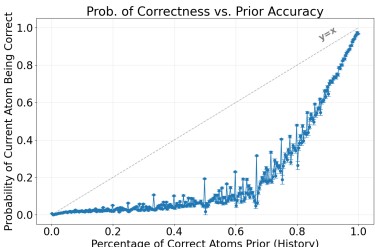

*Figure 3.* Probability of atom correctness (y-axis) vs. proportion of prior correct atoms (x-axis). Points represent bins of atoms grouped by their correctness history; markers show bin means, and error bars indicate standard deviation.

(last-10 atoms) correctness. Answer-context length is not intervened on directly; instead, we estimate its adjusted association with future correctness by evaluating length-response curves at fixed prefix-correctness levels (Figure 27). We use the DNA task for this experiment due to its low semantic dependence between atoms, which allows surgical edits without altering the ground truth of subsequent units. Effects are estimated via cubic B-spline regressions (de Boor, 1978), comparing modified prefixes against original baselines while controlling for answer length. Full details are provided in Appendix B.2.

Figure 4 highlights that both global and local prefix correctness affect $P$(next correct), with global correctness carrying the larger signal. When controlling for mutual adjustment, the gap between the independent signals grow greater. Residual global effect roughly $2\times$ larger than that of local last-10 atoms correctness (Figure 25). The intervention curves are nonlinear and asymmetric, with corruption more damaging than correction is helpful.

Answer-context length also affects future correctness, but it does not behave like the main driver of cascading failure. At fixed prefix-correctness levels, increasing answer-context length produces early degradation across regimes, indicating an independent length-related burden rather than a purely correctness-mediated effect (Figure 27). Yet this burden saturates quickly: the controlled analysis in Figure 26 shows that 95% of the asymptotic drop occurs by 66.6 atoms, and removing correctness controls changes the magnitude of the drop from $-0.43$ to $-0.50$ without changing the timeline. Together, these results suggest that prefix correctness governs the trajectory of future correctness, while answer length contributes a bounded background degradation.

Overall, these results support a path-dependent view of long-form correctness. The quality of the generated prefix directly shapes future correctness, with global prefix correctness carrying the strongest independent signal. Answer-context length adds a separate burden, but one that is bounded and saturates early. Thus, downstream errors are not merely a byproduct of generating longer answers; they are strongly shaped by the correctness of the answer context

the model has already produced.

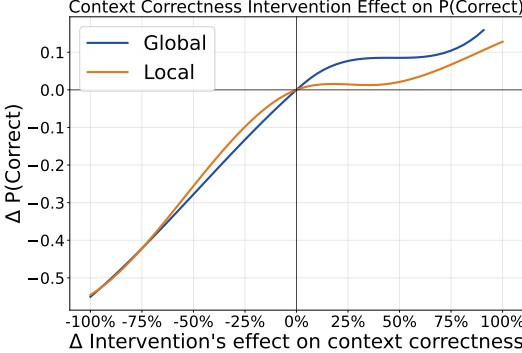

*Figure 4.* Effect of global and local (last 10 atoms) prefix-correctness perturbations on future correctness.

## 5.2. Uncertainty Estimation

SALT reveals that uncertainty estimation in long-form generation depends not only on the model, but also on how confidence is measured, calibrated, and evaluated. We therefore begin with the primary empirical effects, examining granularity and reasoning as two settings where confidence ranking changes substantially. We then analyze the confidence functions and calibration methods that underlie these measurements. Finally, we test whether the observed trends transfer beyond SALT to established benchmarks.

1) **The High-Resolution Ranking Gap.** We observe a sharp granularity gap in confidence ranking: Macro-AUROC is substantially more effective at the *line-unit* level than at the *atomic-unit* level. While line-level evaluation provides a clear signal, atomic-level ranking often struggles; for instance, Micro-AUROC scores for many models cluster between 0.45 and 0.7, indicating performance barely superior to random guessing (Figure 33c). This trend is statistically significant ($p = 7.5 \cdot 10^{-9}$, Figure 5), whereas calibration remains more stable across granularities (Figure 43). These findings reveal a high-resolution failure mode: raw confidence signals often fail to distinguish individual correct atoms from incorrect ones, even when they succeed for coarser units. Score-density analyses confirm this, showing substantial overlap between correct- and incorrect-atom distributions (Figure 44). By exposing this weak atomic-level separability through deterministic labels, SALT motivates the development of confidence functions specifically designed for high-resolution units. We thus treat line-unit Macro-AUROC as the primary ranking measure, as discussed further in Appendix J.

2) **The Reasoning Trade-off**. Having exposed a granularity-dependent ranking gap, we next investigate the reasoning effect on ranking. To that end, we examine two approaches that encourage a reasoning process in LLMs: Chain-of-Thought (CoT) prompting (Wei et al., 2022), an inference-

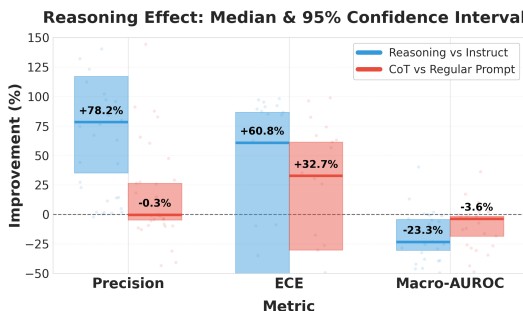

Figure 5. Macro-AUROC at atomic-unit and line-unit resolutions.

*Figure 6.* Impact of reasoning methods on Precision, ECE, and AUROC. Each marker represents the relative change between two identical models, differing only by CoT prompting or reasoning-specific training. Markers above the zero line represent an improvement in that metric relative to the base model's performance, while markers below represent a degradation.

time reasoning, and models trained specifically to reason (Jaech et al., 2024; DeepSeek-AI, 2025). To isolate these effects, we compare paired Instruct and Reasoning models of identical architecture and size, and applying CoT only to the Instruct variants. Our results reveal a reasoning trade-off: a systematic inverse relationship between accuracy and ranking ability (Figure 6).

While reasoning improves Precision (78.2%) and ECE (60.8%), consistent with prior work (Yoon et al., 2025), it simultaneously degrades AUROC (-23.3%). This effect scales with the reasoning mechanism's intensity: CoT prompting, a soft reasoning inducer, yields modest precision gains and slight ranking reductions, whereas specialized reasoning models exhibit dramatic precision improvements alongside a steeper decline in ranking ability.

Mediation analysis (Appendix G.5) clarifies the reasoning-calibration link. While reasoning models appear better calibrated than instruct counterparts, this improvement is statistically explained by increased precision ($p < 0.001$), with no significant residual effect from the reasoning mechanism itself ($p = 0.567$). Thus, reasoning enhances accuracy without independently improving uncertainty reliability.

Consequently, unlike ECE, ranking degradation emerges as a distinct reasoning side effect, independent of precision-driven calibration gains. We further explore the combined effect of reasoning methods, internalized through training and CoT prompting (Appendix G.6, Figure 38).

3) **Which Confidence Functions Work Best at Measuring Calibration and Ranking?** An important question

in uncertainty estimation is whether a single confidence function consistently outperforms in measuring different aspects of uncertainty estimation (ranking, calibration, etc.) (Galil et al., 2023; Fadeeva et al., 2023). To address this, we compare multiple logits-based (Huang et al., 2025b; Geng et al., 2024) and *verbalized-based confidence functions* using a paired Wilcoxon signed-rank test across models and tasks (Wilcoxon, 1945). We find that logits-based functions consistently and significantly outperform verbalized methods across both calibration and ranking metrics ($p < 0.05$). As shown in Figure 7, different confidence functions excel under different evaluation criteria. Several logits-based functions were found to be the most effective choice for calibration ($p < 0.05$), with Max Probability at the top in a tiny margin. In contrast, Logprobs Sum (see Table 7) excelling in ranking, significantly outperforming other functions in terms of AUROC. Notably, the identity of the best-suited confidence function for each metric is often invariant to the unit's resolution: the same confidence functions dominate calibration and ranking evaluations at both the atomic and line levels. Full details of the statistical tests and findings are supplemented in Appendix G.1.

4) **Calibration Methods Reveal a Calibration-Ranking Trade-off.** Logit-based confidence functions often require post-hoc mapping to $[0, 1]$ for calibration assessment. We compare several methods, including Z-score normalization followed by a Sigmoid transform, Min-Max normalization, and empirical binned calibration (Appendix E.1). To ensure a fair comparison, we report the best-performing confidence function for each scheme-metric pair (Section 5.2). As shown in Figure 8, binned calibration consistently minimizes ECE across both atomic and line resolutions. Conversely, binned calibration harms confidence ranking (Figure 28). This reveals that calibration methods optimizing for ECE can degrade ranking, while those preserving ranking structure may yield poor calibration. Accordingly, our

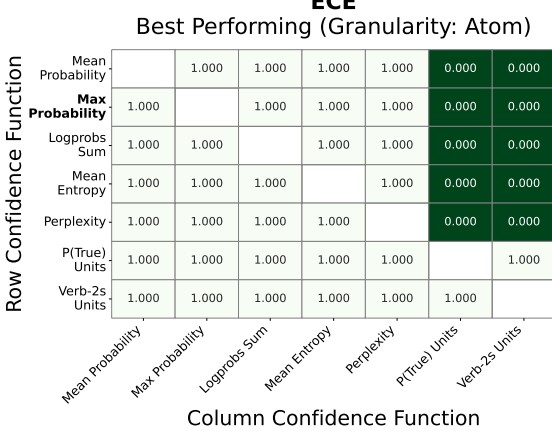

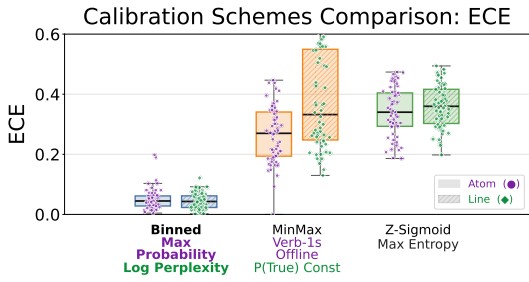

*Figure 8.* Comparison of post-hoc calibration schemes. Bold labels indicate the best-performing method, with the corresponding optimal confidence function listed below. Purple and green markers denote atomic- and line-unit granularities, respectively.

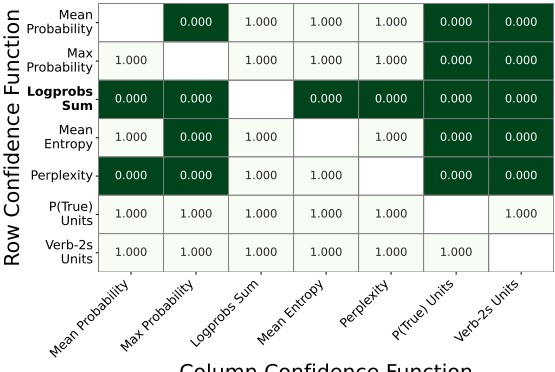

*Figure 7.* Statistical dominance of confidence functions. Green cells denote that the row function significantly outperforms the column function ($p < 0.05$, one-sided Wilcoxon signed-rank test). Evaluations are shown for the atomic-unit resolution.

reported uncertainty analyses use the best-performing calibration method for each metric, ensuring fair comparisons.

5) **Uncertainty Estimation Transferability.** Finally, we examine whether SALT's uncertainty trends generalize by comparing confidence-function rankings on SALT against those from *AIME* and *MMLU-Pro*. While agreement is stronger for AIME, it is non-uniform and highly sensitive to calibration methods. Specifically, AUROC rankings align best under Z-score and Sigmoid transforms, while ECE rankings favor binned calibration. This "calibration-regime effect" suggests that the post-hoc mapping used to derive probabilities largely dictates a metric's perceived transferability. Granularity also modulates agreement. Line-level evaluation yields the most stable Spearman correlations for AIME (0.68 AUROC, 0.74 ECE), likely because AIME targets coarse, answer-level correctness. In contrast, the weaker atomic-level correlation suggests that SALT's fine-grained resolution captures nuanced uncertainty structures invisible to coarse benchmarks. Thus, SALT's findings are transferable but conditional: confidence rankings generalize

well to AIME at line-level resolution, while MMLU-Pro agreement remains weak. See Appendix G.3 for details.

# 6. Conclusion

In this work, we introduced SALT, a long-form generation benchmark that shifts uncertainty evaluation from coarse, noisy judges to high-resolution, deterministic ground truth. By eliminating label noise, SALT analyzes over 50 LLMs, exposing critical dynamics in calibration and ranking.

Our findings reveal that long-form correctness is path-dependent, as prefix quality shapes future accuracy while length effects is bounded. Furthermore, effective ranking signals are resolution-dependent; models struggle with individual atoms but show reliability at coarser granularities, highlighting the need for confidence functions optimized for high-resolution units. Finally, we identify a reasoning trade-off: reasoning mechanisms (CoT or specialized training) boost accuracy but systematically degrade confidence ranking, making models less able to distinguish their own errors as they become more "convincing."

These results suggest that current scaling paradigms and reasoning mechanisms are decoupling capability from reliability. For risk-critical applications, high accuracy is insufficient if it is accompanied by brittle confidence estimates. SALT provides a necessary bedrock for future research to address this widening gap, moving beyond aggregate correctness to ensure that as models scale in intelligence, they also scale in the awareness of their own limitations.

Moving forward, the procedural nature of SALT enables scalable interventions. Its high-resolution error signals offer a rich substrate for training granular confidence classifiers, surpassing methods reliant on coarse sequence-level labels. Furthermore, this dense feedback mechanism is naturally suited for Reinforcement Learning (RL) post-training to mitigate the observed reasoning-ranking trade-off.

## Impact Statement

This work aims to advance uncertainty estimation by improving the reliability of long-form generation. While our research has broad societal implications, particularly for the safety of high-stakes AI applications, we identify no specific consequences requiring exceptional highlight here.

## Acknowledgments

We thank Yonatan Belinkov for the fruitful discussions. The research was partially supported by Israel Science Foundation, grant No 765/23

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

*Table 1.* Overview of SALT tasks and sample sizes

| Task name | Domain | Samples | Atoms |
|---|---|---|---|
| Code | Coding | 148 | 1,934 |
| First-Order Logic | Logic | 400 | 8,410 |
| Matrix Multiplication | Math | 300 | 863 |
| Kronecker Product | Math | 300 | 8,618 |
| DNA Translation | Translation | 300 | 28,972 |
| Words Collection | Multi-Needle | 300 | 6,000 |
| Total | | 1,748 | 54,797 |

*Table 2.* Pearson correlation between Precision and Recall across benchmark tasks.

| Task | Correlation |
|---|---|
| Code | 1.00 |
| First-Order Logic | 0.99 |
| Matrix Multiplication | 0.99 |
| Kronecker Product | 0.99 |
| DNA Translation | 0.99 |
| Words Collection | 0.92 |

# A. Benchmark Tasks

This appendix details the technical implementation and procedural generation of the SALT tasks introduced in Section 4.2. We provide a comprehensive breakdown of dataset statistics in Table 1, while Table 3 specifies the dimensionality constraints for the mathematical tasks. To ensure reproducibility, we include the exact prompt templates used across all tasks in Figures 9–14. The following subsections further analyze task-specific performance and the unique challenges posed by each domain.

**Task-Level Metric Normalization.** Note that the number of samples and atoms varies across tasks (Table 1). To ensure this imbalance does not skew the performance metrics, unless reported for each task independently, all evaluation metrics (precision, ECE, and AUROC) are first computed independently for each task. The final benchmark-level score is then obtained by averaging across tasks, ensuring that each task contributes equally to the overall evaluation, regardless of its size.

As mentioned in Section 2.3, we measure a very high Pearson correlation between the precision and recall across our tasks. The correlations range from 0.92 to 1.0, with an average of 0.98, as can be seen in Table 2.

Figure 15 compares task-specific precision across atomic and line-unit resolutions. We observe a dramatic gap between the atomic and line levels' performance, underscoring the necessity of fine-grained evaluation of long-form generations. Figure 16 exhibits how Calibration differs across tasks and task-dependent ECE comparison of atomic versus line levels. We observe that tasks exhibiting higher precision in Figure 15 also show lower ECE, aligned with the observed high correlation between precision and ECE in Section 5.2. Furthermore, unlike Macro-AUROC, which significantly favored the Line granularity, we observe a milder but opposite trend regarding ECE. We performed a one-sided Wilcoxon paired test to test the ECE preferred unit-level evaluation and find out that, on average, atomic-level performs better than line-level ($p = 3.06 \times 10^{-6}$). However, when inspecting each task individually, it is statistically significant only in the translation and Multi-Needle tasks. Additionally, Figure 17 illustrates the *redundant atoms percentage*—the proportion of generated atomic units that are redundant. For example, a 10-atom generation containing two redundant atoms would yield a 20% redundancy rate of atoms.

## A.1. Maze and ARC-AGI Evaluation

We additionally evaluate SALT on ARC-AGI and Maze, two tasks that extend the benchmark toward abstract induction and spatial planning. Due to incomplete model coverage under compute constraints, these tasks are not included in the aggregate main-body results, and we report them separately here. Despite this limitation, both tasks further illustrate the value of SALT's unit-level evaluation: ARC-AGI naturally decomposes grid outputs into cells, while Maze decomposes paths into sequential steps.

*Table 3.* Matrix dimension constraints in SALT for a given problem size $L$. Neither of the matrices is a Scalar ($1 \times 1$) matrix.

| Operation | Matrix shapes | Dimensional size |
|---|---|---|
| MatMul | $A : m \times k,\ B : k \times n$ | $mn = L,\ k \leq 3$ |
| Kronecker | $A : m \times n,\ B : p \times q$ | $mn \cdot pq = L$ |

**ARC-AGI.** Figure 18 shows a large gap between atom-level and generation-level precision. Models often recover a substantial fraction of the output cells, yet exact full-grid correctness remains much lower, demonstrating how generation-level evaluation hides partial structure recovered by the model. Ranking performance exhibits a related granularity effect: line-level and whole-generation AUROC generally provide stronger signals than atom-level AUROC (Figures 19a and 19b), consistent with the finding that confidence ranking is more reliable at coarser resolutions.

**Maze.** Maze exhibits a different pattern. As shown in Figure 20, both line-level and atom-level exhibit low precision, reflecting the difficulty of preserving longer valid path segments even when individual moves are partially correct. Figure 21 further shows a strong negative association between atom-level precision and Macro-AUROC on the evaluated models, suggesting that models better at finding paths are not necessarily better at ranking their correct and incorrect steps by confidence. While the limited model coverage prevents broad conclusions, this reinforces the paper's central observations that correctness and confidence ranking can decouple, especially in long, structured generations, and that models perform poorly out of the box confidence ranking.

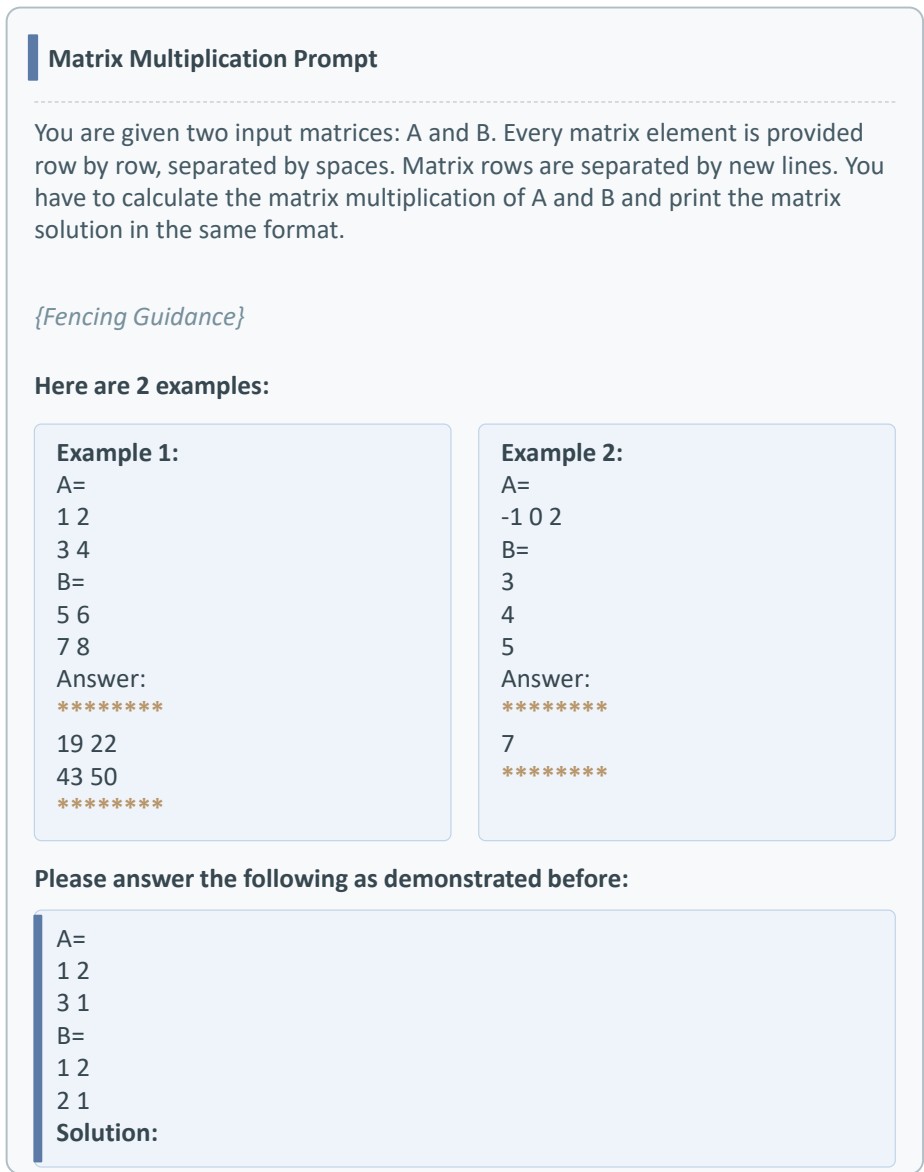

*Figure 9.* The Matrix Multiplication task prompt, demonstrating few-shot examples and strict output fencing instructions.

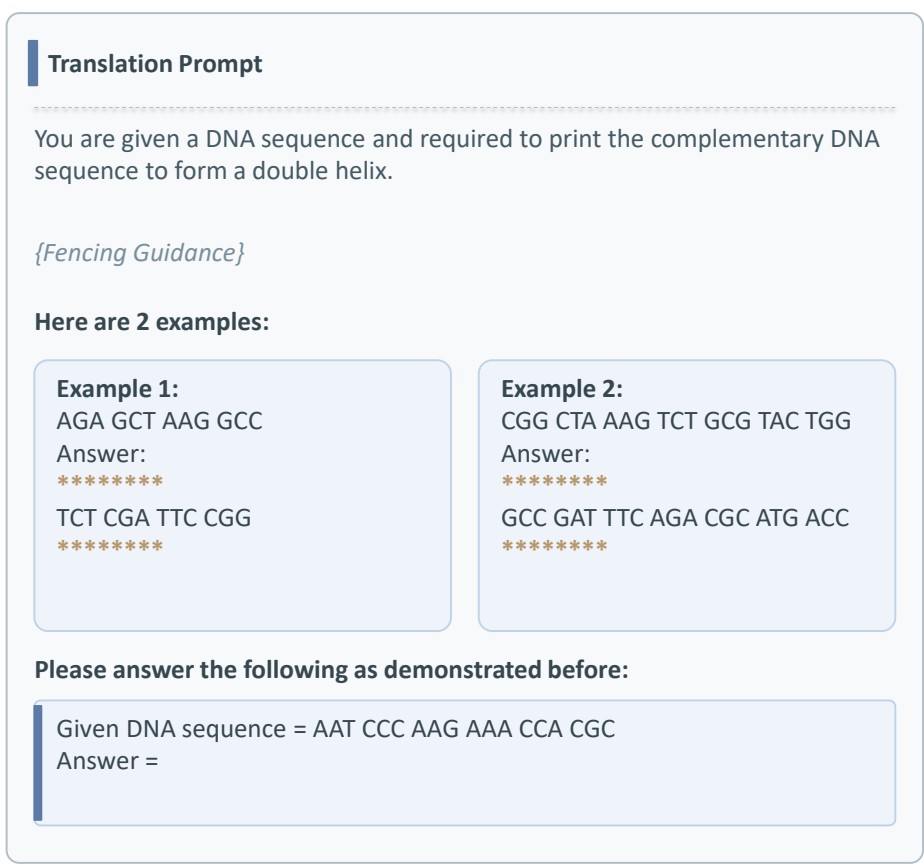

*Figure 10.* The Translation task prompt, demonstrating few-shot examples and strict output fencing instructions.

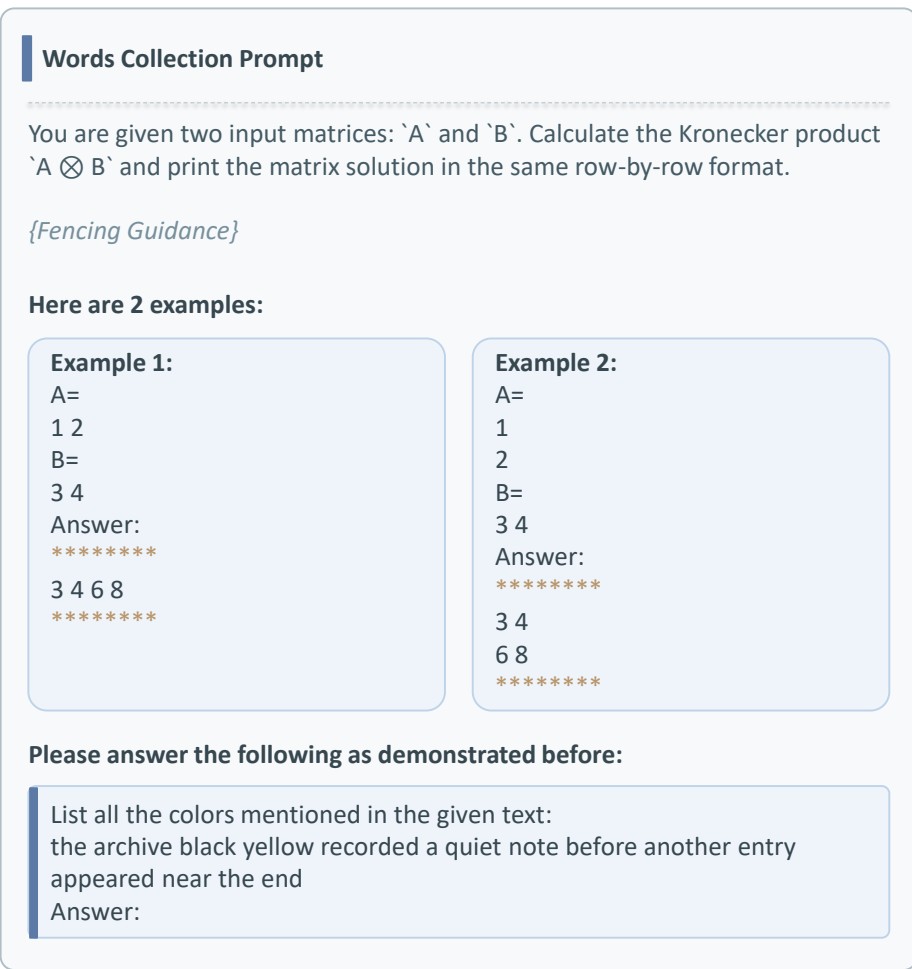

**Words Collection Prompt**

You are given two input matrices: `A` and `B`. Calculate the Kronecker product `A ⊗ B` and print the matrix solution in the same row-by-row format.

*{Fencing Guidance}*

**Here are 2 examples:**

**Example 1:**
A=
1 2
B=
3 4
Answer:
********

3 4 6 8
********

**Example 2:**
A=
1
2
B=
3 4
Answer:
********

3 4
6 8
********

**Please answer the following as demonstrated before:**

List all the colors mentioned in the given text:
the archive black yellow recorded a quiet note before another entry appeared near the end
Answer:

*Figure 11.* The Kronecker Product prompt, demonstrating few-shot examples and strict output fencing instructions.

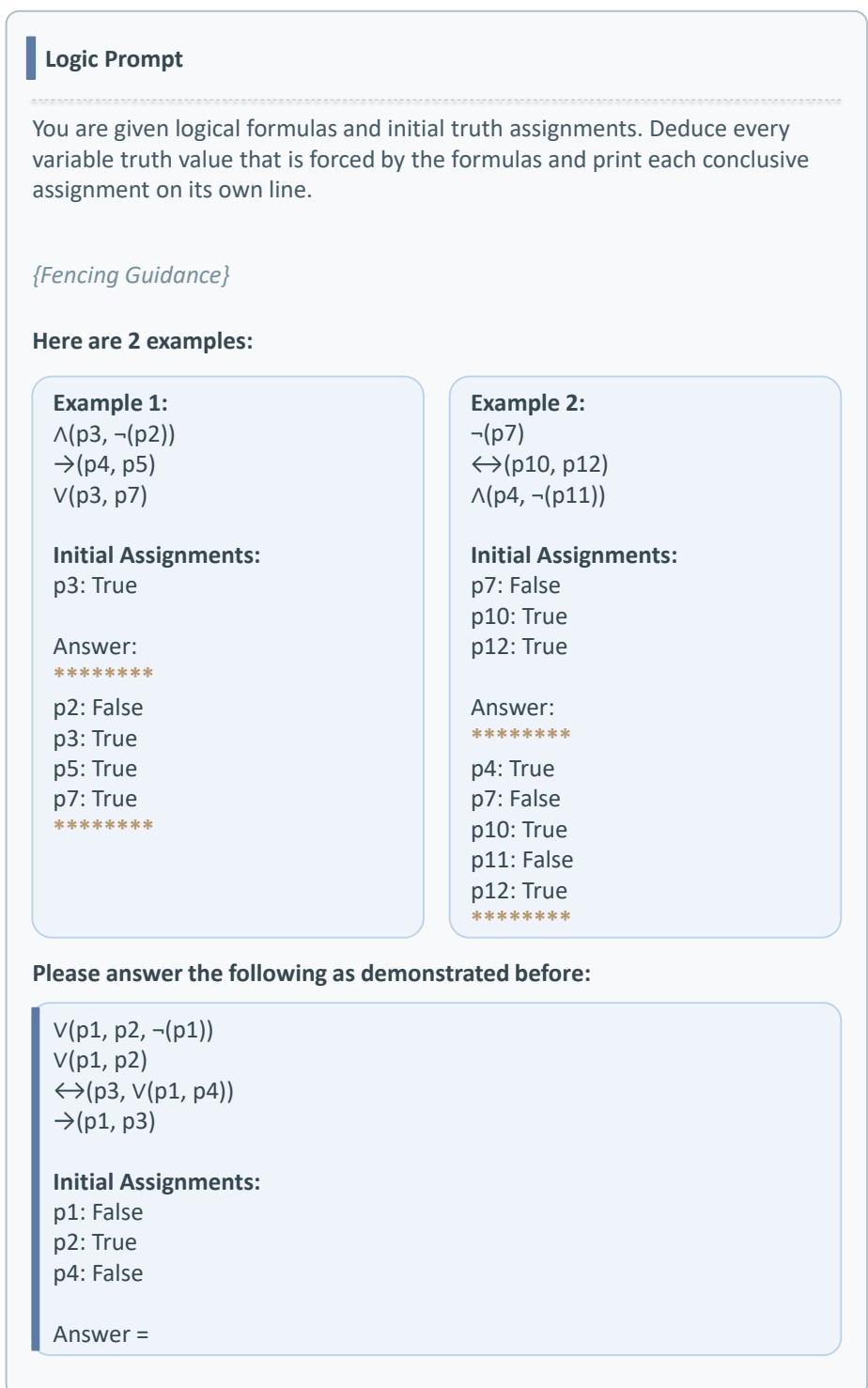

**Logic Prompt**

You are given logical formulas and initial truth assignments. Deduce every variable truth value that is forced by the formulas and print each conclusive assignment on its own line.

*{Fencing Guidance}*

**Here are 2 examples:**

**Example 1:**
∧(p3, ¬(p2))
→(p4, p5)
∨(p3, p7)

**Initial Assignments:**
p3: True

Answer:
********
p2: False
p3: True
p5: True
p7: True
********

**Example 2:**
¬(p7)
↔(p10, p12)
∧(p4, ¬(p11))

**Initial Assignments:**
p7: False
p10: True
p12: True

Answer:
********
p4: True
p7: False
p10: True
p11: False
p12: True
********

**Please answer the following as demonstrated before:**

∨(p1, p2, ¬(p1))
∨(p1, p2)
↔(p3, ∨(p1, p4))
→(p1, p3)

**Initial Assignments:**
p1: False
p2: True
p4: False

Answer =

*Figure 12.* The Translation task prompt, demonstrating few-shot examples and strict output fencing instructions.

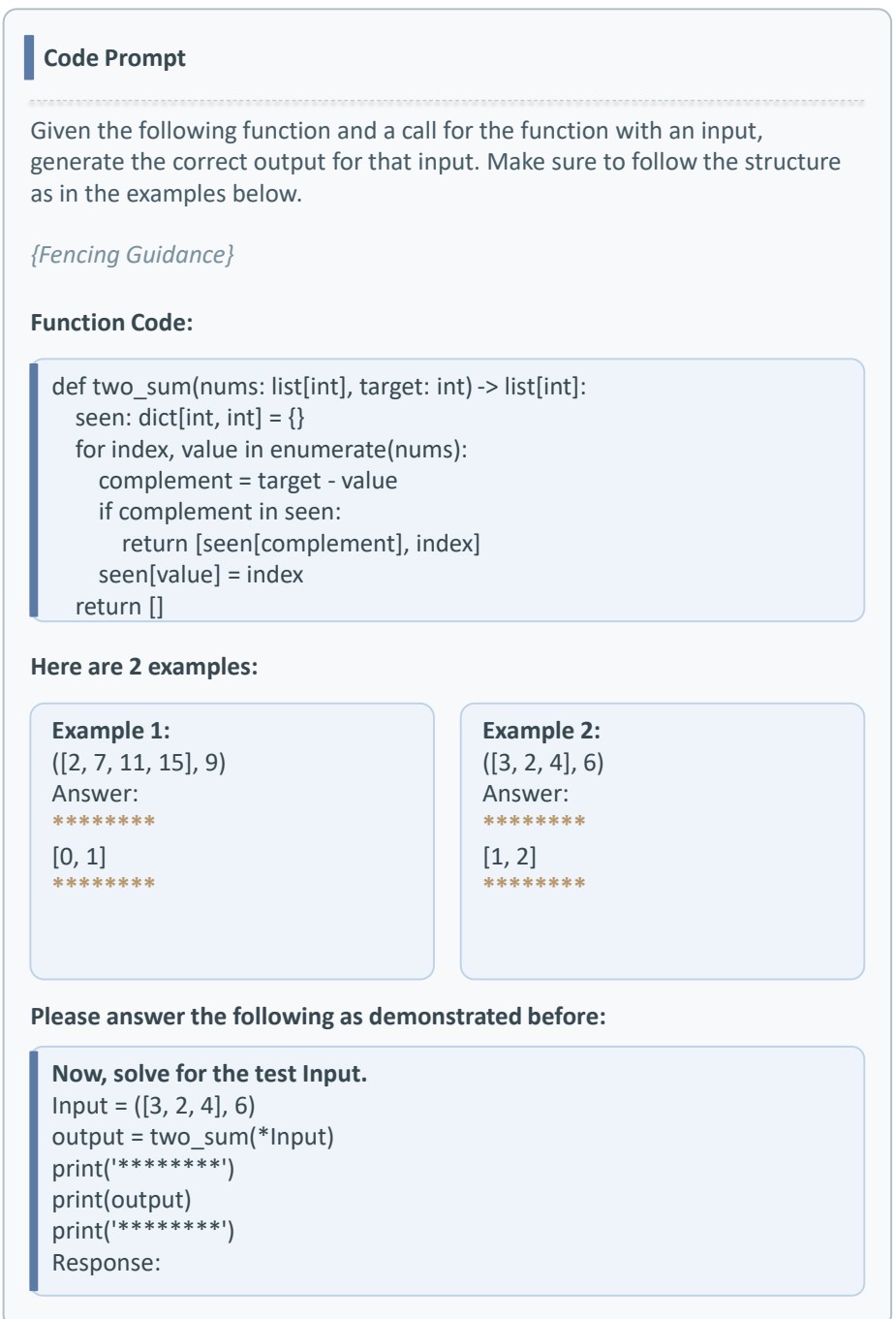

**Code Prompt**

Given the following function and a call for the function with an input, generate the correct output for that input. Make sure to follow the structure as in the examples below.

*{Fencing Guidance}*

**Function Code:**

```
def two_sum(nums: list[int], target: int) -> list[int]:
    seen: dict[int, int] = {}
    for index, value in enumerate(nums):
        complement = target - value
        if complement in seen:
            return [seen[complement], index]
        seen[value] = index
    return []
```

**Here are 2 examples:**

**Example 1:**
([2, 7, 11, 15], 9)
Answer:
********

[0, 1]
********

**Example 2:**
([3, 2, 4], 6)
Answer:
********

[1, 2]
********

**Please answer the following as demonstrated before:**

**Now, solve for the test Input.**
Input = ([3, 2, 4], 6)
output = two_sum(*Input)
print('********')
print(output)
print('********')
Response:

*Figure 13.* The Code task prompt, demonstrating few-shot examples and strict output fencing instructions.

> ### Words Collection Prompt
>
> For the following text, please list all the colors mentioned in it according to their order in the story.
> The delimiter between colors should only be a space ' '. Make sure to use only lower case characters in your final answer.
> For example, if the text was 'roses are red, violets are blue and watermelons are red and green' then your answer should be 'red blue red green'.
>
> *{Fencing Guidance}*
>
> **Here are 2 examples:**
>
> > **Example 1:**
> > List all the colors mentioned in the given text:
> > roses are red, violets are blue and watermelons are red and green
> > Answer:
> > ********
> > red blue red green
> > ********
>
> > **Example 2:**
> > List all the weekdays mentioned in the given text:
> > on monday the report moved from march to april before friday
> > Answer:
> > ********
> > monday friday
> > ********
>
> **Please answer the following as demonstrated before:**
>
> > List all the colors mentioned in the given text:
> > the archive black yellow recorded a quiet note before another entry appeared near the end
> > Answer:

*Figure 14.* The Words Collection task prompt, demonstrating few-shot examples and strict output fencing instructions.

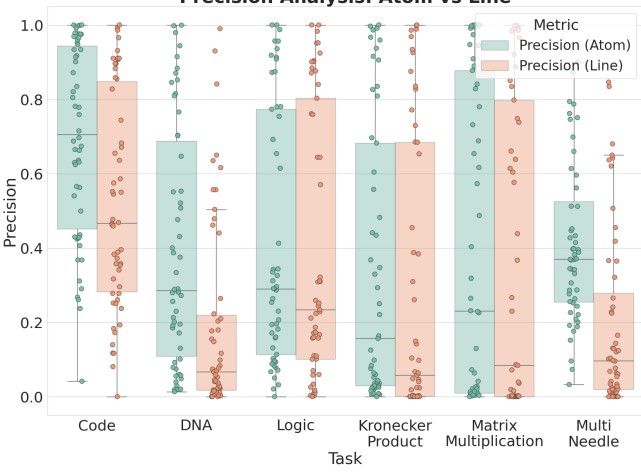

*Figure 15.* Comparison of precision across tasks at atomic and line-unit resolutions.

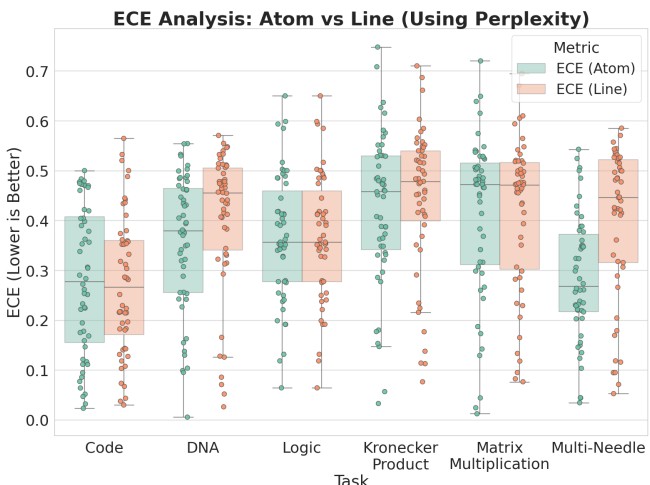

*Figure 16.* Comparison of ECE across tasks at atomic and line-unit resolutions. Confidence is computed using Perplexity, the optimal function for calibration established in Section 5.2.

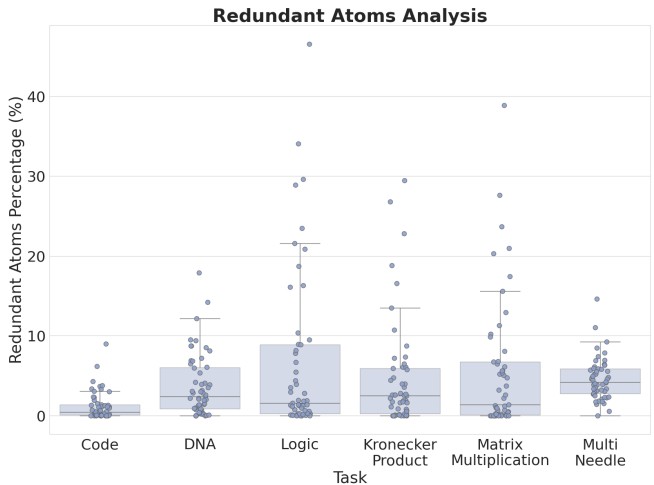

*Figure 17.* The measured relative redundant atoms in generations, across tasks.

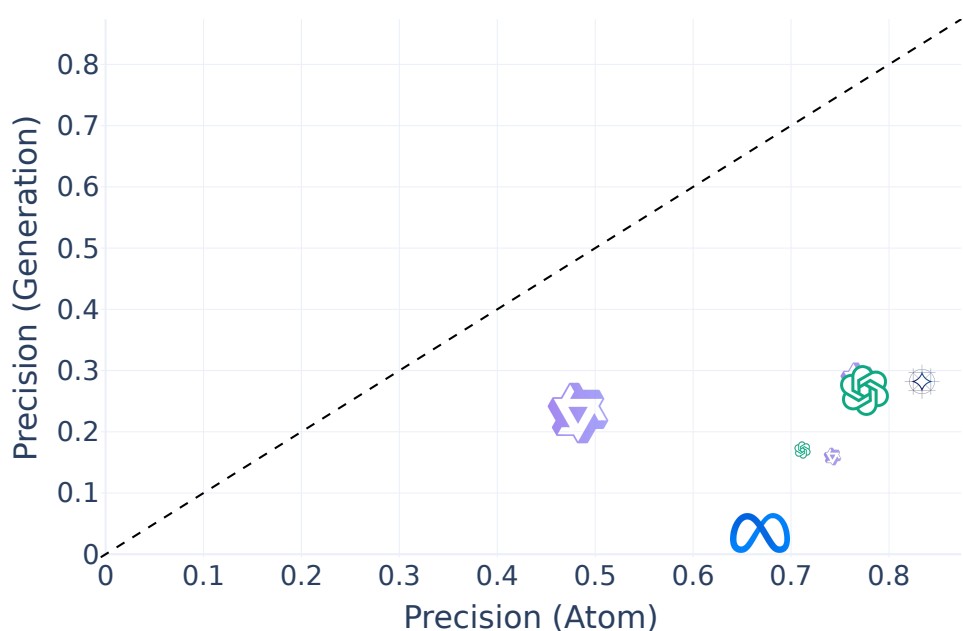

*Figure 18.* ARC-AGI precision at atomic and generation-level resolutions. Atom-level evaluation reveals high partial cell-wise correctness that is obscured by poor exact generation-level scoring.

## ARC_AGI: Macro-AUROC: Atom vs Line

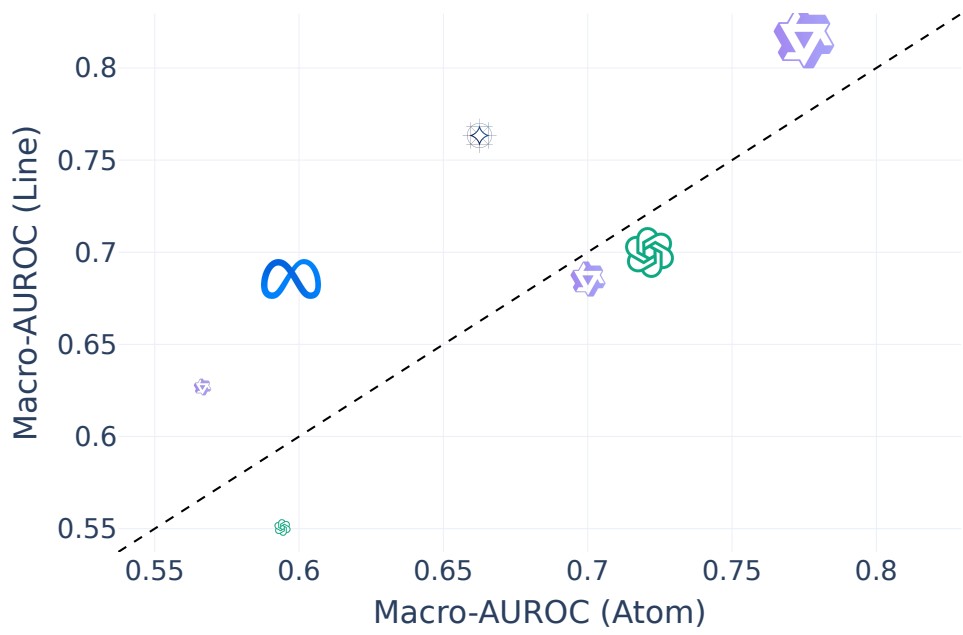

*(a)* ARC-AGI Macro-AUROC at atomic and line-level resolutions.

## ARC_AGI: Macro-AUROC: Atom vs Whole Generation

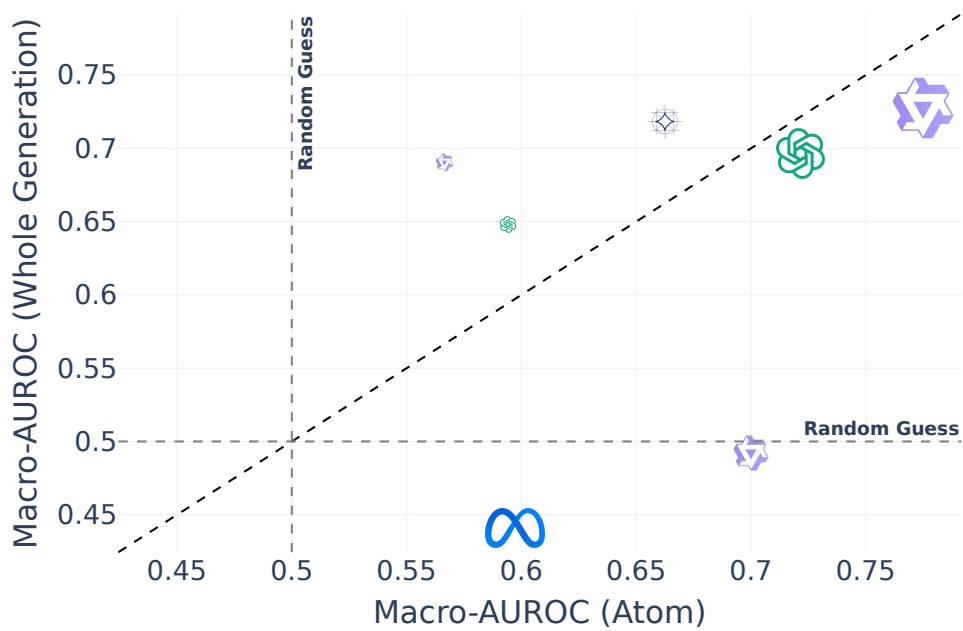

*(b)* ARC-AGI Macro-AUROC at atomic and generation resolutions.

*Figure 19.* ARC-AGI Macro-AUROC comparison at atomic with Line and generation resolutions. Both coarse resolutions often provide a clearer confidence ranking signal than atom-level ranking.

## Maze: Precision: Atom vs Line

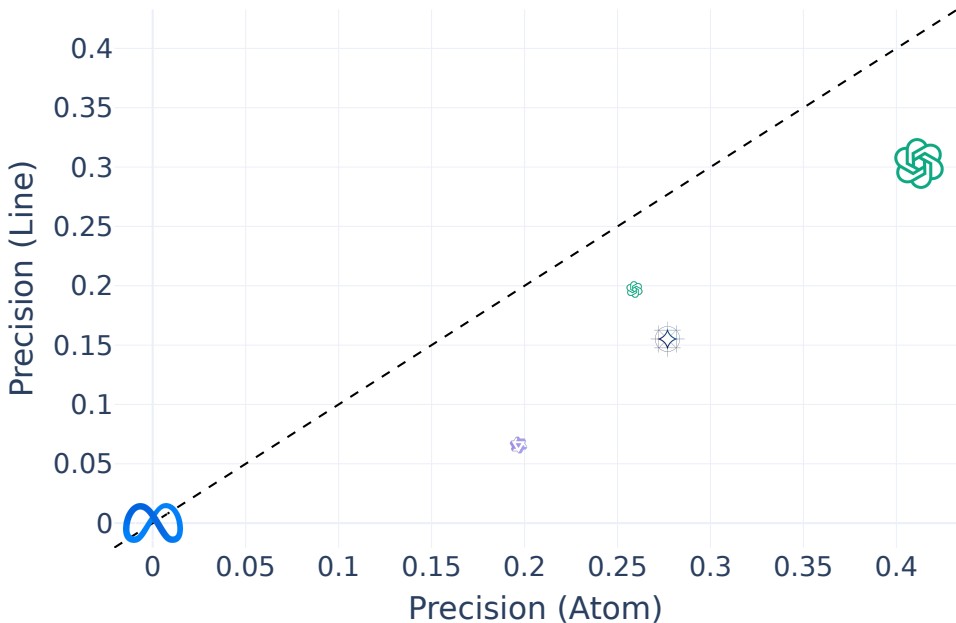

*Figure 20.* Maze precision at atomic and line-level resolutions.

## Maze: Precision vs Macro-AUROC - Atom granularity

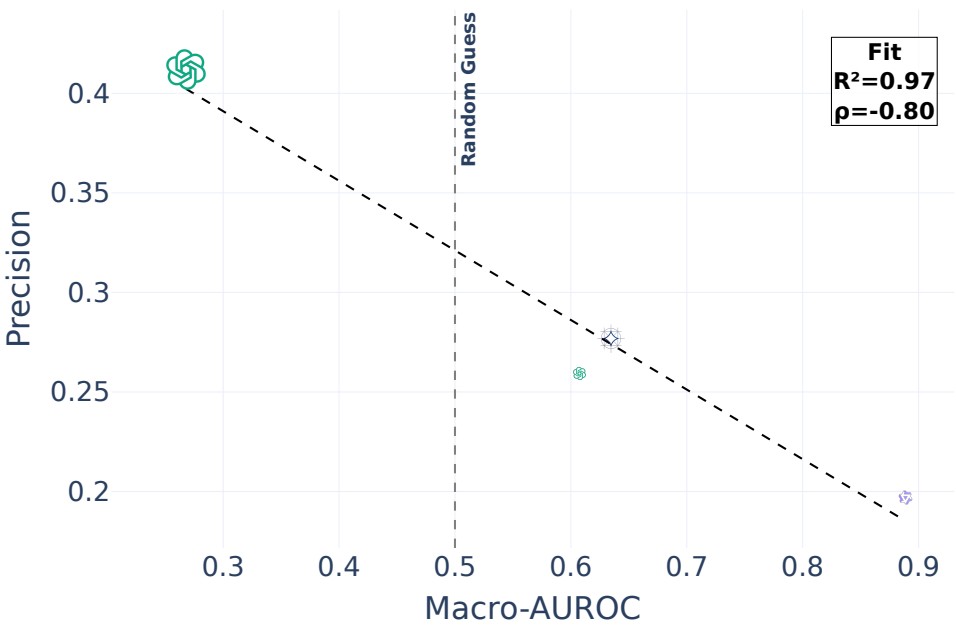

*Figure 21.* Maze atom-level precision versus Macro-AUROC. The evaluated models show a strong negative association, suggesting that higher path accuracy does not necessarily imply better confidence ranking, and vice versa.

# B. More on Correctness Analysis and Drivers

## B.1. LLM Correctness Given Prior Correctness Percentage

While Figure B.1 in Section 5.1 exhibit the relation between LLMs correctness and their prior mistakes, it makes a coarse aggregation across many LLMs, prompting methods, and tasks. Here, we prove it is a solid consistent behavior among sub groups as well.

First, we explore the discussed relation w.r.t each task in our benchmark. As shown in Figure 22, we observe the same pattern in the majority of task, with the exception of the Code and Kronecker product tasks. While LLMs are showing more robustness to past mistakes in the Code task, they are even more vulnerable in the Kronecker product task.

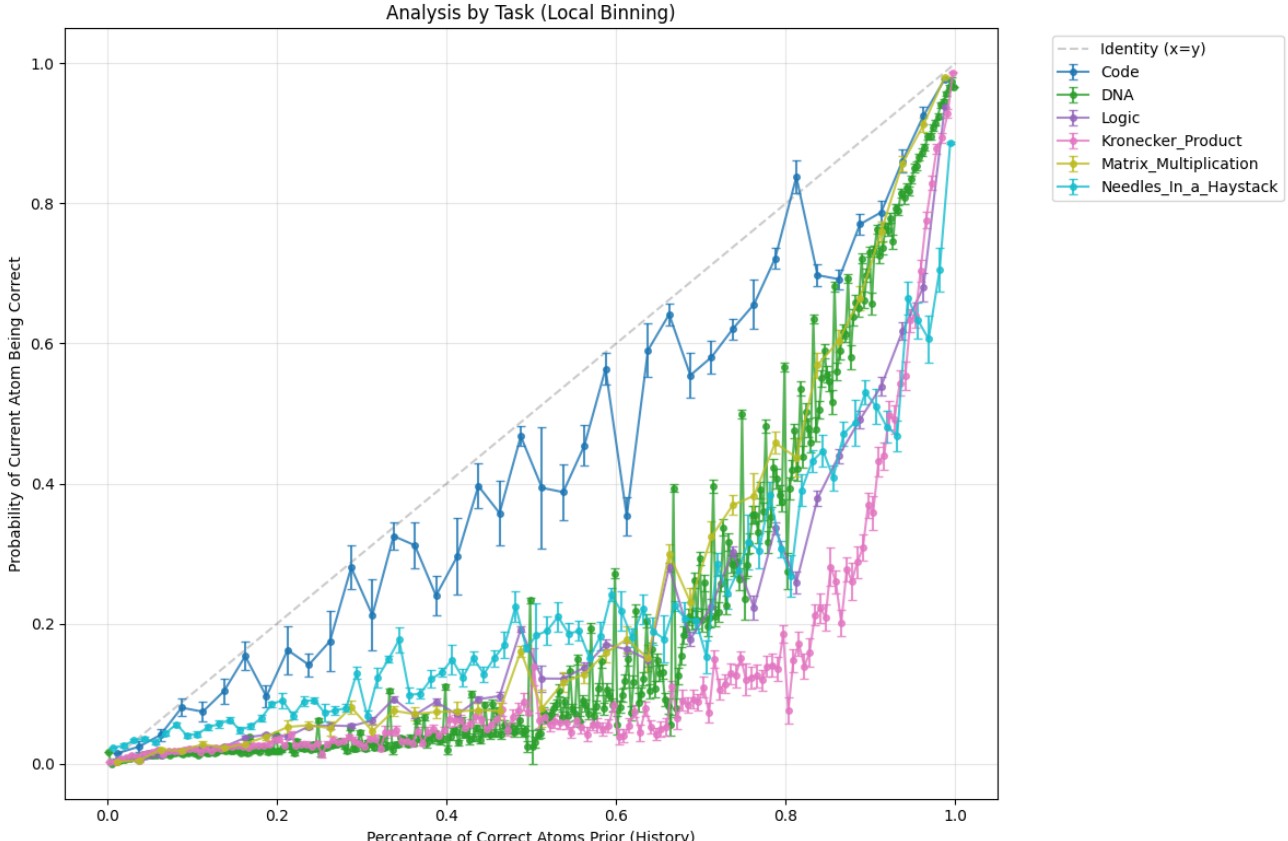

*Figure 22.* A comparison across the benchmark's tasks, of the probability of an atom being correct given the percentage or prior correct atoms.

Second, we validate whether different prompting method plays a role, where we compare a regular prompt with a Chain-of-Thought prompt. Figure 23 proves it makes no difference.

Reasoning models are showing extraordinary precision compared to instruction-tuned models. Lastly, we choose to compare instruction-tuned with reasoning models, in an attempt to check whether reasoning models not only improve accuracy, but also exhibit a different level of robustness to previous mistakes. In Figure 24, we show that both instruction-tuned and reasoning models lack the resilience to previous mistakes.

## B.2. Controlled-Intervention Prefix-Correctness Experiment and Length-response Association

This appendix gives the full identification, inference, and robustness details for the experiment summarized in the main body. The protocol intervenes on the model's own answer atoms; the analysis quantifies the causal effects of two perturbation treatments and one observational adjusted association.

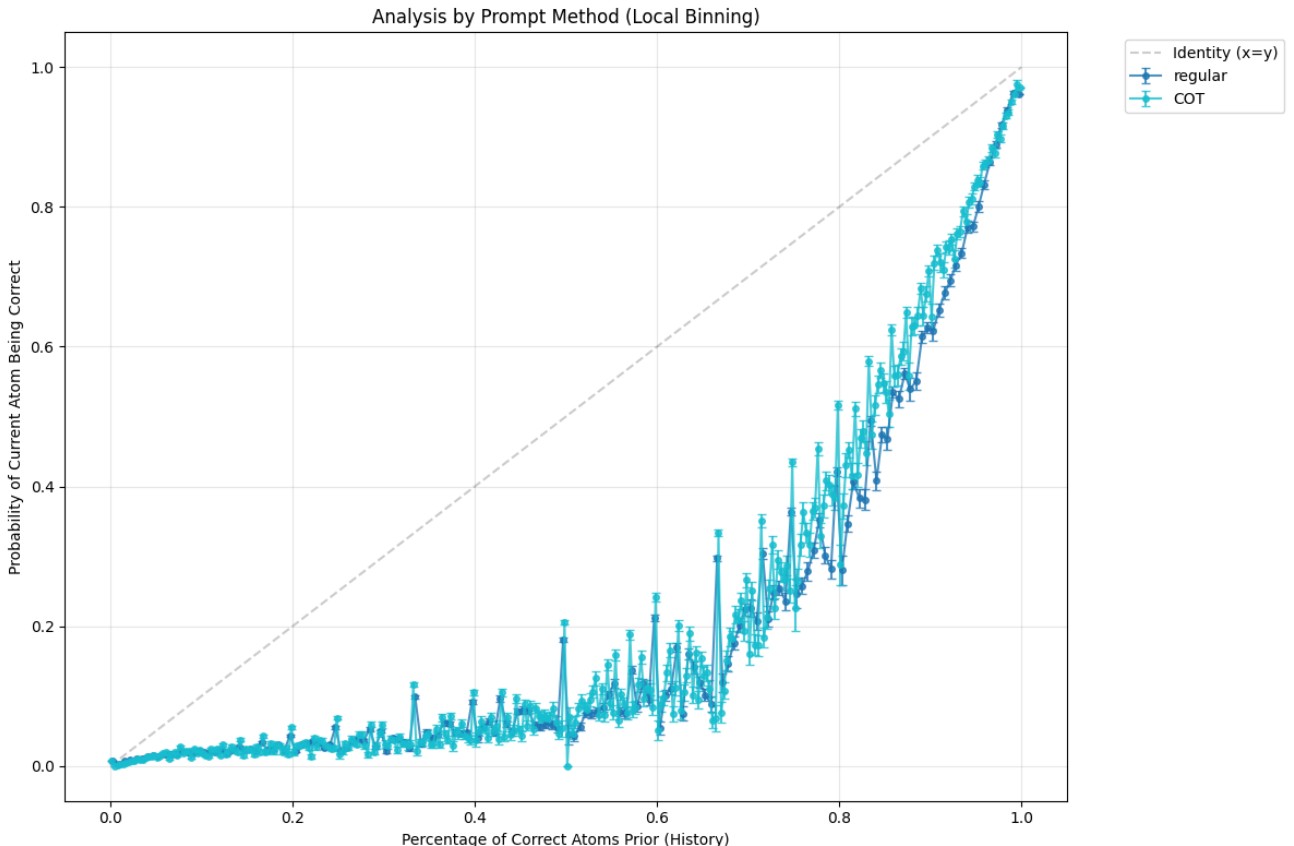

*Figure 23.* A comparison between regular and Chain-of-Thought prompting, of the probability of an atom being correct given the percentage or prior correct atoms.

### B.2.1. PROTOCOL AND TREATMENTS

Let $a_{m,s}^{(p)}$ denote the atom at position $p$ in model $m$'s original generation on sample $s$. For each triple $(m, s, p^\star)$ we retain the original prefix $a_{m,s}^{(1)}, \ldots, a_{m,s}^{(p^\star-1)}$ and produce a family of *modified* prefixes by deterministically flipping selected positions in $\{1, \ldots, p^\star - 1\}$: positions originally wrong are replaced with the gold atom (*correction*), and positions originally correct are replaced with a random non-gold atom (*corruption*). One row per triple has the original prefix and serves as the *paired baseline*. The outcome is

$$Y \;=\; \mathbb{1}[\, \text{model } m \text{ regenerates atom at } p^\star \text{ matching gold}\,] \in \{0, 1\}. \tag{3}$$

We consider three treatments:

- **Global prefix $\Delta$:** $T_{\text{global}} = C_{1:p^\star-1}^{\text{mod}} - C_{1:p^\star-1}^{\text{orig}}$, the change in fraction of correct atoms over the whole prefix.

- **Local last-10 $\Delta$:** $T_{\text{local}} = C_{p^\star-10:p^\star-1}^{\text{mod}} - C_{p^\star-10:p^\star-1}^{\text{orig}}$, restricted to the ten positions immediately before the target.

- **Answer-context length:** $T_{\text{len}} = p^\star - 1$. *Not* an intervention; included so we can adjust for it and quantify its association with $P(Y{=}1)$ conditional on prefix correctness.

### B.2.2. PAIRED-DIFFERENCE OUTCOME

For each row $i$, the matched baseline $b(i)$ has the same $(m_i, s_i, p_i^\star)$ and `is_original_prefix = 1`. The paired outcome

$$\tilde{Y}_i \;=\; Y_i \,-\, Y_{b(i)} \;\in\; \{-1, 0, +1\} \tag{4}$$

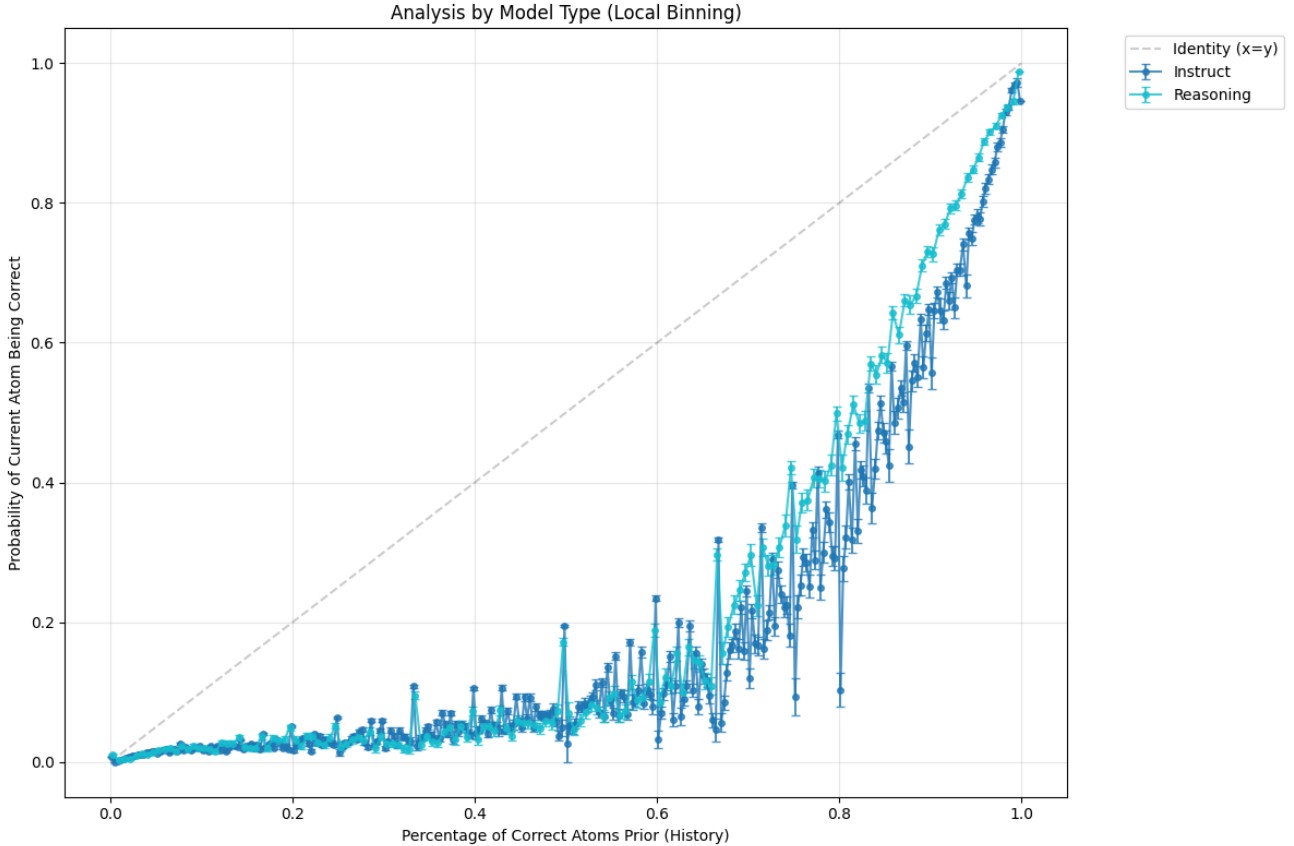

*Figure 24.* A comparison between instruction-tuned and reasoning models, of the probability of an atom being correct given the percentage or prior correct atoms.

equals $0$ on baseline rows by construction, anchoring the regression at the no-intervention point. Pairing absorbs all $(m, s, p^\star)$-level fixed effects — baseline difficulty, sample-specific lexical context, model-specific generation tendencies — at a granularity finer than any dummy variable could provide.

### B.2.3. SPLINE MODEL

For each intervention treatment $T \in \{T_{\text{global}}, T_{\text{local}}\}$ we fit, by OLS,

$$\tilde{Y}_i \;=\; \alpha + \sum_{k=1}^{5} \beta_k \, \phi_k(T_i) + \boldsymbol{\gamma}^\top \mathbf{X}_i + \varepsilon_i, \tag{5}$$

with $\{\phi_k\}_{k=1}^{5}$ a cubic B-spline basis (df=5, deg=3) and *explicit* boundary knots at $[\min_j T_j - \epsilon, \; \max_j T_j + \epsilon]$, $\epsilon = 10^{-6}(\max - \min)$. The explicit-bounds parametrization (i) suppresses Patsy's implicit intercept column, which would otherwise collide with the regression constant and silently corrupt the joint test on $\{\beta_k\}$, and (ii) admits prediction on per-model subsets whose own treatment range is narrower than the pooled range.

For $T_{\text{len}}$ we instead fit a logistic GLM,

$$\text{logit}\,\Pr(Y_i{=}1) \;=\; \alpha + \sum_{k=1}^{5} \beta_k \, \phi_k(T_{\text{len},i}) + \boldsymbol{\gamma}^\top \mathbf{X}_i, \tag{6}$$

because no paired baseline exists for "no length" ($T_{\text{len}}$ is always positive). The predicted curve is recentered at $T_{\text{len}}{=}0$ at plot time.

### B.2.4. CONFOUNDERS

The confounder vector $\mathbf{X}_i$ comprises:

- standardized target position $z = (p^\star - \overline{p^\star})/\mathrm{sd}(p^\star)$;

- original-prefix correctness $C^{\mathrm{orig}}$ (and $C_{10}^{\mathrm{orig}}$ for the local contrasts);

- model-identity dummies $\mathbb{1}\{m_i{=}m\}$;

- sample-identity dummies $\mathbb{1}\{s_i{=}s\}$;

- for each contrast's *mutually adjusted* variant, the *other* $\Delta$ treatment as a linear covariate.

### B.2.5. SIGNIFICANCE VS. NONLINEARITY

We distinguish two null hypotheses per contrast:

1. **Significance** ($H_0^{\mathrm{sig}} \colon \beta_1 = \cdots = \beta_5 = 0$): F-test (OLS) / likelihood-ratio test (GLM) against the model with the spline basis removed. *Rejects flatness, not linearity.*

2. **Nonlinearity** ($H_0^{\mathrm{nl}}$: the spline model reduces to a linear-in-$T$ model). F-test (OLS) / LR test (GLM) between the spline model and $\tilde{Y} = \alpha + \delta T + \boldsymbol{\gamma}^\top \mathbf{X} + \varepsilon$. We also report $\Delta\mathrm{AIC} = \mathrm{AIC}(\mathrm{linear}) - \mathrm{AIC}(\mathrm{spline})$; positive $\Rightarrow$ spline preferred.

Within-triple row correlation makes asymptotic row-level p-values overconfident; we cap reported p-values at $p < 0.001$ and rely on $\Delta\mathrm{AIC}$ for quantitative claims.

### B.2.6. PREDICTION AND CROSS-MODEL AGGREGATION

Predictions are evaluated on a shared 60-point grid spanning the pooled 1st–99th-percentile of $T$. For each model $m \in \{1, \ldots, M\}$ we form a prediction sample $\mathcal{S}_m$ (uniform subsample of up to 2,000 rows from $\{i : m_i = m\}$, fixed seed `20260424`), compute

$$\hat{y}_m(t) \;=\; \frac{1}{|\mathcal{S}_m|} \sum_{i \in \mathcal{S}_m} \hat{Y}_i(t), \tag{7}$$

and cross-model average $\hat{y}(t) = M^{-1} \sum_m \hat{y}_m(t)$. Sharing the grid across models guarantees the cross-model average is well-defined at every $t$.

### B.2.7. SATURATION DIAGNOSTICS FOR THE LENGTH-RESPONSE CURVE

Define

$$\kappa = \min\Big\{t \,:\, |\hat{y}'(t)| < 0.10 \cdot \max_t |\hat{y}'(t)|\Big\}, \tag{8}$$

$$\xi = \min\Big\{t \,:\, \hat{y}(t) - \hat{y}(0) \,\leq\, 0.95 \left(\hat{y}(t_{\max}) - \hat{y}(0)\right)\Big\} \tag{9}$$

(inequalities flipped if $\hat{y}$ is increasing in $t$). $\kappa$ is the slope-knee — the position at which the spline's slope drops below 10% of its peak; $\xi$ is the position at which 95% of the long-context drop has accrued.

95% confidence intervals for $\kappa$ and $\xi$ come from a 1,000-iteration model-resampling bootstrap: in each iteration we draw $M$ model identities uniformly with replacement, re-aggregate the per-model curves, and re-measure $\kappa, \xi$ on the bootstrap-aggregated curve. We report 2.5%–97.5% bootstrap quantiles. The random seed is fixed.

### B.2.8. CAUSAL ESTIMAND: SCOPE AND LIMITATIONS

The intervention contrasts estimate the *average within-triple counterfactual effect* of our prefix-flipping protocol on $P(Y{=}1)$, conditional on the listed confounders — **not** the universal causal effect of "correctness" in the wild. Our flipping procedure samples uniformly across positions and atom values, whereas natural errors are heterogeneous in type and location; the two distributions may produce systematically different downstream effects.

*Table 4.* Paired-difference spline regression results for the two intervention treatments, with and without mutual adjustment. The slope is the secant slope per unit $\Delta$ over the full data range. $\overline{\Delta P}_{\pm}$ are the mean $\Delta P$ values over the positive ($T > 0$, correction) and negative ($T < 0$, corruption) regions. $\Delta\text{AIC} = \text{AIC(linear)} - \text{AIC(spline)}$.

| Contrast | slope/unit | $\overline{\Delta P}_+$ | $\overline{\Delta P}_-$ | $\lvert\overline{\Delta P}_-/\overline{\Delta P}_+\rvert$ | Sig. $p$ | Nonlin. $p$ | $\Delta\text{AIC}$ |
|---|---|---|---|---|---|---|---|
| Global $\Delta$, no local control | $+0.42$ | $+0.117$ | $-0.231$ | $2.0\times$ | $<0.001$ | $7\times10^{-71}$ | $+325.8$ |
| Global $\Delta$ + local control | $+0.37$ | $+0.083$ | $-0.214$ | $2.6\times$ | $<0.001$ | $3\times10^{-78}$ | $+359.8$ |
| Local last-10 $\Delta$, no global control | $+0.29$ | $+0.063$ | $-0.238$ | $3.8\times$ | $<0.001$ | $4\times10^{-290}$ | $+1340.1$ |
| Local last-10 $\Delta$ + global control | $+0.19$ | $+0.025$ | $-0.164$ | $6.6\times$ | $<0.001$ | $6\times10^{-157}$ | $+724.4$ |

The length-response result is an *adjusted length-response association*, not a causal length effect; length is not randomized in our protocol.

Row-level F-tests overstate degrees of freedom because rows within a triple are correlated. We cap p-values at $p < 0.001$ in the main text and rely on $\Delta\text{AIC}$ and the bootstrap CIs (§B.2.5, §B.2.7) for quantitative claims.

### B.2.9. RESULTS

**Intervention contrasts.** Table 4 reports the four paired-OLS intervention contrasts on $n_{\text{rows}} = 51{,}260$ rows ($n_{\text{baselines}} = 4{,}910$). After mutual adjustment, the residual global slope ($+0.37$) is approximately $2\times$ the residual local slope ($+0.19$); the global term loses only $11\%$ of its slope when local is added, but the local term loses $35\%$ when global is added. Corruption ($T < 0$) is between $2\times$ and $6\times$ more damaging than correction ($T > 0$) across all four contrasts. All four spline-vs-linear comparisons overwhelmingly favor the spline ($\Delta\text{AIC} \geq 325.8$).

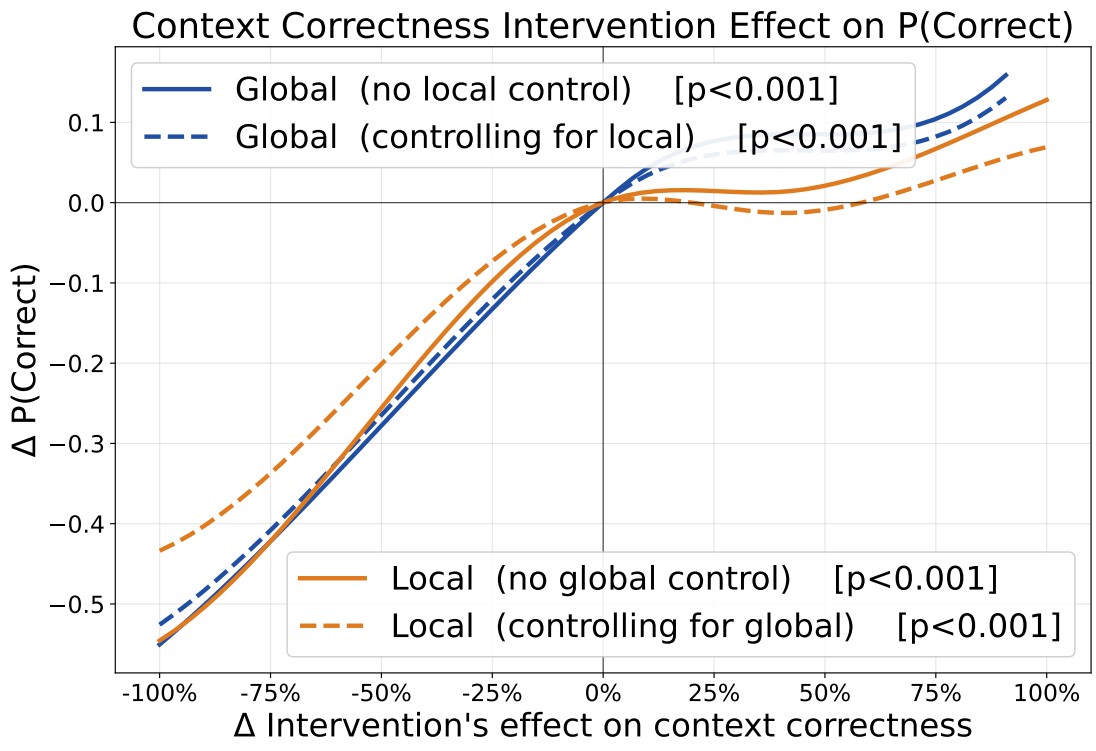

*Figure 25.* Effect of controlled prefix-correctness perturbations on future correctness.

**Length-response curve.** Table 5 reports the saturation diagnostics for the controlled and uncontrolled context-length curves (binomial GLM, significance $p < 0.001$, nonlinearity $\Delta\text{AIC} = +715.9$). The temporal saturation is unchanged by the correctness controls; the asymptotic magnitude inflates by $\approx 0.07$ ($\approx 16\%$) when correctness is dropped from the regressors. Figure 26 overlays both curves.

*Table 5.* Saturation diagnostics for the answer-context length response curve. $\kappa$ is the slope-knee (where the slope drops below 10% of peak); $\xi$ is the 95%-of-asymptote position. 95% CIs from 1,000 model-resampling bootstrap iterations.

| Variant | $\kappa$ (atoms) [95% CI] | $\xi$ (atoms) [95% CI] | $\hat{y}(t_{\max}) - \hat{y}(0)$ |
|---|---|---|---|
| Controlled | 20.8 [20.8, 23.3] | 66.6 [53.8, 71.6] | $-0.430$ |
| Uncontrolled | 20.8 [18.3, 23.3] | 61.5 [46.2, 69.1] | $-0.500$ |

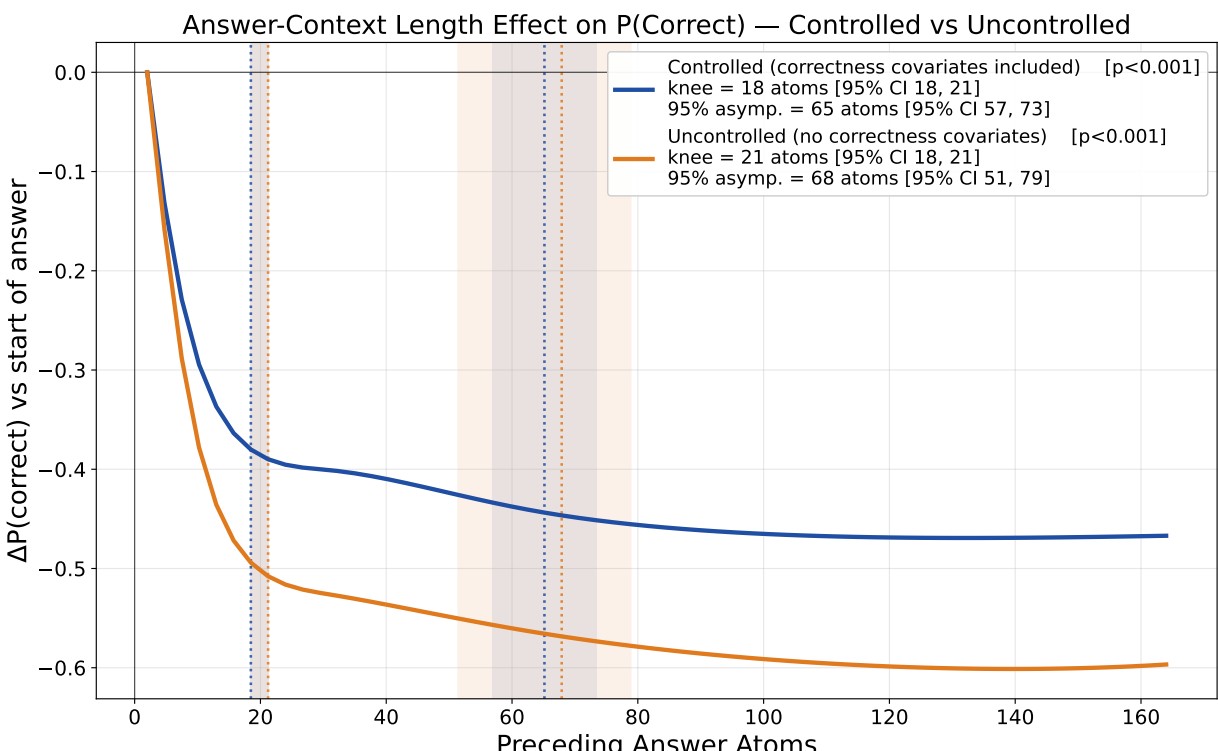

*Figure 26.* Length-response curves, controlled vs. uncontrolled for prefix correctness. Both saturate on the same timeline; the uncontrolled curve drops to a deeper asymptote, attributable to prefix-correctness confounding. Legend reports the joint-significance p-value, the slope-knee with 95% CI, and the 95%-asymptote position with 95% CI for each curve.

**Fixed-correctness curves.** Holding prefix correctness fixed at one of $\{0\%, 20\%, \ldots, 100\%\}$ via override of all correctness covariates produces six within-level length-response curves (Figure 27). The model's predicted $\Pr(Y{=}1)$ at the start of the answer fans out from $\hat{P}(0\%) = 0.25$ to $\hat{P}(100\%) = 0.92$ and converges into the narrow range $[0.02, 0.23]$ at the longest supported context. The absolute headroom lost to length is therefore much larger for clean prefixes ($\approx 0.69$ at 100%) than for dirty ones ($\approx 0.23$ at 0%).

### B.2.10. ROBUSTNESS

**Per-model fits.** Replacing the pooled spline fit with separate per-model spline fits and averaging the per-model predictions leaves the local-$\Delta$ findings essentially unchanged but increases tail variance on the global-$\Delta$ curve, consistent with per-model treatment ranges being narrower than the pooled range. The pooled fit is the reference; the per-model fit is a sensitivity analysis.

## C. Benchmarked Models

Table 6 provides a comprehensive overview of the models evaluated in this study. The selection spans a wide range of scales, from compact 0.5B parameter models to massive 358B architectures, with exclusive closed-source models potentially exceeding these limits. Our ensemble consists of 55 models in total: 15 specialized reasoning models and 40 instruction-tuned models, comprising 46 open-source and 9 closed-source variants.

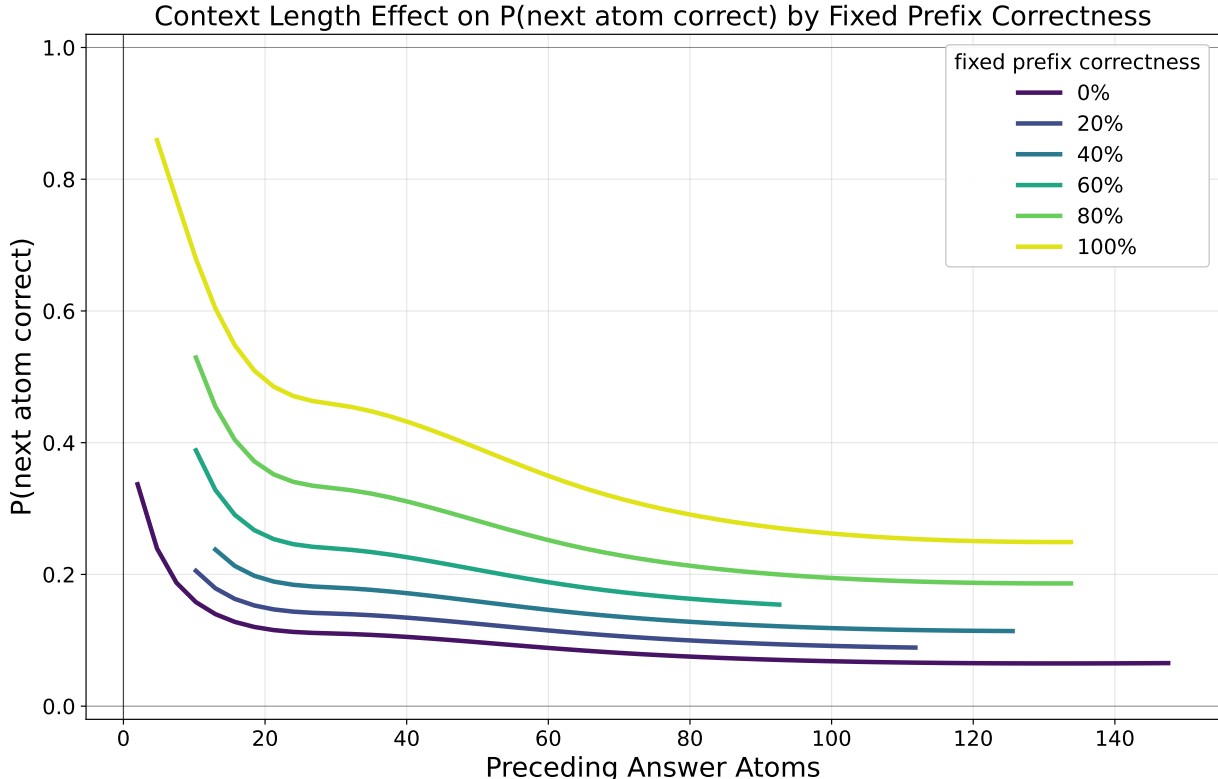

*Figure 27.* Predicted $P$(next correct) as a function of preceding answer atoms with prefix correctness held fixed at different levels.

*Table 6.* Detailed list of the benchmarked models. "# Params" and "# Active" represent the total and active parameter count. "Type" reflects whether the model is an instruction-tuned model or a reasoning model. "Source" reflects whether the model is open or closed source.

| Model | Organization | # Params | # Active | Type | Source |
|---|---|---|---|---|---|
| Nemotron-Nano-9B-v2 | NVIDIA | 9B | 9B | Instruct | Open |
| Nemotron-Nano-12B-v2 | NVIDIA | 12B | 12B | Instruct | Open |
| Nemotron-3-Nano-30B-A3B-BF16 | NVIDIA | 30B | 3B | Instruct | Open |
| K2-V2-Instruct | LLM360 | 65B | 65B | Reasoning | Open |
| MiniMax-M2.1 | MiniMax | 230B | 10B | Reasoning | Open |
| MiniMax-M2.5 | MiniMax | 230B | 10B | Reasoning | Open |
| GLM-4.7-Flash | Z.ai | 31B | 31B | Reasoning | Open |
| GLM-4.7-FP8 | Z.ai | 358B | 358B | Reasoning | Open |
| Gemma-3-1b-it | Google | 1B | 1B | Instruct | Open |
| Gemma-3-4b-it | Google | 4B | 4B | Instruct | Open |
| Gemma-3-12b-it | Google | 12B | 12B | Instruct | Open |
| Gemma-3-27b-it | Google | 27B | 27B | Instruct | Open |
| Gemma-4-26B-A4B-it | Google | 27B | 4B | Instruct | Open |
| Gemma-4-31B-it | Google | 33B | 33B | Instruct | Open |
| Llama-3.1-8B-Instruct | Meta | 8B | 8B | Instruct | Open |
| Llama-3-8B-Instruct | Meta | 8B | 8B | Instruct | Open |
| Llama-3-70B-Instruct | Meta | 70B | 70B | Instruct | Open |
| Llama-3.1-70B-Instruct | Meta | 70B | 70B | Instruct | Open |
| Llama-3.2-3B-Instruct | Meta | 3B | 3B | Instruct | Open |
| Llama-3.3-70B-Instruct | Meta | 70B | 70B | Instruct | Open |

**Table 6 – continued**

| Model | Organization | # Params | # Active | Type | Source |
|---|---|---|---|---|---|
| DeepSeek-R1-Distill-Llama-70B | DeepSeek | 70B | 70B | Reasoning | Open |
| DeepSeek-R1-Distill-Llama-8B | DeepSeek | 8B | 8B | Reasoning | Open |
| Phi-3.5-mini-instruct | Microsoft | 3.8B | 3.8B | Instruct | Open |
| Phi-3.5-MoE-instruct | Microsoft | 42B | 6.6B | Instruct | Open |
| Phi-4-reasoning-plus | Microsoft | 14B | 14B | Reasoning | Open |
| Phi-4-reasoning | Microsoft | 14B | 14B | Reasoning | Open |
| phi-4 | Microsoft | 14B | 14B | Instruct | Open |
| phi-4-mini-reasoning | Microsoft | 3.8B | 3.8B | Reasoning | Open |
| phi-4-mini-instruct | Microsoft | 3.8B | 3.8B | Instruct | Open |
| Mistral-7B-Instruct-v0.3 | Mistral AI | 7B | 7B | Instruct | Open |
| Mixtral-8x7B-Instruct-v0.1 | Mistral AI | 47B | 13B | Instruct | Open |
| Mixtral-8x22B-Instruct-v0.1 | Mistral AI | 141B | 39B | Instruct | Open |
| Mistral-Nemo-Instruct-2407 | Mistral AI | 12B | 12B | Instruct | Open |
| Mistral-Large-Instruct-2407 | Mistral AI | 123B | 123B | Instruct | Open |
| Mistral-Large-Instruct-2411 | Mistral AI | 123B | 123B | Instruct | Open |
| Mistral-Medium-3.5-128B | Mistral AI | 128B | 128B | Instruct | Open |
| Qwen2.5-0.5B-Instruct | Qwen | 0.5B | 0.5B | Instruct | Open |
| Qwen2.5-32B-Instruct | Qwen | 32B | 32B | Instruct | Open |
| Qwen2-0.5B-Instruct | Qwen | 0.5B | 0.5B | Instruct | Open |
| Qwen2-1.5B-Instruct | Qwen | 1.5B | 1.5B | Instruct | Open |
| Qwen2-7B-Instruct | Qwen | 7B | 7B | Instruct | Open |
| Qwen2-72B-Instruct | Qwen | 72B | 72B | Instruct | Open |
| Qwen3-4B-Thinking-2507 | Qwen | 4B | 4B | Reasoning | Open |
| Qwen3-4B-Instruct-2507 | Qwen | 4B | 4B | Instruct | Open |
| Qwen3-30B-A3B-Thinking-2507 | Qwen | 30B | 3B | Reasoning | Open |
| Qwen3-30B-A3B-Instruct-2507 | Qwen | 30B | 3B | Instruct | Open |
| Qwen3-Next-80B-A3B-Thinking | Qwen | 80B | 3B | Reasoning | Open |
| Qwen3-Next-80B-A3B-Instruct | Qwen | 80B | 3B | Instruct | Open |
| Qwen3.6-35B-A3B | Qwen | 35B | 3B | Reasoning | Open |
| Qwen3.6-35B-A3B-NoThink | Qwen | 35B | 3B | Instruct | Open |
| Qwen3.6-27B | Qwen | 27B | 27B | Reasoning | Open |
| Qwen3.6-27B-NoThink | Qwen | 27B | 27B | Instruct | Open |
| phi-3-medium-128k-instruct | Microsoft | 14B | 14B | Instruct | Open |
| gemini-3-flash-preview | Google | Unknown | Unknown | Reasoning | Closed |
| gemini-3-pro-preview | Google | Unknown | Unknown | Reasoning | Closed |
| gpt-oss-20b | OpenAI | 20B | 20B | Reasoning | Open |
| gpt-oss-120b | OpenAI | 120B | 120B | Reasoning | Open |
| gpt-4o | OpenAI | Unknown | Unknown | Instruct | Closed |
| gpt-4o-mini | OpenAI | Unknown | Unknown | Instruct | Closed |
| gpt-4.1-2025-04-14 | OpenAI | Unknown | Unknown | Instruct | Closed |
| gpt-4.1-mini-2025-04-14 | OpenAI | Unknown | Unknown | Instruct | Closed |
| gpt-4.1-nano-2025-04-14 | OpenAI | Unknown | Unknown | Instruct | Closed |
| gpt-5.1-2025-11-13 | OpenAI | Unknown | Unknown | Reasoning | Closed |
| gpt-5-nano-2025-08-07 | OpenAI | Unknown | Unknown | Reasoning | Closed |

# D. Background and Prior Work

In this work, we focus on *information-based* and *verbalization* methods as our primary uncertainty estimators. *Information-based* methods aggregate token-level log-probabilities to derive unit-level scores, such as perplexity or entropy (Min et al., 2023; Geng et al., 2024; Huang et al., 2025b), and constitute the main family we study empirically. *Verbalization* methods prompt the model to explicitly assign confidence in its answer (e.g., "Verb. 1S" or "Verb. 2S") (Kadavath et al., 2022; Tian

et al., 2023), which we additionally evaluate as a complementary signal. We also note additional prominent methods that we do not directly benchmark in this work. *Internal-state* methods analyze the model's hidden-layer representations (e.g., activations or selected neurons), typically via probing or representation-space analysis (Azaria & Mitchell, 2023; Chen et al., 2024). *Sampling* methods assess uncertainty via the consistency across multiple generated outputs, such as measuring the agreement rate among several independent responses (Manakul et al., 2023; Farquhar et al., 2024).

## E. More on Measuring ECE

As exhibited in Equation 2.3, $m$, the number of bins, is a user-defined hyperparameter. We follow Guo et al. (2017) in setting the number of bins to $m = 15$. We calculate the ECE across all evaluation units in generations from the same model, task, and prompt method.

### E.1. Confidence Functions Calibration

As discussed in Section 2.2, many confidence functions (e.g., perplexity, entropy) produce scores outside $[0, 1]$, which is required for calibration assessment. We therefore map their outputs to $[0, 1]$ using one of three calibration methods, each applied independently per (model, task, prompt method) tuple. All schemes follow the same protocol: scores are split into train and test sets, statistics are estimated on the train split, and the learned mapping is applied to the test split.

**Z-Sigmoid** (Algorithm 1) standardizes test scores via Z-score normalization using the training mean and standard deviation, then applies the Sigmoid function to map them to $[0, 1]$. **MinMax** (Algorithm 2) linearly rescales test scores to $[0, 1]$ using the training range, with the bounds computed as trimmed quantiles for robustness to outliers. We pick $q = 0.01$. **Binned** (Algorithm 3) partitions the training score range into $B$ equal-width bins and replaces each score with the empirical correctness rate of its bin, estimated on the training set. We evaluate this scheme with $B \in \{5, 10, 100\}$. As discussed in Section 5.2, while binned calibration minimize measured ECE (Figure 8), it has the opposite effect on the AUROC (Figure 28), revealing that a chosen calibration methods have varying, and even reversed effects, on the different aspects of uncertainty estimation.

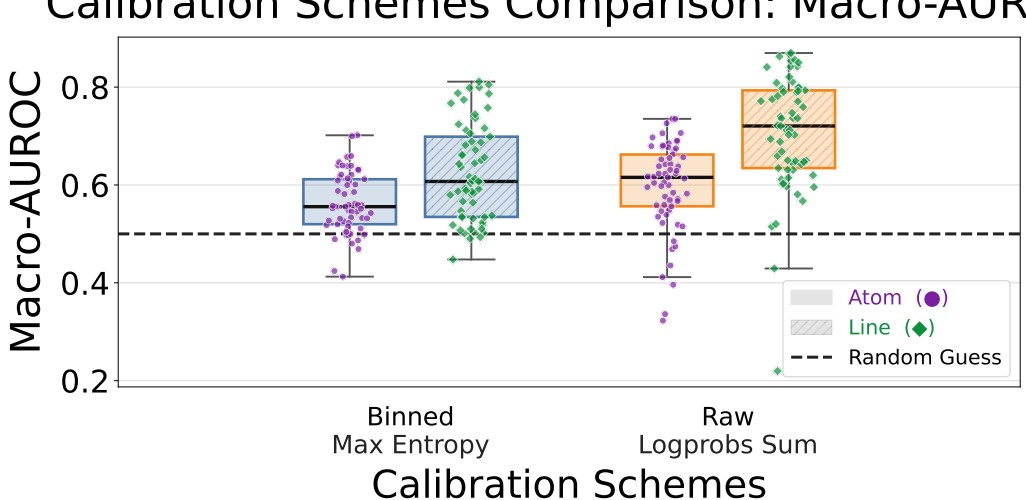

*Figure 28.* Effect of post-hoc calibration's schemes on ranking.

We observed that the best-performing confidence functions on the ECE metrics consistently were among those that require normalization. One can argue that the calibration step unfolds the probability distribution in a way that ECE benefits from it. To test that possibility, we experimented with calibrating all confidence functions' scores, regardless of whether needed, and we observed that the best-performing confidence functions were still consistent, i.e., perplexity and entropy for calibration. With that in mind, we do note that even confidence functions which do not require calibration, still benefit from it in the ECE scores.

---

**Algorithm 1** Z-Sigmoid Scores Calibration

---

**Input:** confidence scores $\mathbf{x}$
**Output:** calibrated confidence scores $\hat{\mathbf{p}}$
$\mathbf{x}_{\text{train}}, \mathbf{x}_{\text{test}} \leftarrow \text{TRAIN\_TEST\_SPLIT}(\mathbf{x})$
$\mu \leftarrow \text{MEAN}(\mathbf{x}_{\text{train}})$
$\sigma \leftarrow \text{STD}(\mathbf{x}_{\text{train}})$
$\tilde{\mathbf{x}}_{\text{test}} \leftarrow (\mathbf{x}_{\text{test}} - \mu)/\sigma$ {Z-score normalization using training statistics}
$\hat{\mathbf{p}} \leftarrow \text{SIGMOID}(\tilde{\mathbf{x}}_{\text{test}})$ {Map normalized scores to $[0, 1]$}

---

**Algorithm 2** MinMax Confidence Calibration

---

**Input:** confidence scores $\mathbf{x}$, trim quantile $q$
**Output:** calibrated confidence scores $\hat{\mathbf{p}}$
$\mathbf{x}_{\text{train}}, \mathbf{x}_{\text{test}} \leftarrow \text{TRAIN\_TEST\_SPLIT}(\mathbf{x})$
$x_{\min} \leftarrow Q_q(\mathbf{x}_{\text{train}}), \quad x_{\max} \leftarrow Q_{1-q}(\mathbf{x}_{\text{train}})$ {Robust range: $q$-th and $(1-q)$-th quantiles of the training scores}
$\hat{\mathbf{p}} \leftarrow \text{CLIP}\left( \dfrac{\mathbf{x}_{\text{test}} - x_{\min}}{x_{\max} - x_{\min}},\ 0,\ 1 \right)$

---

### E.2. ECE as Generation Unfolds

A key advantage of SALT is that long-form generations are decomposed into sequential, task-aligned atoms whose correctness is known a priori. This enables calibration to be tracked not only at the level of the full response, but also as a function of position within the generated answer. We therefore measure ECE over relative atom positions, allowing us to inspect how calibration evolves as the model conditions on an increasingly long self-generated context.

Figure 29 compares this temporal calibration behavior across post-hoc calibration methods, each paired with its best-performing confidence function. The raw Mean Probability baseline becomes steadily less calibrated as generation progresses, with ECE increasing from roughly $0.2$ near the beginning to above $0.5$ near the end. In contrast, binned calibration methods exhibit the opposite trend: after an initially high-error prefix, their ECE rapidly drops and remains substantially lower throughout most of the generation, showing similar trajectories. MinMax and Z-Sigmoid calibration remain comparatively stable but at higher ECE levels, suggesting that their aggregate performance reflects persistent calibration mismatch rather than position-specific degradation. Overall, these trends reinforce the observation that the choice of calibration method strongly shapes calibration quality, and further show that this effect persists throughout the generation rather than only in aggregate.

## F. The Confidence Functions

In this section, we provide the explicit mathematical definitions for the unit-level confidence functions evaluated in this study. For a generated evaluation unit $\hat{u}_i$ consisting of $k$ tokens, let $p_j = p(\hat{y}_j \mid \mathbf{x}, \hat{y}_{1:j-1})$ denote the model's predictive probability for the $j$-th token, and $H(\hat{y}_j) = -\sum_{v \in \Sigma} p(v \mid \mathbf{x}, \hat{y}_{1:j-1}) \log p(v \mid \mathbf{x}, \hat{y}_{1:j-1})$ denote the per-token Shannon entropy (Shannon, 1948) calculated over all tokens $v$ in the vocabulary $\Sigma$. Table 7 summarizes the aggregation methods used to derive logit-based unit-level scores from these token-level signals. The "Trend" column indicates whether a higher

---

**Algorithm 3** Binned Confidence Calibration

---

**Input:** confidence scores $\mathbf{x}$, labels $\mathbf{y}$, number of bins $B$
**Output:** calibrated confidence scores $\hat{\mathbf{p}}$
$\mathbf{x}_{\text{train}}, \mathbf{x}_{\text{test}} \leftarrow \text{TRAIN\_TEST\_SPLIT}(\mathbf{x})$
$x_{\min} \leftarrow \text{MIN}(\mathbf{x}_{\text{train}}), \quad x_{\max} \leftarrow \text{MAX}(\mathbf{x}_{\text{train}})$
$\tilde{x}_i \leftarrow \text{CLIP}\left( \dfrac{x_i - x_{\min}}{x_{\max} - x_{\min}},\ 0,\ 1 \right) \quad$ for each $x_i \in \mathbf{x}_{\text{train}}$
$b_i \leftarrow \min(\lfloor \tilde{x}_i \cdot B \rfloor,\ B-1) \quad$ for each training sample $i$ {Assign each training sample to one of $B$ equal-width bins}
$v_b \leftarrow \text{MEAN}\left( y_i \mid b_i = b \right) \quad$ for each $b \in \{0, \dots, B-1\}$ {Empirical correctness rate in bin $b$}
Apply the same normalization and bin assignment to $\mathbf{x}_{\text{test}}$ to obtain $b_i^{\text{test}}$
$\hat{p}_i \leftarrow v_{b_i^{\text{test}}} \quad$ for each test sample $i$ {Replace score with its bin's empirical accuracy}

---

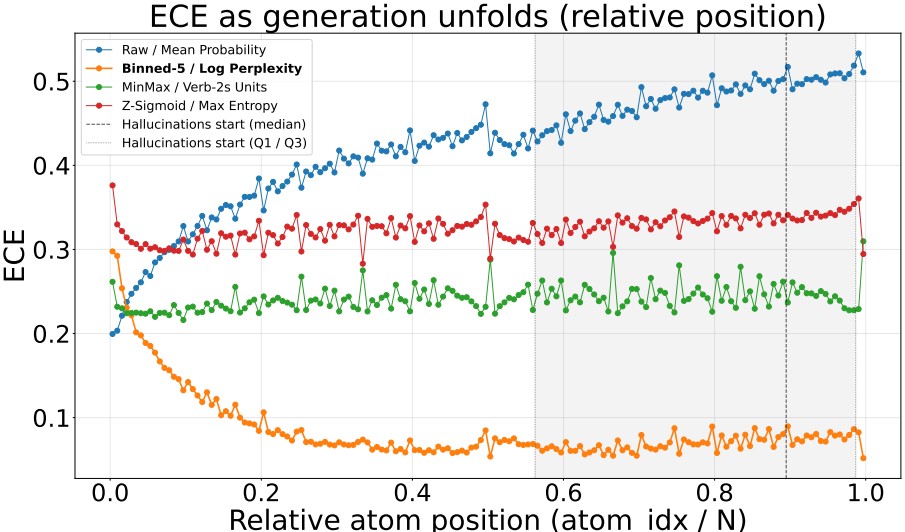

*Figure 29.* ECE (y-axis) as generation unfolds (x-axis). Points represent bins of atoms' relative position. Markers show bin means.

value signifies higher confidence ($\uparrow$) or higher uncertainty ($\downarrow$).

We additionally evaluate *verbalized* confidence functions, which elicit a confidence value by querying the model itself rather than aggregating token-level statistics. Two probe types are considered: $P(\text{True})$ (Kadavath et al., 2022), which reads the normalized next-token probability on the labels `True` vs. `False` when the model is asked whether a piece of generated content is correct, and *Verbalized 2S* (Tian et al., 2023), which asks the model to state a numeric confidence in their response-ready in the $[0, 1]$ range. We denote these per-content elicitors by

$$\rho_M(z) \;=\; \frac{p_M(\text{T} \mid z)}{p_M(\text{T} \mid z) + p_M(\text{F} \mid z)} \in [0,1], \qquad c_M(z) \in [0,1],$$

where $z$ is the piece of generated content the model is asked to judge. Each probe is queried at the generation level (over the full response $\hat{\mathbf{y}}$) and at the unit level by passing the full unit decomposition $\hat{u}_{1:n}$ jointly in a single prompt and reading the per-unit output, which we write as $\rho_M(\hat{u}_i \mid \hat{u}_{1:n})$ and $c_M(\hat{u}_i \mid \hat{u}_{1:n})$. Generation-level scores are additionally broadcast unchanged to every unit, providing a zero-resolution control at unit granularity. The six resulting functions are summarized in Table 8.

## G. More on Uncertainty Estimation results

### G.1. How to Find the Best Confidence Function

In this section, we detail the tests we performed to decide which confidence score function is on top among the confidence score functions we tested in this work. Without loss of generally, for each pair of confidence score functions, $\kappa_1, \kappa_2$, we pick the sets of paired results. A paired result in this context, is a pair of confidence scores extracted from applying the same LLM on the same dataset in this benchmark, using the same prompt, and evaluating on the same level of granularity, with the only exception that we use either $\kappa_1$ or $\kappa_2$ to get the ultimate confidence scores. We end up with over 500 paired samples for each comparison of two confidence score functions. Since the samples are paired, we employ the one-sided Wilcoxon paired test, where the tested hypothesis is whether $\kappa_1$ is better than $\kappa_2$, where the term better means greater for AUROC metrics, and lower for the ECE metric. We report the p-value of all tests in a matrix, so it is visible which confidence functions are better than which. In the case where multiple confidence functions are statistically better than the rest, and the Wilcoxon paired test is not sufficient, we pick the one with the best median score among the group of the best, statistically. Finally, we supplement the full results for the ECE, Adaptive ECE, AUROC, and PRR in Figures 30, 31, and 32, respectively.

*Table 7.* Summary of Unit-Level Confidence Functions. $k$ denotes the number of tokens within the unit.

| Metric Name | Definition | Trend |
|---|---|---|
| Perplexity | $\exp\left(-\dfrac{1}{k}\sum_{j=1}^{k}\log p_j\right)$ | ↓ |
| Log-Perplexity | $-\dfrac{1}{k}\sum_{j=1}^{k}\log p_j$ | ↓ |
| Mean Probability | $\dfrac{1}{k}\sum_{j=1}^{k}p_j$ | ↑ |
| Median Probability | $\mathrm{median}\left(\{p_j\}_{j=1}^{k}\right)$ | ↑ |
| Max Probability | $\max_{j=1}^{k}p_j$ | ↑ |
| Logprobs Sum | $\sum_{j=1}^{k}\log p_j$ | ↑ |
| Mean Entropy | $\dfrac{1}{k}\sum_{j=1}^{k}H(\hat{y}_j)$ | ↓ |
| Max Entropy | $\max_{j=1}^{k}H(\hat{y}_j)$ | ↓ |

## G.2. Confidence Ranking

As discussed in Section 5.2, while we observe a strong correlation between precision and calibration, no such relationship is evident between precision and confidence ranking metrics. Figure 33 illustrates this lack of correlation and highlights the comparative ineffectiveness of ranking at the atomic-unit level versus the line-unit level, which we discuss further in Appendix J.

## G.3. Uncertainty Estimation Transferability of SALT

In Section 5.2, we analyze whether uncertainty-estimation trends observed on SALT transfer to external benchmarks. Here, we provide the full experimental protocol and correlation analysis. Unlike the precision-based comparison in Section 4, which compares precision rankings, this analysis compares *confidence-function rankings*: we ask whether the confidence functions' performance ranking on SALT correlates well with external benchmarks.

We evaluate this transferability on a subset of models for which results are available on both SALT and the external benchmark, considering *AIME* and *MMLU-Pro*. For each matched model, we evaluate all logits-based confidence functions introduced in Appendix F. We repeat the comparison across uncertainty metrics, calibration methods, and SALT granularity levels. The evaluated metrics are AUROC and ECE, and the evaluated calibration methods include Z-score followed by a sigmoid transform, min-max normalization, and binned calibrations.

We compute transferability using two complementary procedures.

**Per-model rank correlation.** First, for each matched model, we compute the Spearman correlation between the ranking of confidence functions on SALT and the corresponding ranking on the external benchmark:

$$\rho_m(D,r,c,g) = \rho_{\mathrm{Spear}}\left(\{s_{m,\kappa,r,c,g}^{\mathrm{SALT}}\}_{\kappa\in\mathcal{K}}, \{s_{m,\kappa,r,c}^{D}\}_{\kappa\in\mathcal{K}}\right).$$

We then average these correlations across matched models:

$$\bar{\rho}(D,r,c,g) = \frac{1}{|\mathcal{M}|}\sum_{m\in\mathcal{M}}\rho_m(D,r,c,g).$$

*Table 8.* Summary of Verbalized Confidence Functions.

| Confidence Function | Definition | Trend | Granularity |
|---|---|---|---|
| $P(\text{True})$, Response | $\rho_M(\hat{\mathbf{y}})$ | ↑ | Generation Only |
| $P(\text{True})$, per-Unit | $\rho_M(\hat{u}_i \mid \hat{u}_{1:n})$ | ↑ | Any Unit |
| $P(\text{True})$, Broadcast | $\rho_M(\hat{\mathbf{y}})$, replicated to each $\hat{u}_i$ | ↑ | Any Unit |
| Verbalized 2S, Response | $c_M(\hat{\mathbf{y}})$ | ↑ | Generation Only |
| Verbalized 2S, per-Unit | $c_M(\hat{u}_i \mid \hat{u}_{1:n})$ | ↑ | Any Unit |
| Verbalized 2S, Broadcast | $c_M(\hat{\mathbf{y}})$, replicated to each $\hat{u}_i$ | ↑ | Any Unit |

This procedure preserves the rank comparison within each model and then aggregates across models. We therefore view it as the more rank-faithful estimate of transferability. The results are shown in Figure 34.

**Average-score rank correlation.** Second, we average each confidence function's score across matched models and then compute a single Spearman correlation:

$$\bar{s}^D_{\kappa,r,c} = \frac{1}{|\mathcal{M}|} \sum_{m \in \mathcal{M}} s^D_{m,\kappa,r,c},$$

and analogously for SALT. We then compute

$$\rho_{\text{avg}}(D, r, c, g) = \rho_{\text{Spear}} \left( \{\bar{s}^{\text{SALT}}_{\kappa,r,c,g}\}_{\kappa \in \mathcal{K}}, \{\bar{s}^D_{\kappa,r,c}\}_{\kappa \in \mathcal{K}} \right).$$

This procedure first smooths model-specific variation and then compares the global ordering of confidence functions. Empirically, it tends to produce stronger-magnitude correlations than the per-model procedure, since averaging reduces idiosyncratic model-level noise before ranking. The results are shown in Figure 35.

Across both procedures, we find that AIME agrees substantially better with SALT than MMLU-Pro.

### G.4. PRR and AUROC Correlation

We evaluate the Spearman correlation between AUROC and PRR on both the SALT benchmark and external datasets. Chen et al. (2026) reports high Spearman correlations of 0.9, 0.96, and 0.98 for the GSM8K, SciQ, and TriviaQA datasets, respectively. Similarly, Yaldiz et al. (2025) observes an average Spearman correlation of 0.96 across NaturalQA, TriviaQA, GSM8K, and WebQA. On SALT, we consistently find nearly perfect Spearman correlations between AUROC and PRR at the macro-level (see creffig: prr and auroc correlations scatter plot), and a high correlation at the micro-level (see Figure 37).

### G.5. Calibration Driven by Precision: A Causal Investigation

This section details the methodology and results for the causal investigation of the relationship between model precision and calibration (measured via ECE) reported in Section 5.2. We restricted this analysis to the optimal confidence function identified in Section 5.2, perplexity-based estimation.

G.5.1. METHODOLOGY

To rigorously assess the causal link Precision → ECE on observational data ($N = 50$ LLMs), we employed a three-stage framework:

**1. Regime Stratification:** Preliminary non-parametric analysis revealed a non-monotonic "V-shaped" relationship between precision and calibration. To avoid confounding effects arising from this non-linearity (where opposite trends cancel each

*Table 9.* Causal analysis of the Precision → ECE relationship across two regimes. **Confounding Tests** assess if the link persists after controlling for model size ($p < 0.05$ indicates a significant link). **Directionality Tests** (ANM) assess if the causal model fits the data ($p > 0.05$ indicates the model is accepted; residuals are independent noise).

| Test Category | Metric | Learning Regime (Precision < 0.6) | Overconfidence Regime (Precision ≥ 0.6) |
|---|---|:---:|:---:|
| *Sample Size* | $N$ | 34 | 16 |
| **1. Robustness to Confounding (Model Size)** | | | |
| Partial Correlation | Spearman $\rho$ | $-0.90$ ($p < 0.001$) | $+0.84$ ($p < 0.001$) |
| Conditional Independence | HSIC Test | Reject $H_0$ ($p < 0.001$) | Reject $H_0$ ($p < 0.001$) |
| Causal Effect (DML) | Estimate | $-0.60$ ($p < 0.001$) | $+0.62$ ($p < 0.001$) |
| **2. Causal Directionality (ANM)** | | | |
| **Forward Model** | HSIC p-value | **0.99** | 0.07 |
| (Precision → ECE) | Decision | **Accepted** | Accepted (Ambiguous) |
| **Backward Model** | HSIC p-value | **0.03** | 0.08 |
| (ECE → Precision) | Decision | **Rejected** | Accepted (Ambiguous) |

other out), we stratified the analysis into two distinct regimes: a *Low Precision* regime (Precision $< 0.6$, $N = 34$) and a *High Precision* regime (Precision $\geq 0.6$, $N = 16$).

**2. Confounding Assessment (Robustness to Model Scale):** Within each regime, we tested whether the Precision-ECE link was spurious and driven simply by model capacity. We controlled for *Total Parameters* and *Active Parameters* using three complementary tests:

- **Partial Correlation (Spearman):** Measures monotonic association while holding model size constant.

- **Conditional Independence (HSIC):** A non-parametric kernel-based test ($H_0 : X \perp Y \mid Z$) to detect non-linear dependencies that linear controls might miss (Gretton et al., 2005).

- **Non-Linear Double Machine Learning (DML):** We estimated the causal effect using an EconML LinearDML estimator (Chernozhukov et al., 2016), employing Random Forests to model the nuisance parameters.

**3. Causal Directionality (ANM):** To validate the direction of causality, we utilized the Additive Noise Model (ANM) framework (Hoyer et al., 2008). We modeled the functional relationship using Gaussian Process regression and tested independence between the predictor and the residuals using HSIC. A high p-value indicates that the residuals are independent noise (supporting the model), while a low p-value indicates structure in the residuals (rejecting the model).

G.5.2. RESULTS AND CONCLUSION

Table 9 summarizes the results for both regimes.

The analysis provides strong observational evidence for a non-monotonic causal mechanism:

1. **Robustness:** In both regimes, the relationship between Precision and ECE is robust to confounding. All three tests (Partial Correlation, HSIC, and DML) confirm that the link persists even after rigorously controlling for model scale.

2. **The Low Precision Regime** ($< 0.6$)**:** We find a clear causal signal that improvements in precision drive better calibration. The ANM test decisively accepts the forward direction ($p = 0.99$) and rejects the reverse ($p = 0.03$), suggesting precision is the functional driver of calibration error in this phase.

3. **The High Precision Regime** ($\geq 0.6$)**:** The relationship inverts significantly. Higher precision causes *higher* calibration error, with a positive causal effect of $+0.62$. While the directionality tests are ambiguous, we hypothesize it is likely due to the small sample size, $N = 16$.

G.5.3. RESIDUAL ANALYSIS

We fit a linear regression ECE $\sim$ Precision on the 35 non-reasoning models in our dataset. This establishes a baseline relationship between precision and expected ECE. We then examined how the 9 reasoning models deviate from this baseline.

**Procedure:**

1. Fit: $\widehat{\text{ECE}} = \beta_0 + \beta_1 \cdot \text{Precision}$ on non-reasoning models

2. For each reasoning model $i$, compute residual: $r_i = \text{ECE}_i - \widehat{\text{ECE}}_i$

3. Test whether $\{r_i\}$ are systematically different from zero using one-sided t-tests

**Results:**

- Mean residual: $+0.039$ (slight positive deviation)

- One-sided t-test (worse than expected): $p = 0.052$

- One-sided t-test (better than expected): $p = 0.948$

The residuals are not significantly different from zero in either direction, indicating reasoning models' calibration aligns with the precision-based prediction.

G.5.4. CAUSAL MEDIATION ANALYSIS

We employed the DoWhy framework (Sharma & Kiciman, 2020; Blöbaum et al., 2024), which builds upon Pearl's do-calculus (PEARL, 1995) to decompose the effect of reasoning architecture on calibration.

**Causal Graph:**  We specified the following directed acyclic graph:

$$\text{Is\_Reasoning} \rightarrow \text{Precision} \rightarrow \text{ECE}$$
$$\text{Is\_Reasoning} \rightarrow \text{ECE}$$

The framework decomposes the Total Effect into:

- **Natural Indirect Effect (NIE)**: Effect mediated through Precision

- **Natural Direct Effect (NDE)**: Direct effect independent of Precision

**Estimation:**  We used two-stage linear regression with 1000 bootstrap iterations to estimate confidence intervals:

1. **First stage**: Regress Precision on Is_Reasoning

2. **Second stage**: Regress ECE on both Is_Reasoning and Precision

| Effect | Estimate | 95% CI | Significance |
|---|---|---|---|
| Total Effect | $-0.082$ | — | Overall improvement |
| NIE (via Precision) | $-0.0995$ | $[-0.151, -0.058]$ | Significant ($p < 0.001$) |
| NDE (Direct) | $+0.005$ | $[-0.026, 0.049]$ | Not significant ($p = 0.798$) |

*Table 10.* Mediation analysis results. The NIE accounts for 115% of the Total Effect, indicating complete mediation. The NDE confidence interval spans zero.

**Results:**

**ANCOVA Confirmation:**  We confirmed these findings via ANCOVA. Regressing ECE on both Precision and Is_Reasoning yielded:

- Precision coefficient: $\beta = -0.247$, $p < 0.001$ (highly significant)

- Is_Reasoning coefficient: $\beta = 0.005$, $p = 0.798$ (not significant)

This corroborates the NDE estimate, showing no residual effect of reasoning architecture when controlling for precision.

**Interpretation:** The NIE being larger in magnitude than the Total Effect (with opposite-signed NDE) indicates *complete mediation* with slight inconsistency. This means: (1) reasoning models' calibration benefit flows entirely through their higher precision (NIE significant, negative), and (2) the architecture itself contributes no additional calibration advantage (NDE not significant, CI spans zero). The slight positive NDE point estimate suggests, if anything, a minor penalty when precision is held constant, though this is within sampling error.

### G.6. The combined Effect of Reasoning + CoT

To determine the combined effect of intrinsic reasoning (Reasoning models) and extrinsic prompting (Chain-of-Thought), we evaluated a hybrid configuration: **Reasoning Models prompted with CoT**. We compared this combined setup against the standard baseline (Instruct models with regular prompting).

The results, visualized in Figure 38 (Purple series), reveal a compounding effect that further validates the "Reasoning-Ranking Trade-off":

- **Precision Saturation:** The combination of Reasoning models and CoT yields a median precision improvement of ∼78%, which is statistically indistinguishable from using Reasoning models alone (Blue series). This suggests that the intrinsic reasoning capabilities of the model already capture the majority of the achievable accuracy gains, making either the extra CoT tokens redundant for correctness, or reasoning models are simply generating CoT tokens regardless of whether requested.

- **Calibration Additivity:** Interestingly, despite the saturation in precision, the calibration gains continue to stack. The hybrid configuration achieves the highest reduction in Expected Calibration Error (ECE) (∼21%), outperforming Reasoning models alone (∼16%). This results in the most calibrated configuration among all tested setups, suggesting that the additional reasoning traces help align the model's confidence with its (high) accuracy, even if they do not further improve the accuracy itself.

- **Ranking Aggravation:** Crucially, while accuracy saturates, the degradation in uncertainty estimation continues to worsen. The hybrid configuration suffers the most severe drop in ranking ability (AUROC), with a median degradation of ∼11.6%, compared to ∼5% for Reasoning models alone and ∼2% for CoT alone.

**Conclusion:** This creates a paradox where the "Reasoning + CoT" setup is simultaneously the **most calibrated** and the **least discriminative** (lowest AUROC). It suggests that while extended reasoning traces help the model hit a precise aggregate confidence level (low ECE), the added noise or internal coherence flattens the relative ordering of correct vs. incorrect atoms, making it harder to rank individual generations reliably.

While Figures 6, 38 demonstrate the general reasoning trade-off, they lack the exploration of the individual tasks and models. For completeness, we provide Figures 39 and 40 to demonstrate the ECE-AUROC and the Precision-AUROC trade-offs induced by reasoning.

## H. Unit Alignment via Dynamic Programming

In this section, we explore an alternative evaluation setting that utilizes the Needleman–Wunsch algorithm (Needleman & Wunsch, 1970) to align generated units $\hat{\mathbf{u}}$ with ground-truth units $\mathbf{u}$. As discussed in Section 4.3, long-form generations are susceptible to "index shifts" where a single omitted or hallucinated unit causes all subsequent units to be labeled as incorrect under strict indexing, even if they are factually accurate.

The Needleman–Wunsch algorithm, a global alignment technique based on dynamic programming, addresses this by maximizing a similarity score between two sequences. In our implementation, we assign a score of +1 for a match (identical strings) and 0 for mismatches or gaps. This formulation identifies the longest common subsequence while preserving relative ordering, effectively allowing for localized "skips" in the generation.

Figure 41 illustrates the empirical impact of this alignment across all tasks. We observe that dynamic alignment consistently yields higher precision scores compared to strict indexing, with a particularly pronounced impact on tasks like First-Order Logic. This confirms that index shifts do occur and that for certain research applications—specifically those focused on factual retrieval where structural ordering is secondary—the units-alignment setting may be a preferred configuration for SALT.

Despite these improvements in measured performance, we chose to focus our primary analysis on the non-alignment (strict indexing) setting. Our reasoning is rooted in the fundamental definitions of our metrics in Section 2.3. The alignment algorithm is designed to maximize the overlap between sequences, which inherently maximizes recall. However, in doing so, it can obscure the presence of redundant units or structural failures. For example, consider a ground truth units sequence of $[1, 1, 1, 1, 3]$ and a generated units output of $[1, 3]$. Under strict indexing, the units are compared as pairs $(1, 1)$ and $(1, 3)$, yielding a recall of $20\%$ and precision of $50\%$. By aligning the sequences to match the final digit, recall increases to $40\%$, but precision drops to $40\%$, demonstrating how alignment can artificially shift precision as a byproduct of maximizing recall. As precision is our primary measure of accuracy—specifically because it captures the model's tendency to hallucinate extraneous information—we believe the non-alignment setting provides a more realistic and rigorous evaluation. In real-world long-form generation, a model is expected to maintain logical and structural consistency on its own; a correctly calculated value placed in the wrong row of a matrix or a logical conclusion appearing out of sequence is often as problematic as a factual error. Therefore, we prioritize strict indexing to ensure that models are penalized for failing to maintain the deterministic structure required by the task.

# I. Generalized Analysis of AUROC Sensitivity to Label Noise

A critical advantage of our procedural benchmark is the guarantee of deterministic ground truth ($e = 0$). In contrast, non-deterministic benchmarks introduce label noise. Here, we analyze the impact of $e$ label errors on the AUROC metric without assuming a perfect initial ranking. We demonstrate that while the *average* sensitivity scales with model quality, the *worst-case* vulnerability remains critically high regardless of the model's initial performance.

## I.1. Problem Setup

Let $D = \{(x_i, y_i)\}_{i=1}^N$ be a dataset with $n_1$ Positives and $n_0$ Negatives ($N = n_1 + n_0$). Let $A$ denote the initial true AUROC score of the model. The AUROC is equal to the normalized Mann-Whitney U statistic:

$$A = \frac{1}{n_1 n_0} \sum_{i \in P} \sum_{j \in N} \mathbb{1}(s_i > s_j) \tag{10}$$

We analyze the absolute shift $\Delta$ caused by $e$ errors.

## I.2. Average-Case Bound (Generalized)

In the realistic case of random annotation errors, the expected shift depends heavily on the model's initial quality ($A$).

**Theorem I.1.** *For a balanced dataset, the expected shift in AUROC caused by $e$ random label errors is proportional to the model's distance from random guessing:*

$$\mathbb{E}[\Delta] \approx \frac{e}{N}|2A - 1| \tag{11}$$

*Proof.* Let $A$ be the initial AUROC. By definition, the expected number of Negatives outscored by a random Positive is $n_0 A$. Consider flipping a random sample $x_k$:

1. **Scenario 1 ($x_k \in P \to N$):** We lose the wins $x_k$ contributed. On average, this is $\mathbb{E}[W_k] = n_0 A$. As a new Negative, $x_k$ is compared to remaining Positives. Assuming independence, a random sample is outscored by half the population, so it contributes $\approx n_1/2$ losses.

2. **Scenario 2 ($x_k \in N \to P$):** Symmetric logic applies.

The net change in the numerator for a single flip is the difference between the "wins lost" and "losses gained":

$$\mathbb{E}[\delta_{\text{num}}] \approx \left| \frac{n_1}{2} - n_0 A \right| \tag{12}$$

For balanced data ($n_0 = n_1 = N/2$), this simplifies to $\frac{N}{4}|1 - 2A|$. Dividing by the total pairs ($N^2/4$) yields the shift per error: $\frac{|1-2A|}{N}$. Summing for $e$ errors completes the proof. $\square$

### I.3. Worst-Case Bound (Global)

Unlike the average case, the worst-case bound does *not* diminish for lower-performing models.

**Theorem I.2.** *Regardless of the initial AUROC A, the maximum shift caused by e adversarial errors in a balanced dataset is bounded by:*

$$\Delta_{max} \leq \frac{2e}{N} \tag{13}$$

*Proof.* The worst-case attack targets specific outliers rather than the average distribution. Even a model with mediocre global performance ($A \approx 0.5$) may assign a very high score to a single "lucky" Positive instance $x_{max}$. If $x_{max}$ is ranked above all Negatives, flipping it causes the loss of $n_0$ concordant pairs. The shift is:

$$\Delta \approx \frac{n_0}{n_1 n_0} = \frac{1}{n_1} = \frac{2}{N} \tag{14}$$

Since such an outlier arrangement is compatible with almost any global AUROC score (by balancing it with poor rankings elsewhere), the bound $\frac{2e}{N}$ holds globally for any $A$. □

### I.4. Empirical Simulation

To validate these bounds, we simulated models with True AUROC scores $A \in [0.5, 1.0]$ against random label noise levels ($e/N \in \{0.05, 0.10, 0.20\}$). As shown in Figure 42, the simulation (solid lines) aligns with our average-case derivation (dashed lines, Equation 11), confirming that unbiased noise systematically pulls discriminative performance toward the random baseline of $0.5$.

### I.5. Implications for Benchmarking

These findings demonstrate that label noise is not a constant bias but a *quality-dependent compressor*. For models near the random baseline ($A \approx 0.5$), noise has negligible impact. However, for state-of-the-art models ($A \to 1.0$), sensitivity is maximized. In a benchmark with $10\%$ noise, a perfect model would be measured at $0.90$, potentially erasing the performance margins that distinguish leading architectures. By providing a zero-noise environment, SALT ensures that improvements in long-form uncertainty estimation are accurately captured rather than being suppressed by the statistical limitations of the evaluation set.

## J. More on Granularity

For each input sample, SALT pre-defines a ground-truth *sequence*. While we focus on decompositions to atoms and lines, based on semantic meaning, one could choose any desired decomposition criterion. In fact, SALT can even decompose sequences at the character and token levels.

In Section 5.2, we discuss the reversed trends between ECE and AUROC: the former is stable across granularity evaluation levels (see Figure 43), whereas the latter emerges more clearly at the line level. Here, we provide additional empirical evidence for this discrepancy; further analysis of AUROC itself is given in Appendix G.2.

To investigate the poor ranking performance at fine granularities, we analyze the probability density functions (PDFs) of confidence scores for correct and incorrect atoms. Using the Z-Sigmoid method—which, as a monotonic transformation, faithfully preserves the model's original ranking—Figure 44 reveals a persistent and substantial overlap between these distributions across all models and tasks. This poor separability demonstrates that raw token-level confidence signals are inherently insufficient for high-resolution error identification. These findings highlight a critical need for either stronger auxiliary signals or the development of dedicated confidence functions trained specifically for atomic evaluation units.

Figure 45 provides a task-level view of the gap between atomic- and line-level ranking. The figure suggests that this gap is not uniform across tasks: for some tasks, such as DNA Translation, the difference between atomic- and line-level ranking appears relatively small, whereas for others, such as Multi-Needle, it is substantially larger.

At this stage, we do not claim a complete explanation for this phenomenon. One possible partial explanation is the semantic dependence structure of the task, which we formalize separately in Appendix K. Another distinct factor is positional sensitivity under strict indexing, which we analyze separately in Appendix H. We therefore view the gap between atomic- and line-level ranking as a task-dependent phenomenon that may arise from multiple sources.

## K. Semantic Dependence Across Tasks

Building on the task design described in Section 4.2, this section provides a formal characterization of the semantic dependence within each SALT task. Using the notation of Section 2.1, let $\mathbf{x}$ be the input and let $\mathbf{u} = (u_1, \ldots, u_{U_{gt}})$ be the ground-truth sequence of evaluation units. Let the input be partitioned into task-specific spans $\mathbf{s}(\mathbf{x}) = (s_1, \ldots, s_K)$, where a span may correspond, for example, to a codon in DNA Translation, a row or column fragment in Matrix Multiplication, or a candidate item in Multi-Needle.

In this appendix, we focus only on the *semantic* dependence structure of target evaluation units. Namely, we ask which input spans and which prior units are required in order to determine the correctness of a future unit. Positional effects caused by insertions or deletions under strict indexing are conceptually distinct, and are discussed separately in Appendix H.

**Semantic dependence of a target unit.** Consider a target unit $u_i$. Let

$$S_i = \{k_1, \ldots, k_r\} \subseteq \{1, \ldots, K\}$$

be a set of input-span indices, and let

$$G_i = \{q_1, \ldots, q_m\} \subseteq \{1, \ldots, i-1\}$$

be a set of indices of prior ground-truth output units. We say that $(S_i, G_i)$ is *sufficient* for $u_i$ if there exists a deterministic task-specific function $\phi_i$ such that

$$u_i = \phi_i\big((s_k)_{k \in S_i}, (u_q)_{q \in G_i}\big).$$

That is, the correct value of $u_i$ can be determined from the input spans indexed by $S_i$ together with the previously established target units indexed by $G_i$.

This yields a preliminary notion of semantic dependence. The set $S_i$ captures the *input-span dependence* of $u_i$, namely how much of the input is needed in order to determine it. The set $G_i$ captures the *logical output dependence* of $u_i$, namely whether the correct value of $u_i$ depends on previously established target units.

When $G_i = \emptyset$, the target unit is *output-independent*: in principle, it is determined solely from the input spans in $S_i$. When $G_i \neq \emptyset$, the target unit is *logically output-dependent*: its correct value depends on one or more prior target units.

**Task-level interpretation.** This distinction helps characterize the tasks in SALT.

In **DNA Translation**, each unit depends only on a small local input span, e.g., a specific codon or nucleotide span, and does not depend on prior output units. Thus, for every $u_i$, one may take $G_i = \emptyset$, while $S_i$ is small.

In **Kronecker Product**, each unit is determined by a small local subset of the input: specifically, by the relevant entry of the first matrix, the relevant entry of the second matrix, and the fixed Kronecker-product rule. Thus, the task exhibits input-span dependence, but it is relatively local compared to Matrix Multiplication. The output units are logically independent of one another, and therefore one may take $G_i$ to be empty for all $i$.

In **Matrix Multiplication**, each unit may depend on a broader subset of the input, such as an entire row and column, so $S_i$ can be substantially larger than in DNA. However, the output units are still logically independent of one another: each matrix element is determined directly from the input matrices, and therefore $G_i = \emptyset$ for all $i$.

In **Code**, each unit is determined by the program logic together with the specific input instance on which the program is evaluated. Depending on the program, this may require reasoning over a relatively broad subset of the input code, so the input-span dependence can be moderate or large. However, under the task formulation, the target units do not semantically depend on previously established output units, and therefore one may typically take $G_i$ to be empty.

In **Multi-Needle**, the dependence on the input is broader and more order-sensitive. The target unit $u_i$ corresponds to the $i$-th matching item induced by the input passage, so determining it may require identifying and ordering a larger set of relevant input spans. Thus, compared to DNA, Multi-Needle exhibits stronger input-span dependence. However, this does not necessarily imply logical dependence on prior target units: a model may miss an earlier item and thereby shift later generated items, but this is better viewed as a positional or generation-time effect rather than as semantic dependence of the ground-truth target units themselves.

In **First-Order Logic**, the dependence is stronger in a different sense. If the correct value of $u_i$ can only be derived after first establishing prior units $u_q, \ldots, u_p$, then these units belong to $G_i$, and therefore $u_i$ is logically output-dependent. In

addition, because the final answer must list variables in ascending order, the task is also sensitive to positional mistakes, but that effect is not part of the semantic dependence notion defined here.

Thus, different tasks expose different semantic dependence profiles: DNA Translation has low input-span dependence and no logical output dependence; Matrix Multiplication has broader input-span dependence but still no logical output dependence; Multi-Needle has broader, order-sensitive input-span dependence; and First-Order Logic may depend both on the input and on prior output units.

**Interpretation.** This notion may be relevant for interpreting the correctness-dynamics analysis in the main text. In tasks with non-empty $G_i$, an early semantic mistake may affect the correctness of later units because later units may rely on previously derived content. By contrast, in tasks such as DNA Translation and Matrix Multiplication, where $G_i = \emptyset$, the correctness of the current target unit is determined by the input alone rather than by earlier target units.

This semantic distinction may also be relevant when interpreting the atom vs. line ranking gap discussed in Appendix J. However, we do not view semantic dependence as a complete explanation of that phenomenon.

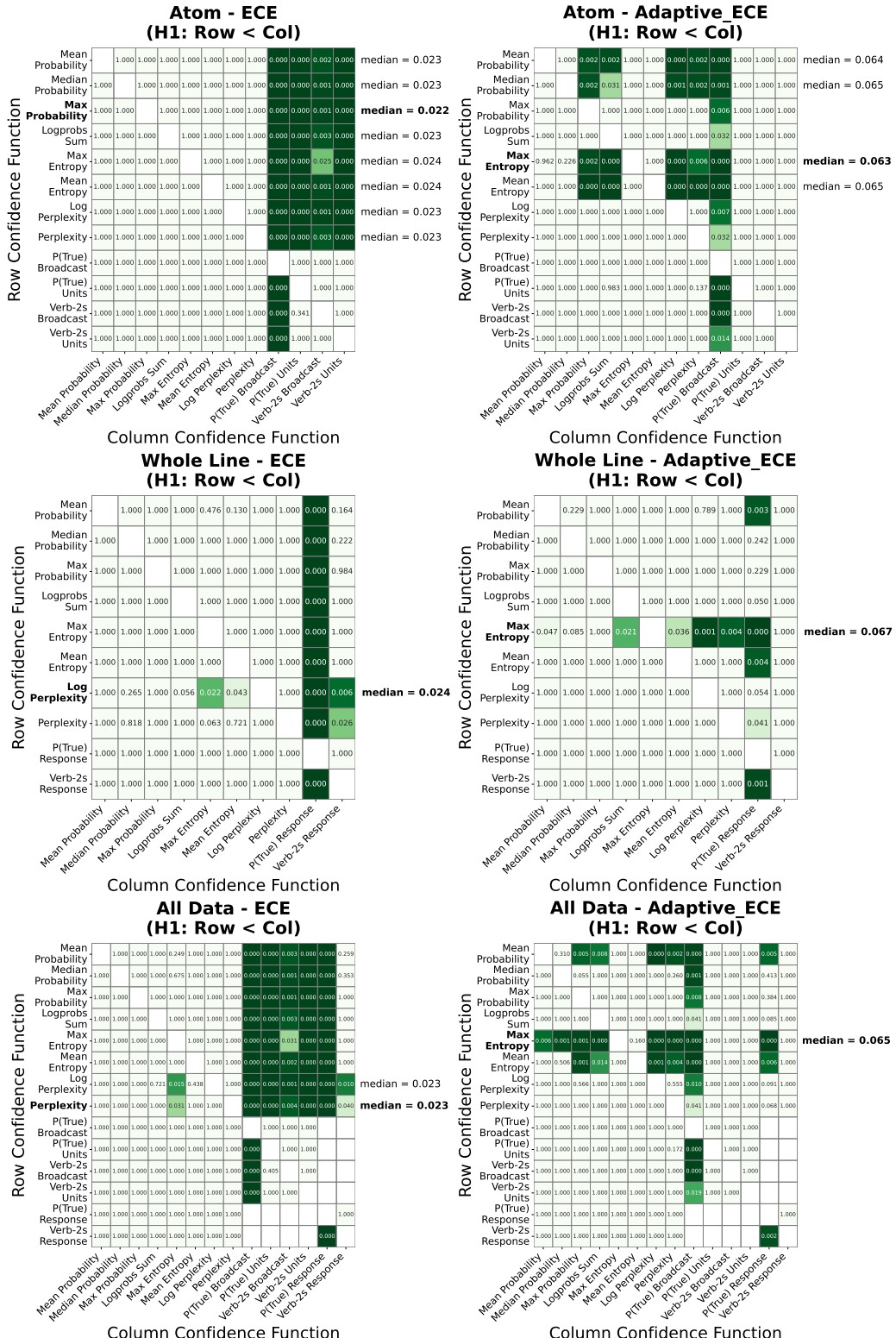

*Figure 30.* Which confidence function is optimal for Calibration among the tested confidence functions.

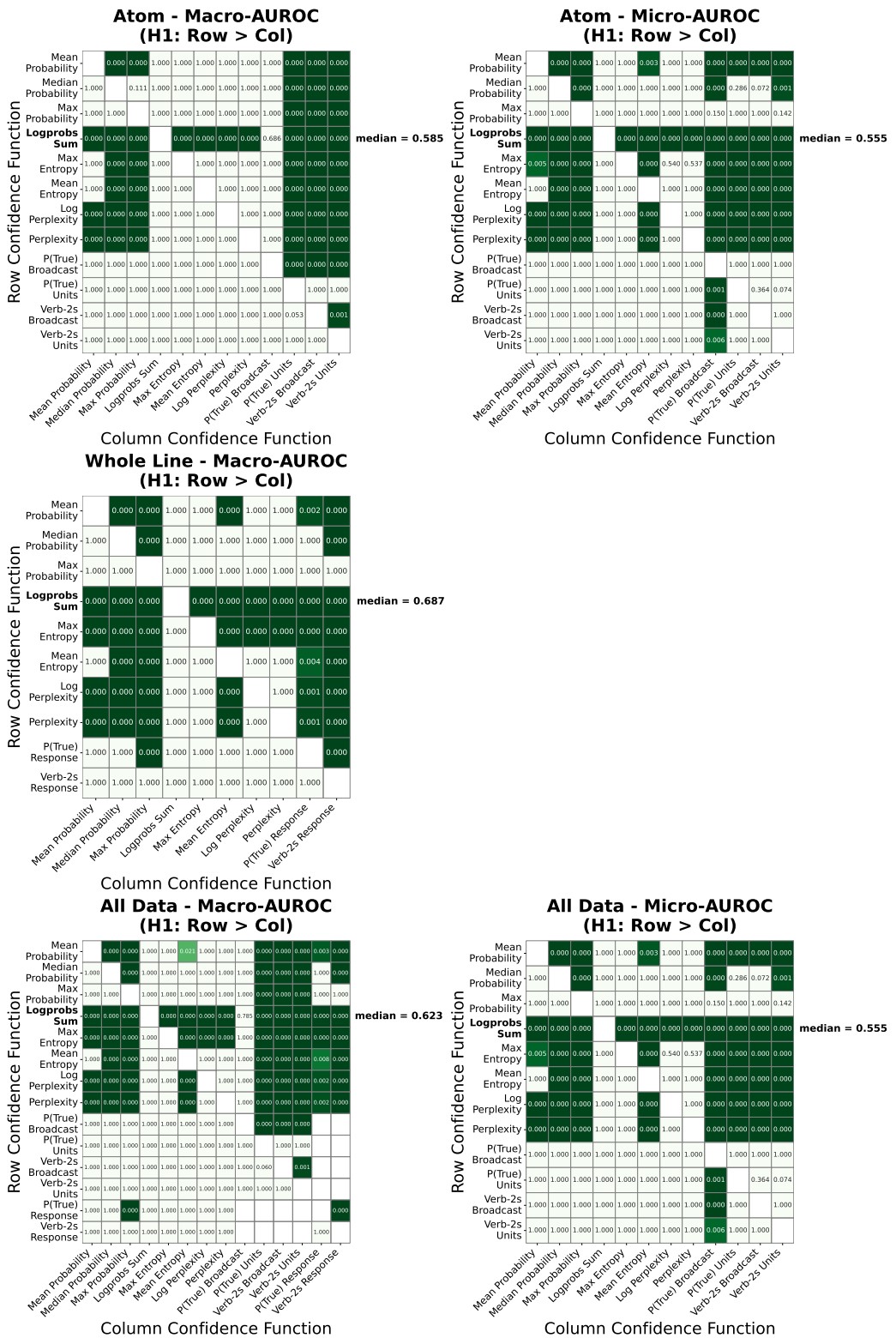

*Figure 31.* Which confidence function is optimal for Ranking among the tested confidence functions.

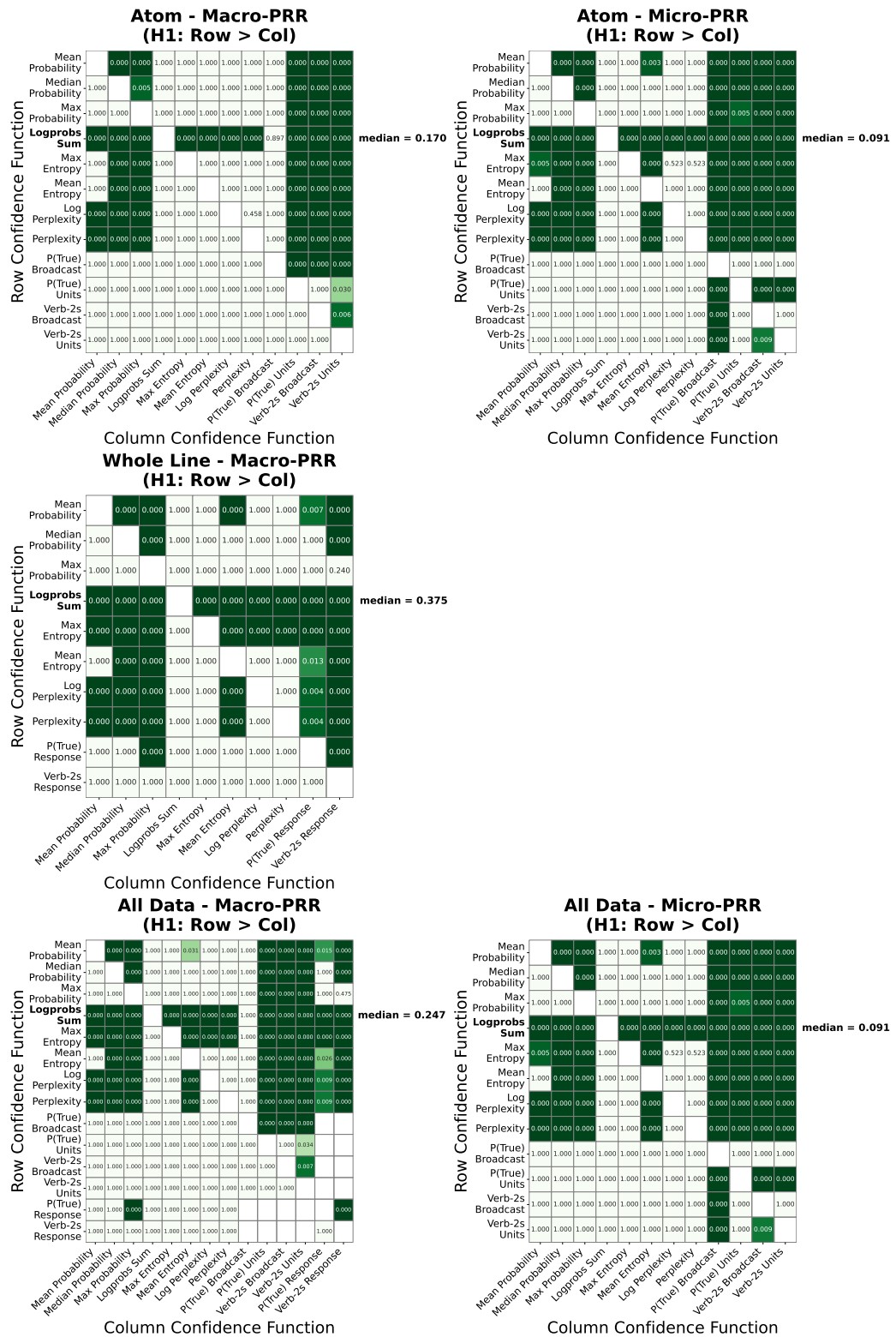

*Figure 32.* Which confidence function is optimal for the prediction-rejection ratio among the tested confidence functions.

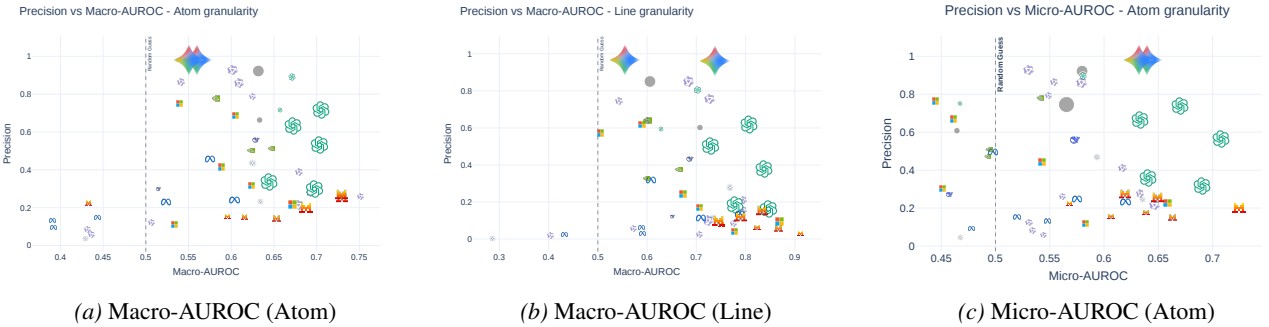

*(a)* Macro-AUROC (Atom)   *(b)* Macro-AUROC (Line)   *(c)* Micro-AUROC (Atom)

*Figure 33.* Relationship between LLM precision and confidence ranking metrics.

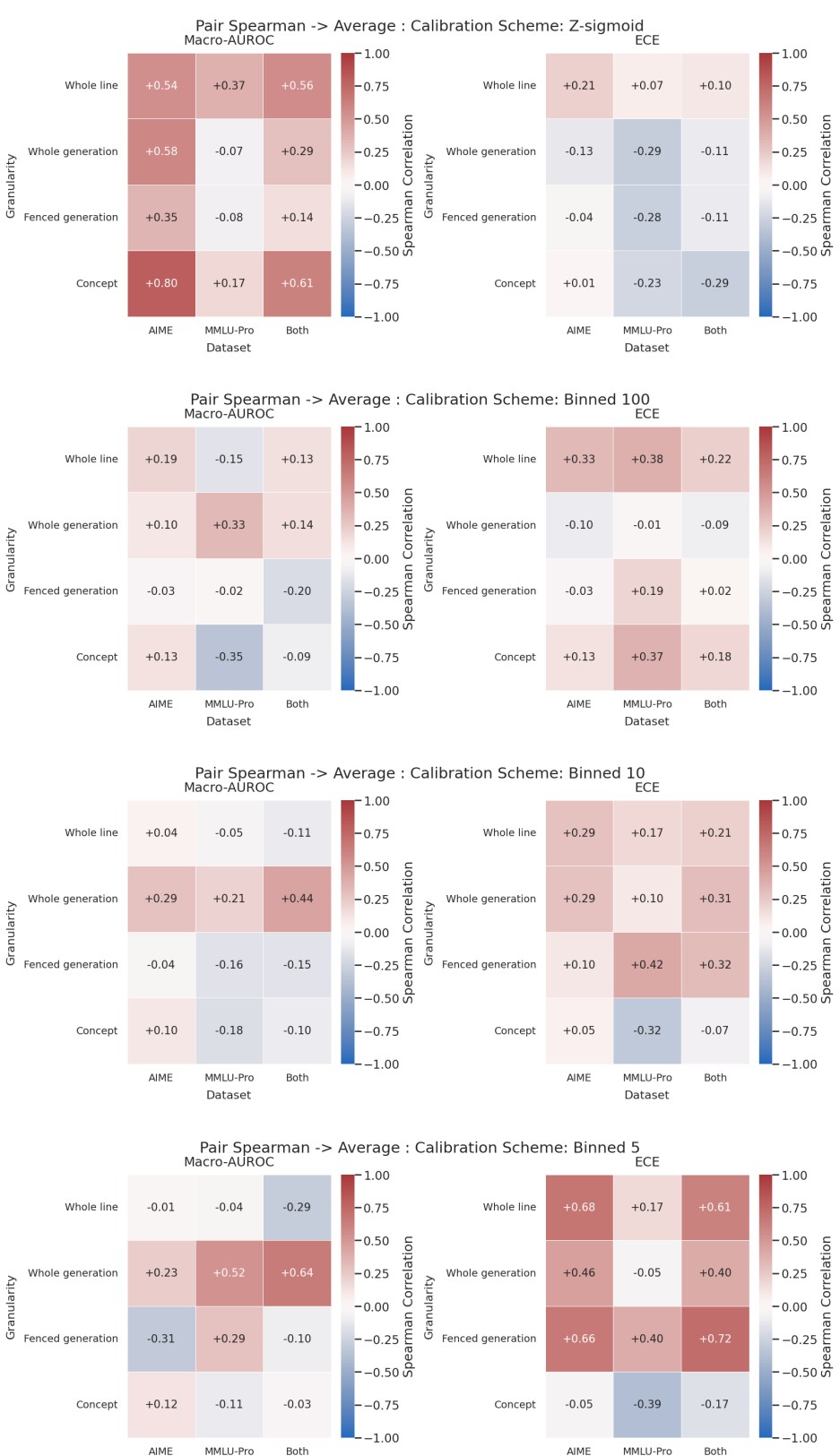

*Figure 34.* Per-model rank correlation

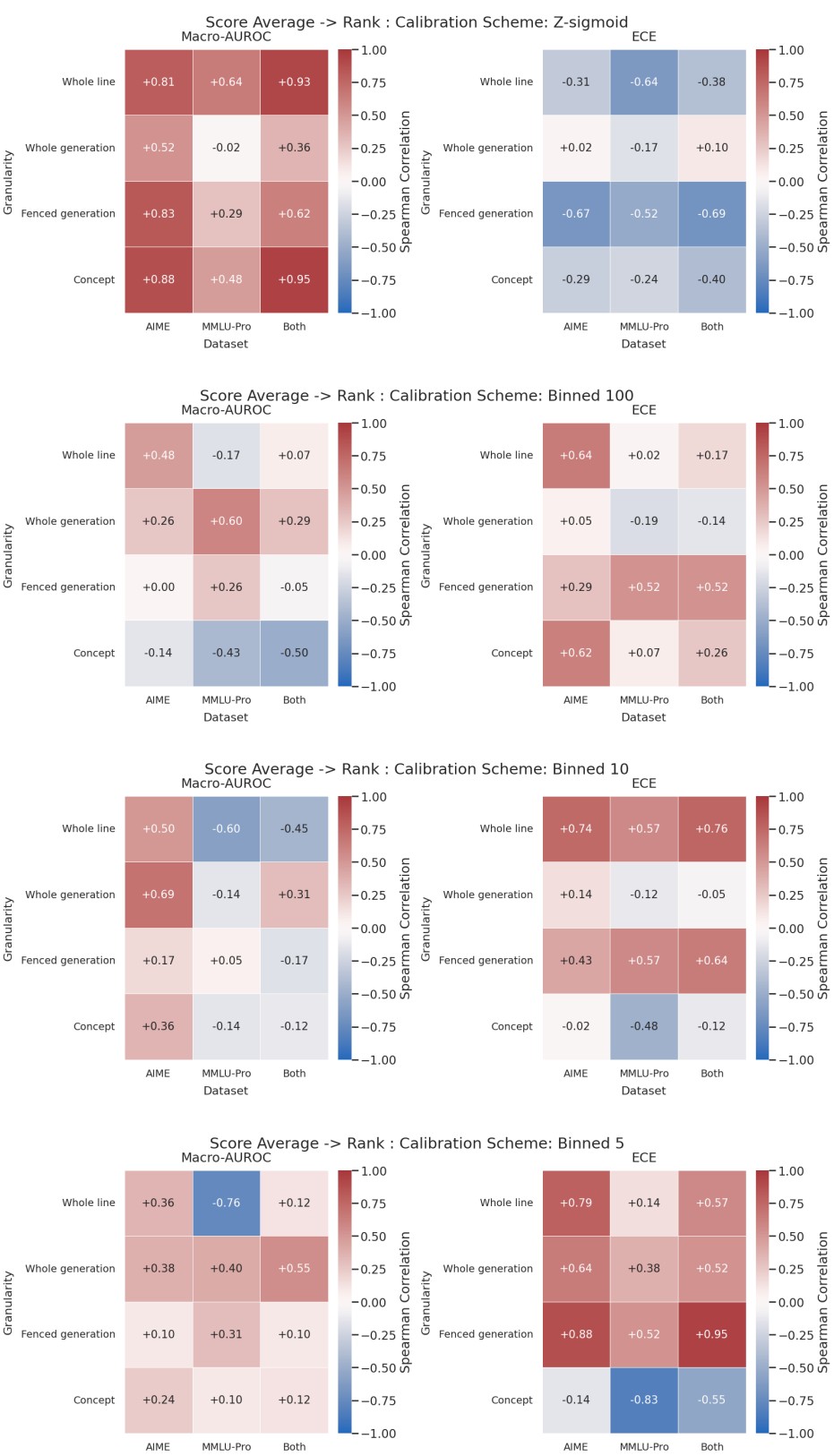

*Figure 35.* Average-score rank correlation.

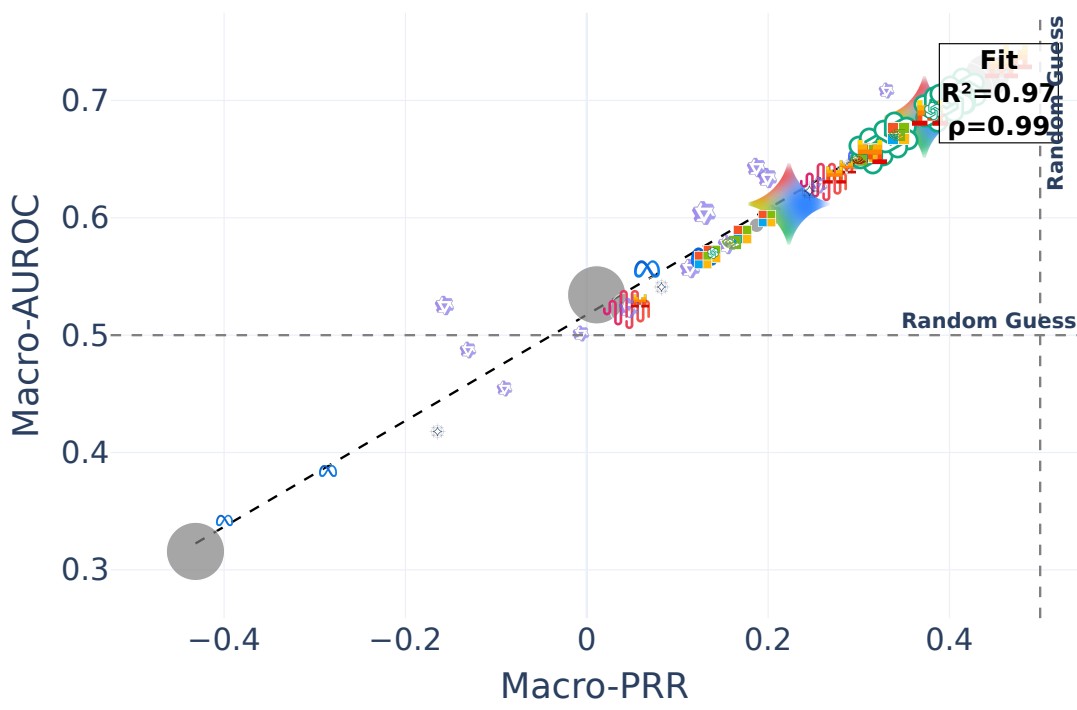

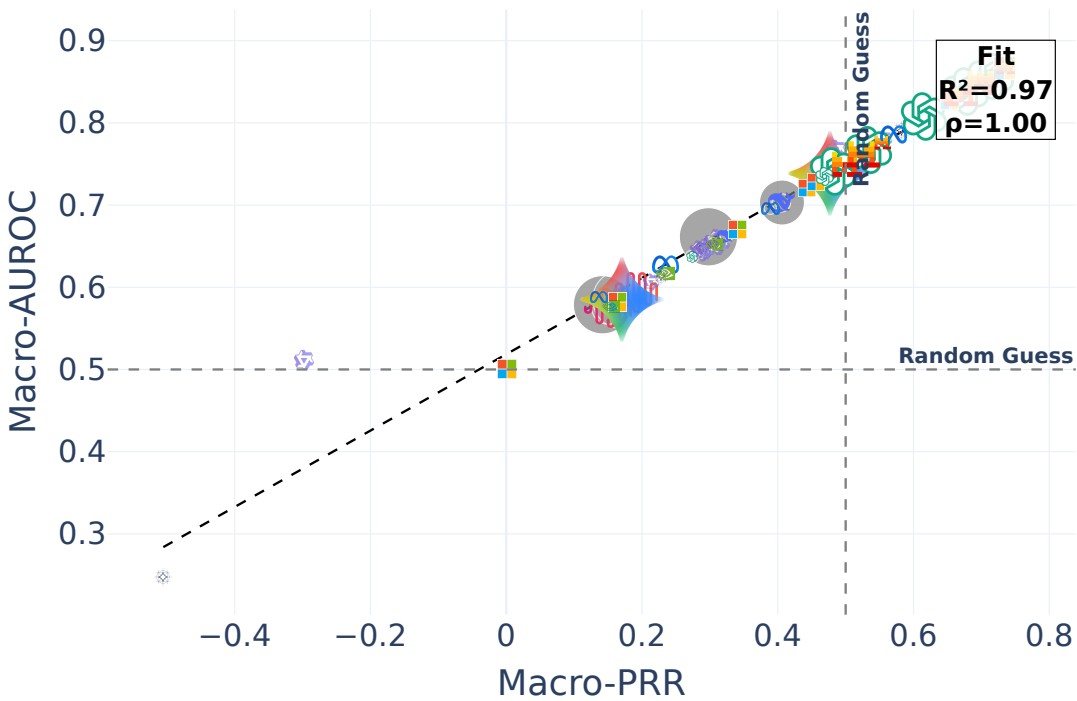

*Figure 36.* Macro-AUROC and Maro-PRR nearly perfect correlations at the atomic and line-unit levels.

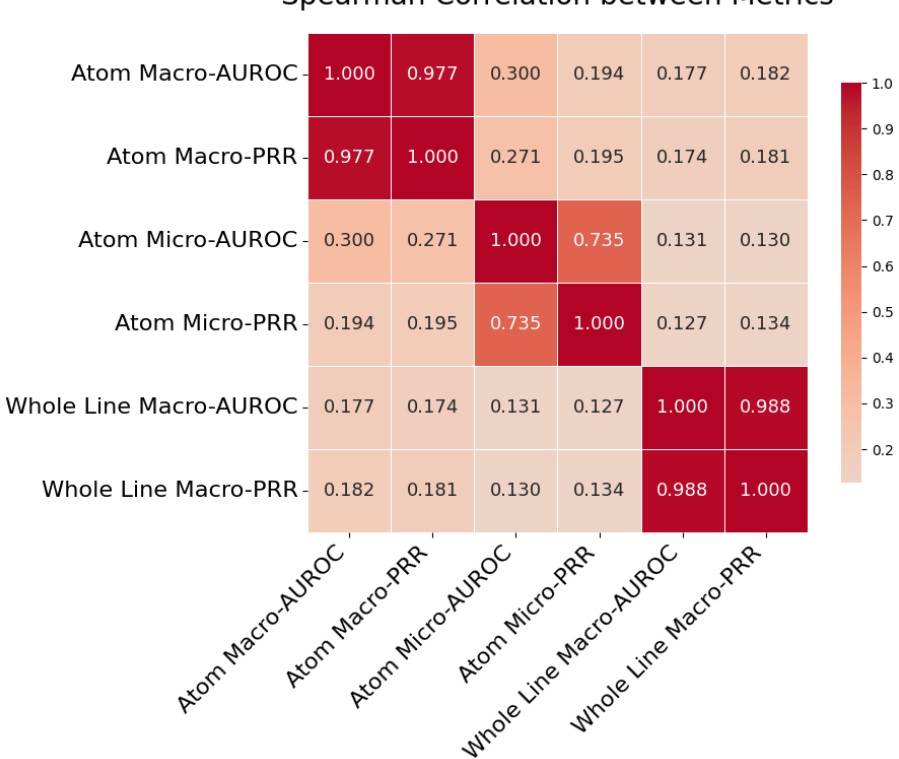

*Figure 37.* Spearman correlation matrix between PRR and AUROC. Both atom and line units show a high correlation between the two.

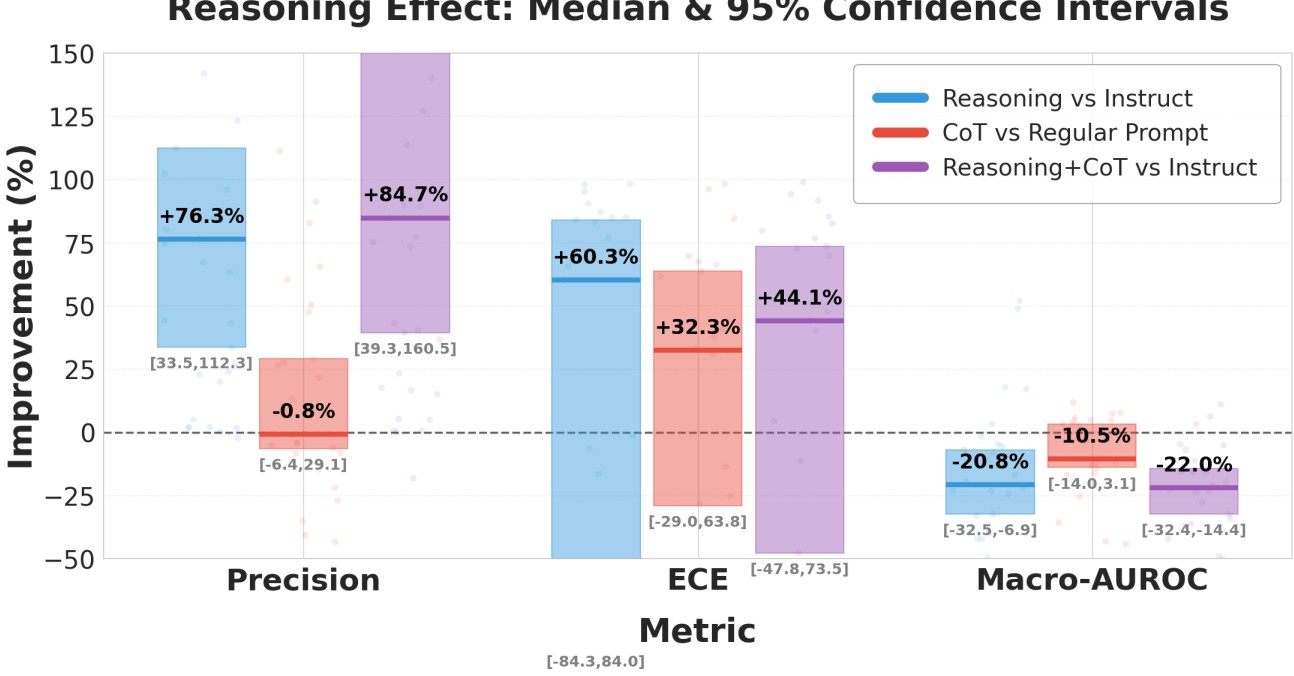

*Figure 38.* A comparison of the effect of the CoT and Reasoning combination with each reasoning strategy individually, on Precision, ECE, and AUROC

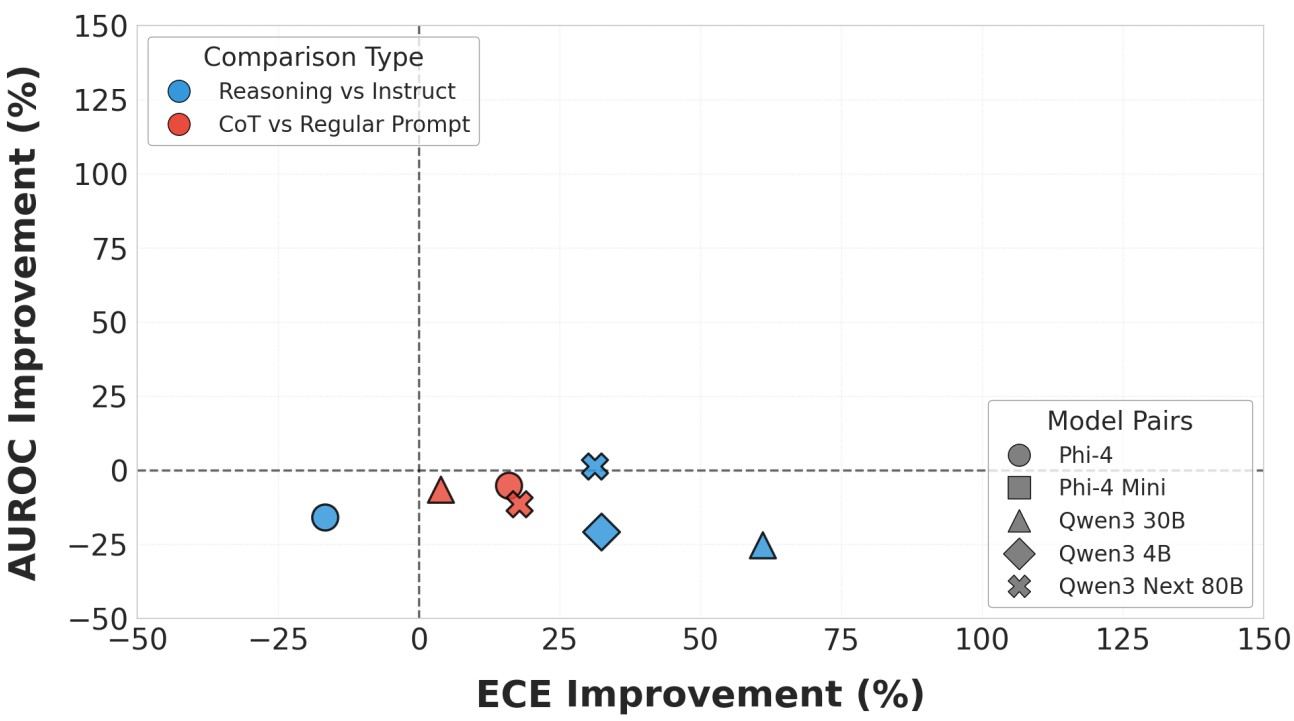

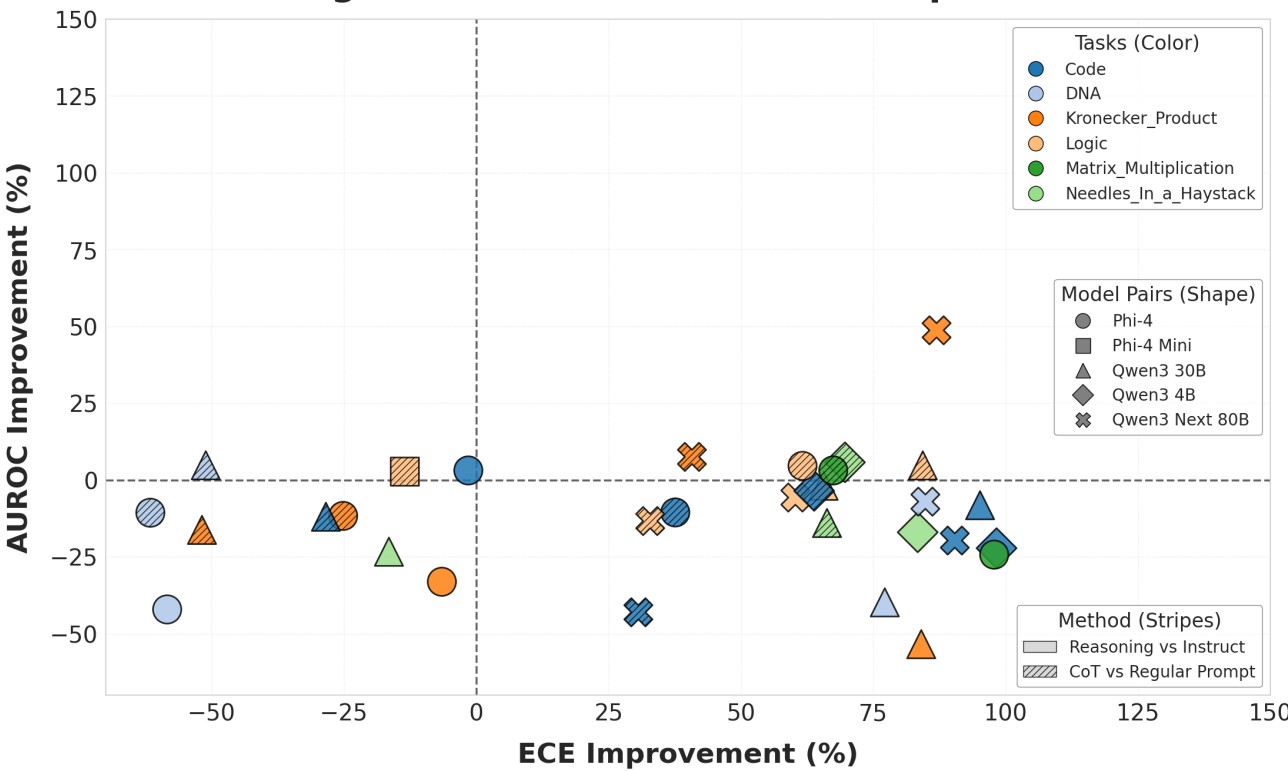

*Figure 39.* Impact of reasoning methods on model performance. Each marker represents the relative change between two identical models, differing only by CoT prompting or reasoning-specific training. Markers above the zero line represent an improvement in that metric relative to the base model's performance, while markers below represent a degradation.

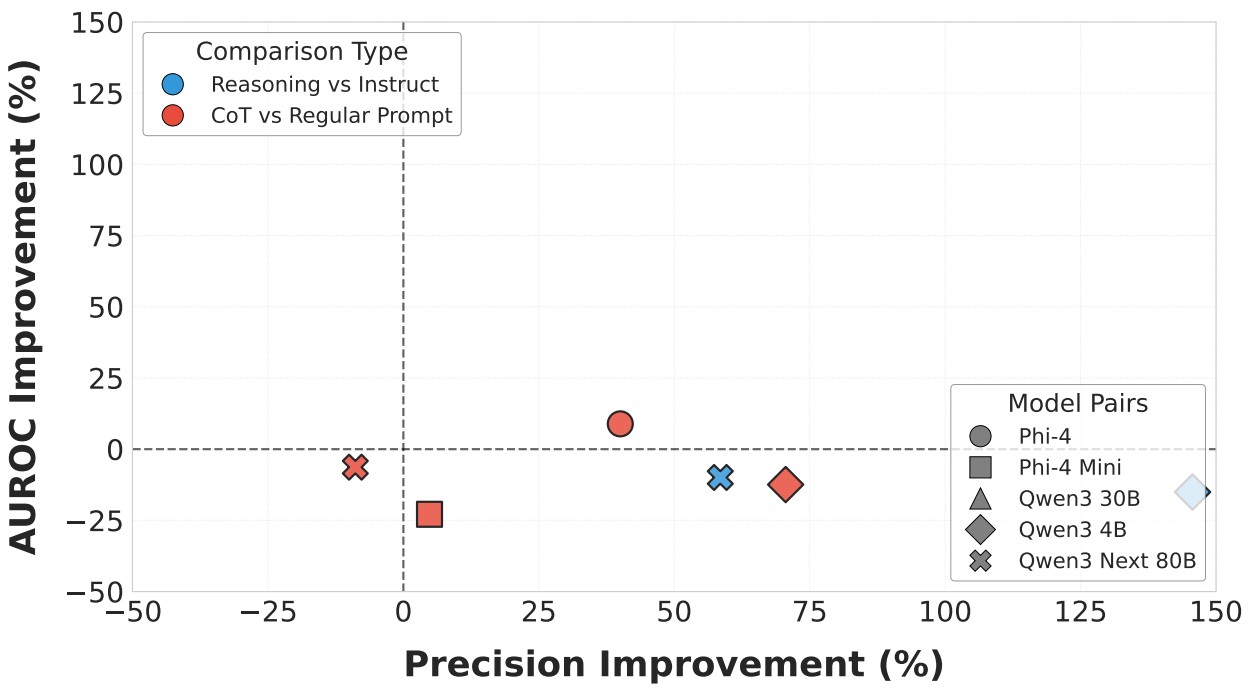

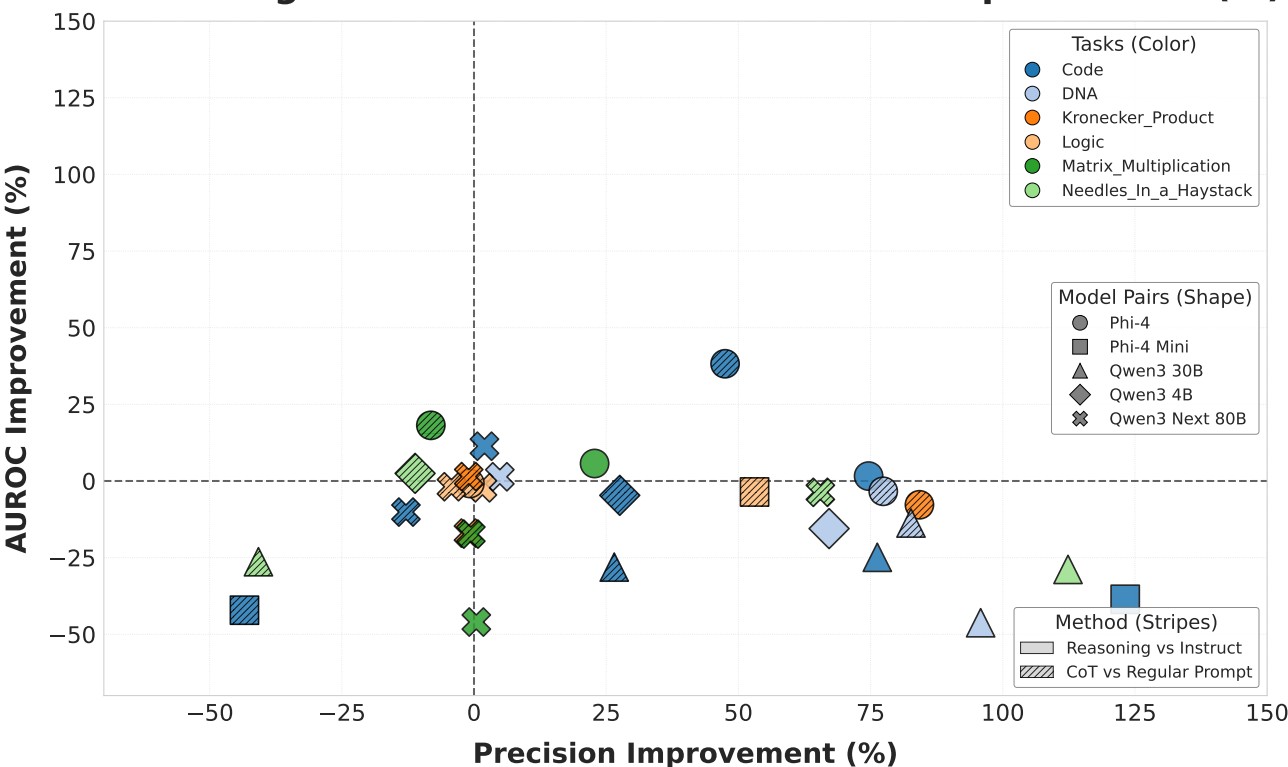

*Figure 40.* Impact of reasoning methods on model performance. Each marker represents the relative change between two identical models, differing only by CoT prompting or reasoning-specific training. Markers above the zero line represent an improvement in that metric relative to the base model's performance, while markers below represent a degradation.

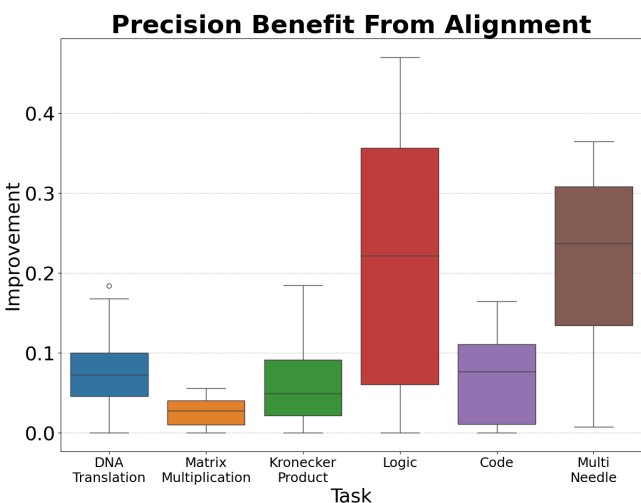

*Figure 41.* Improvement in precision induced by aligning the generated atomic-units with the ground-truth atomic-units using the Needleman–Wunsch algorithm, across tasks.

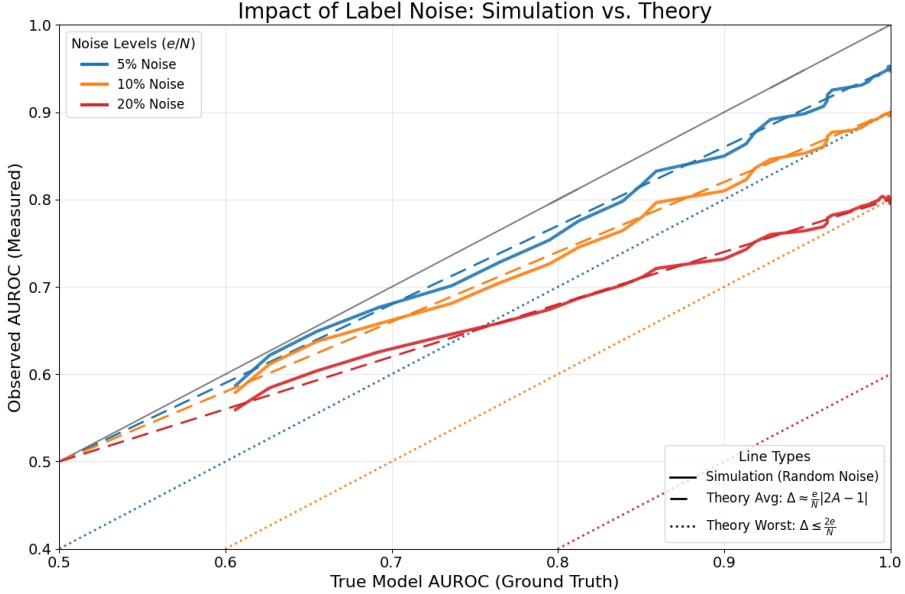

*Figure 42.* Simulation of AUROC degradation under varying label noise. The results demonstrate a "ceiling effect": as model quality ($A$) increases, the sensitivity to noise grows non-linearly. This noise acts as a compressor, artificially reducing the gap between top-tier models and random baselines, which masks true progress in uncertainty estimation.

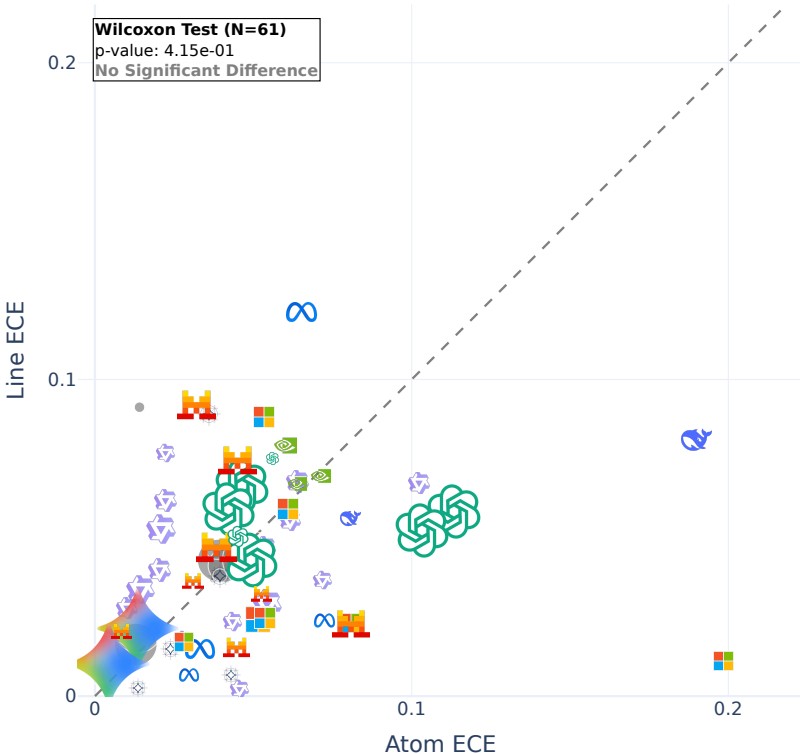

*Figure 43.* A comparison of the evaluated ECE at the atomic-unit and the line-unit resolutions.

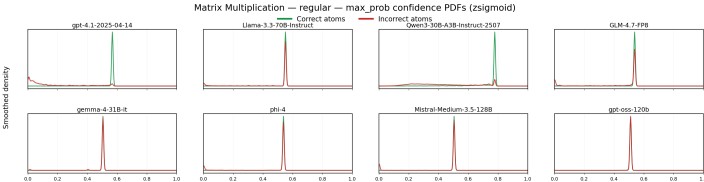

*(a)* PDFs of correct and incorrect atoms' confidence within the Matrix Multiplication Task

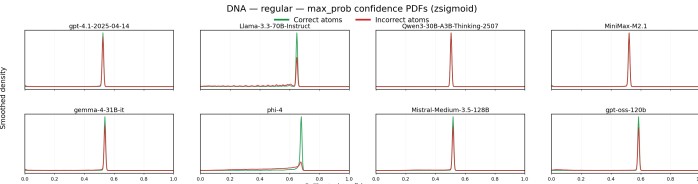

*(b)* PDFs of correct and incorrect atoms' confidence within the DNA Task

*Figure 44.* probability density function of z-sigmoid calibrated maximum-probability confidence scores for correct and incorrect atoms on the Matrix Multiplication (44a) and DNA (44a) tasks. The substantial overlap between the two distributions across models illustrates weak atomic-level separability, consistent with the observed high-resolution ranking gap.

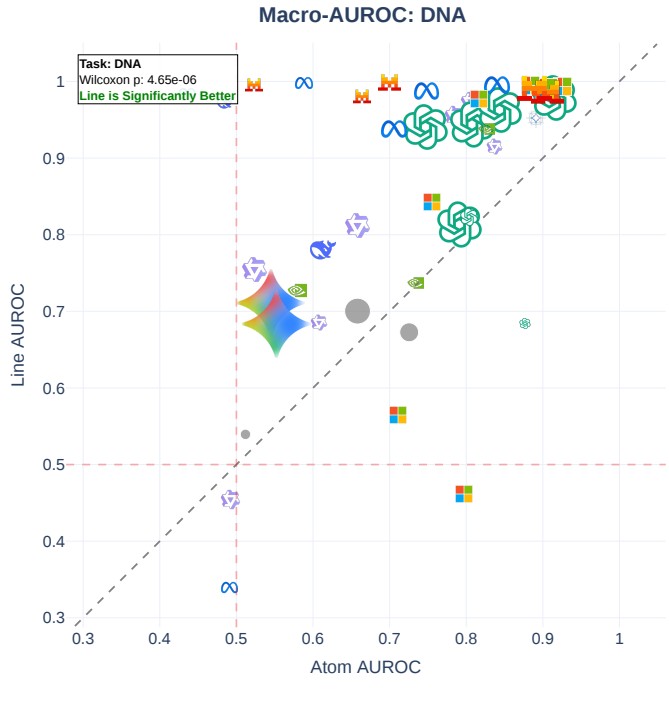

*(a)* DNA Translation Task

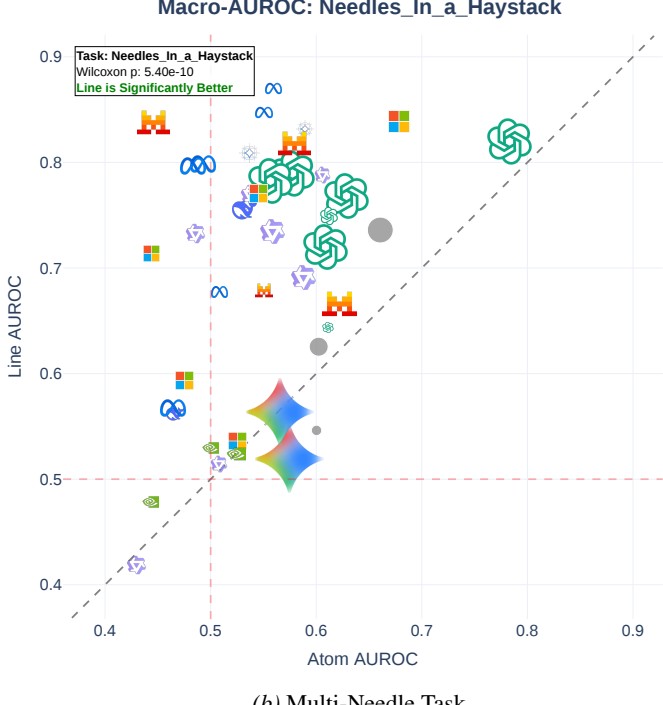

*(b)* Multi-Needle Task

*Figure 45.* Task-level view of the gap between atomic- and line-level ranking. The difference is smaller in DNA Translation than in Multi-Needle, suggesting that the atom-vs.-line ranking gap depends on task structure rather than arising uniformly across tasks.

