# OpenReview forum: "Evaluating LLM Uncertainty in Long-Form Generation Using Deterministic Ground Truth"
_ICML.cc/2026/Conference — ICML 2026 regular_

### Official Review · Reviewer_Lf4F · 2026-03-10

**Soundness:** 2
**Presentation:** 2
**Significance:** 2
**Originality:** 3
**Overall Recommendation:** 4
**Confidence:** 3

**Summary:**

This paper proposes a new benchmark named SALT (Single-answer Atomic Long-form Target). The key issue this benchmark addresses is the evaluation of long-form generations. The authors mention 2 key problems with current benchmarks: 1. determining the ground truth is challenging for atomic claims, 2. decomposing long-form text is noisy and not perfect.

The paper curates a benchmark where generations are in the long-form, while there is an unambiguous ground truth to the questions. There are 6 tasks in the benchmark: matrix multiplication, Kronecker Product (atomic claims are numeric each entity in the final matrix), coding (output of the Python function for a given input is predicted), DNA translation (generating the complement of a dns sequence, each nucleotide is an atomic claim), words collection (identifying all items of a category in a text, each item is an atomic claim). By selecting tasks that have concrete ground truths, authors evaluate both the long-form generation performance of the models and information-based uncertainty estimation techniques.

**Compliance With Llm Reviewing Policy:**

Affirmed.

**Final Justification:**

The authors have addressed my concerns. Therefore, I've increased my score.

**Key Questions For Authors:**

- Which binning scheme is used to calculate ECE?

**Limitations:**

Yes

**Strengths And Weaknesses:**

Strengths:
- Evaluation consists of a comprehensive list of LLMs.
- Uncertainty quantification for long-form generations is an important problem.
- Correctness dynamics on the long-form generation section are interesting.
- Uncertainty quantification analysis is done both for calibration and selective prediction.

Weaknesses:
- Task selections are very niche, hence not convincing. Evaluating uncertainty on these tasks, for example, DNA mapping, does not translate into UE performance on real long-form generations, such as an output to "Explain topic X".
- AUROC values may not be directly comparable, especially for the granularity sensitivity analysis. Since the accuracy of the atomic-unit vs line-unit is highly likely to differ, just looking at AUROC scores being higher for line-unit does not necessarily translate into the effectiveness of line-unit. Prev works use PRR to avoid task accuracy bias [1]
- Tasks are not clearly explained; providing at least one example per task is crucial, especially when proposing a new benchmark.
- In the main body "product of log-probabilities" is mentioned twice. However, I assume it corresponds to log of the product of probabilities from looking at the table 5. These two correspond to different things.

[1] Malinin and Gales, Uncertainty Estimation in Autoregressive Structured Prediction, ICLR 2021

---

> ### Author Rebuttal · Authors · 2026-03-31
>
> Thank you for your constructive feedback.
>
>  > *”Task selections are very niche, hence not convincing. Evaluating uncertainty on these tasks does not translate into UE performance on real long-form generations ”*
>
> We agree it is desirable for uncertainty estimation conclusions on SALT to translate to other long-form generation settings. To assess this directly, we extended the analysis beyond SALT and benchmarked a subset of models on AIME (2024–2025, 60 samples), comparing their uncertainty estimation performance to the corresponding SALT results.
>
> We find strong agreement in the relative ranking of confidence functions across the two benchmarks, with Spearman correlations of 0.83 for AUROC and 0.93 for ECE. This suggests that the choice of confidence function matters beyond SALT, and that its relative ranking largely transfers from SALT to AIME.
>
> We also find that another confidence-related conclusion transfers across the two benchmarks: confidence-function performance differs across uncertainty estimation objectives. In both SALT and AIME, the correlation between confidence-function performance on AUROC versus ECE remains low (Spearman 0.07 on SALT and 0.17 on AIME), supporting the conclusion that different uncertainty objectives require different confidence functions.
>
> Overall, these results suggest that our main conclusions also transfer beyond SALT, and we will include a more detailed analysis in the revision.
>
>  > *”AUROC values may not be directly comparable, especially for the granularity sensitivity analysis. Since the accuracy of the atomic-unit vs line-unit is highly likely to differ, just looking at AUROC scores being higher for line-unit does not necessarily translate into the effectiveness of line-unit. Prev works use PRR to avoid task accuracy bias [1]”*
>
> Thank you for this helpful point and for pointing us to [1]. Our intention in the granularity analysis was not to claim that one granularity is universally superior, but to study how confidence ranking behaves under different evaluation resolutions. In this work, we focus on two uncertainty-estimation aspects: calibration and ranking. For ranking, we use AUROC, whereas PRR evaluates a different, complementary aspect: selective prediction.
>
> We agree that selective prediction is also important, and we intend to include it in the revision. We had originally considered AURC / E-AURC for this purpose [2], and PRR is an excellent suggestion that we will embrace as well, together with citing [1]. We will therefore make the distinction between ranking and selective prediction more explicit in the revision, and report a selective-prediction analysis alongside the current ranking results.
>
> At the same time, prior work has shown that AUROC remains a meaningful metric for uncertainty ranking even when model accuracies differ [3]. Accordingly, we do not interpret the higher line-level AUROC as showing that line granularity is universally “better,” but rather that confidence ranking emerges more clearly at that coarser semantic resolution.
>
> [2] Yonatan Geifman, Guy Uziel, Ran El-Yaniv. Bias-Reduced Uncertainty Estimation for Deep Neural Classifiers. In International Conference on Learning Representations (ICLR), 2019.
>
> [3] Ido Galil, Mohammed Dabbah, Ran El-Yaniv. What Can we Learn From The Selective Prediction And Uncertainty Estimation Performance Of 523 Imagenet Classifiers? In International Conference on Learning Representations (ICLR), 2023.
>
>  > *”Tasks are not clearly explained; providing at least one example per task is crucial”*
>
> We agree, and we will add at least one full example per task in the revision. Additionally, we will provide the full code of the tasks, as well as the full curated datasets for reproducibility.
>
>  > *”In the main body "product of log-probabilities" is mentioned twice. However, I assume it corresponds to log of the product of probabilities from looking at the table 5. These two correspond to different things.”*
>
> You are correct, thank you for catching this. Table 5 gives the correct definition, namely the sum of log-probabilities, equivalently the log of the product of probabilities, not the product of log-probabilities. We will correct this terminology in the revision.
>
>  > *”Which binning scheme is used to calculate ECE?”*
>
> We use fixed equal-width binning with m=15 bins, following standard practice. The ECE definition appears in Section 2.3, and the choice of m=15 is specified in Appendix D.

---

> > ### Author Rebuttal · Reviewer_Lf4F · 2026-04-04
> >
> > I thank the authors for their response. I've updated my score

---

> > > ### Author Response · Authors · 2026-04-08
> > >
> > > We sincerely thank the reviewer for acknowledging our work and for raising the score. We appreciate the time and effort spent reviewing our paper.

---

### Official Review · Reviewer_RUSW · 2026-03-11

**Soundness:** 4
**Presentation:** 3
**Significance:** 3
**Originality:** 3
**Overall Recommendation:** 5
**Confidence:** 5

**Summary:**

This paper test the calibration characteristics of a large number of LLMs on long-form generation problems, under different settings and via different metrics. Crucially, the paper defines units for each task over which correctness can meaningfully be measured, allowing the authors to measure calibration at the level of sub-outputs. This is done over matrix manipulation, logic, DNA translation, code, and needle-in-a-haystack reordering tasks. The authors include a careful analysis that tries to disentangle the effect of changes in accuracy to changes in calibration.
The paper finds different effects on smaller/open-source models than larger models, and provides insights on which confidence estimation methods to use under which circumstances.

**Compliance With Llm Reviewing Policy:**

Affirmed.

**Final Justification:**

I think this is a strong paper and should be accepted.

**Key Questions For Authors:**

While the paper establishes accuracy correlations to other generation tasks, is there any way to also establish confidence correlations? I understand that this is effectively  the problem that the paper is seeking to address (measuring confidence on long-form generation being non-trivial) but it would help address the concerns I have about whether findings in the given domains translate to other domains like long-form QA.

**Limitations:**

yes

**Strengths And Weaknesses:**

## Strengths:
Overall, I think this paper does a good job of adding depth in an area where there is already a lot of work. Some points that are especially strong:

- The causal analysis is very valuable. I have seen this relationship discussed more but have not seen any progress towards quantifying or controlling for it. I think this is a valuable addition.
- The insights in 5.2 generally are valuable. The paper tests an impressive number of models and settings.

I also appreciate the clarity of the writing and the fact that text-to-code is one of the tasks.

## Weaknesses
Some of the tasks feel artificial to me, especially matrix multiplication and DNA translation. While I understand that these are intended as diagnostic tasks, I worry about whether conclusions here would translate to real-world text-generation domains. Others (code, logic) are have more real-world applicability, but I'm still unsure about the degree to which this would translate into a task like long-form report generation.

A smaller weakness is that some of the findings seem unsurprising, especially the finding that models are more accurate earlier in generation than later. This feels like a foregone conclusion especially for something like DNA translation, where an early error will offset everything after it (although the string matching experiment addresses this to some extent.)

## Smaller points/typos:

- Related work: https://arxiv.org/abs/2506.01114
- Double Machine Learning -> citation needed
- On using fixed ECE: could also try adaptive ECE (https://arxiv.org/pdf/1904.01685)
- L1115 should be citep, same for L1318
- Table 6 should be smaller

---

> ### Author Rebuttal · Authors · 2026-03-31
>
> Thank you for the positive review, the very helpful notes, the typo corrections, and for pointing us to relevant resources.
>
>  > *”Some of the tasks feel artificial to me, I worry about whether conclusions here would translate to real-world text-generation domains”*
>
> > *”While the paper establishes accuracy correlations to other generation tasks, is there any way to also establish confidence correlations? It would help address the concerns I have about whether findings in the given domains translate to other domains like long-form QA.”*
>
> We agree that establishing confidence correlations beyond SALT would strengthen the paper. To address this concern, we extended the analysis beyond accuracy and benchmarked a subset of models on AIME (2024–2025, 60 samples), comparing their confidence behavior to the corresponding SALT results.
>
> We find strong agreement in the relative ranking of confidence functions across the two benchmarks, with Spearman correlations of 0.83 for AUROC and 0.93 for ECE. This suggests that the choice of confidence function matters beyond SALT, and that its relative ranking largely transfers from SALT to AIME.
>
> We also find that another confidence-related conclusion transfers across the two benchmarks: confidence-function performance differs across uncertainty estimation objectives. In both SALT and AIME, the correlation between confidence-function performance on AUROC versus ECE remains low (Spearman 0.07 on SALT and 0.17 on AIME), supporting the conclusion that different uncertainty objectives require different confidence functions.
>
> Overall, these results suggest that our main conclusions also transfer beyond SALT, and we will include a more detailed analysis in the revision.
>
> > *”A smaller weakness is that some of the findings seem unsurprising, especially the finding that models are more accurate earlier in generation than later. This feels like a foregone conclusion especially for something like DNA translation, where an early error will offset everything after it (although the string matching experiment addresses this to some extent.)”*
>
> We agree that the original presentation of Section 5.1 did not make the core question sharp enough. Although degradation with longer context and degradation following prefix errors may appear intuitively related, the relationship between them is less clear: do they reflect the same underlying phenomenon, or distinct sources of degradation? And if they are distinct, which of the two is more impactful?
>
> Importantly, the aggregate prior-correctness trend in Section 5.1 is not driven by DNA alone: Appendix B shows the same qualitative pattern across the benchmark tasks. Leveraging the unique properties of SALT, we therefore added a deeper error analysis that decouples the effects of *context correctness* from *context length*. For this follow-up analysis, we use DNA not because it makes the phenomenon trivial, but because each output atom is determined directly by the input sequence, so the validity of the next atom does not inherently depend on previously generated output atoms. This makes DNA a particularly suitable setting for separating the effects of prefix correctness from those of context length.
>
> This deeper analysis yields two concrete findings:
>
> - **Impact of Context Correctness**: By holding generation length constant and isolating context correctness, we directly measure its influence on accuracy. We observe a 15% improvement in accuracy when the preceding context is strictly correct.
>
> - **Context Enhancement Experiment**: To determine which factor is more dominant, context quality or context length, we introduce additional correct atoms into the prefix. This creates two opposing pressures:
>
>     1) **Positive**: increasing the overall correctness ratio.
>     2) **Negative**: increasing the total context length.
>
> Under this intervention, we observe an accuracy degradation of up to 6%, suggesting that context length poses a greater challenge to model stability than the ratio of correct-to-incorrect prior atoms.
>
> > *"Smaller points/typos..."*
>
> Thank you, we will make sure to incorporate these into our revision. We actually considered reporting adaptive calibration error (ACE) as well, and following your feedback and interest, we are now even more inclined to do so in the revision.

---

> > ### Author Rebuttal · Reviewer_RUSW · 2026-04-02
> >
> > Thanks for the responses, which address my questions -- I hope to see this paper at the conference.

---

> > > ### Author Response · Authors · 2026-04-08
> > >
> > > We sincerely thank the reviewer for their appreciation of our work. We value the time and effort spent reviewing our paper.

---

### Official Review · Reviewer_jNBU · 2026-03-12

**Soundness:** 3
**Presentation:** 4
**Significance:** 3
**Originality:** 3
**Overall Recommendation:** 4
**Confidence:** 3

**Summary:**

This paper introduces SALT (Single-answer Atomic Long-form Target), a benchmark for evaluating uncertainty estimation in long-form generation under a deterministic, fine-grained setting. The key idea is to avoid noisy judge-based verification by constructing procedurally generated long-output tasks with a single known ground-truth sequence, allowing correctness to be evaluated exactly at multiple granularities (especially atomic-unit and line-unit levels). The benchmark includes six task families spanning mathematics, code execution, logic, translation, and word collection, and is used to evaluate more than 50 LLMs.

Using SALT, the paper studies several aspects of uncertainty estimation in long-form generation, including correctness dynamics over the course of generation, the relative performance of different confidence functions for calibration and ranking, the effect of evaluation granularity on ranking quality, and the impact of reasoning mechanisms such as CoT prompting and reasoning-tuned models. The paper reports several notable findings, including: (i) error propagation over long generations, (ii) different confidence functions being preferable for calibration vs. ranking, (iii) substantially better ranking performance at coarser granularity, and (iv) a reasoning trade-off where reasoning improves correctness but degrades confidence ranking.

**Compliance With Llm Reviewing Policy:**

Affirmed.

**Final Justification:**

The rebuttal addressed my main concerns constructively and makes me more comfortable with acceptance, but some limitations remain, particularly regarding transfer beyond structured deterministic settings and the strength of certain methodological conclusions. I therefore remain at Weak Accept.

**Key Questions For Authors:**

1. The calibration results depend on mapping non-probabilistic confidence scores into [0,1] via z-score normalization and a sigmoid transform. Could the authors justify why this should be interpreted as calibration rather than score normalization, and report whether the main calibration conclusions hold under more standard post-hoc calibration methods?
2. For the reasoning trade-off, can the authors provide stronger evidence that the degradation in AUROC is attributable to reasoning itself rather than correlated differences in output distribution, verbosity, or other post-training changes between instruct and reasoning model variants?
3. Could the authors better delimit the scope of their claims and clarify which conclusions they believe are specific to structured deterministic tasks versus likely to transfer to broader long-form generation settings?

**Limitations:**

yes

**Strengths And Weaknesses:**

Strength:
- The motivation is strong. As LLMs are increasingly deployed in settings involving long-form outputs, understanding uncertainty at a finer granularity than the full response is highly relevant. The paper makes a compelling case for moving beyond coarse sequence-level evaluation.
- The main contribution, SALT, is meaningful. The benchmark design is coherent: model-generated outputs, unit decomposition aligned with task semantics, deterministic ground truth, and unbounded procedural instance generation. This is a strong benchmark contribution rather than a minor repackaging of existing tasks.
- The experimental study is broad and more insightful than a simple leaderboard. It includes several interesting empirical observations. In particular, the degradation of correctness over the course of generation, the dependence of ranking quality on granularity, and the reported reasoning trade-off are all potentially valuable findings for the community. These results make the paper more insightful than a standard benchmark paper that only reports aggregate comparisons.

Weakness:
- The benchmark supports strong claims about uncertainty estimation in structured deterministic long-form outputs, but weaker claims about long-form generation more broadly. I would therefore encourage the authors to narrow some of the broader framing.
- A second concern is the treatment of calibration. For confidence functions such as perplexity and entropy, the paper maps scores into  [0,1] via z-score normalization followed by a sigmoid transform and then evaluates calibration metrics such as ECE. This seems closer to a monotonic remapping than to a standard post-hoc calibration procedure. As a result, I am not fully convinced that conclusions such as “perplexity is best for calibration” should be interpreted as robust calibration findings rather than findings under a particular score transformation.
- Third, some of the causal language is stronger than I am comfortable with given the evidence. I would be more convinced if these results were framed as suggestive evidence of a directional relationship rather than as strong causal claims. For example, I do not think the current evidence fully isolates reasoning itself as the causal factor.

In general, the benchmark contribution is substantial and the empirical findings are valuable, I would suggests some claims should be narrowed and certain methodological choices deserve stronger justification.

---

> ### Author Rebuttal · Authors · 2026-03-31
>
> Thank you for the detailed review and for the careful questions that helped us sharpen the framing of our claims.
>
> > *“The benchmark supports strong claims about uncertainty estimation in structured deterministic long-form outputs, but weaker claims about long-form generation more broadly. I would encourage ... to narrow some of the broader framing.”*
>
> >*”3. Could the authors ... clarify which conclusions they believe are specific to structured deterministic tasks versus likely to transfer to broader long-form generation settings?”*
>
> Thank you for raising this point. SALT most directly supports claims in the structured deterministic setting, and we will narrow the broader framing accordingly. To evaluate which conclusions may transfer beyond SALT, we benchmarked a subset of models on AIME (2024-25, 60 samples) and compared their uncertainty estimation behavior to the corresponding SALT results.
>
> We find strong agreement in the relative ranking of confidence functions across the two benchmarks, with Spearman correlations of 0.83 for AUROC and 0.93 for ECE. This supports one confidence-related conclusion beyond SALT: confidence function choice matters, and worst-performing functions are broadly consistent across SALT and AIME.
>
> We also find that another confidence-related conclusion transfers across the two benchmarks: confidence-function performance differs across uncertainty estimation objectives. In both SALT and AIME, the correlation between confidence-function performance on AUROC versus ECE remains low (Spearman 0.07 on SALT and 0.17 on AIME), supporting the conclusion that different uncertainty objectives require different confidence functions.
>
> Overall, these results suggest that our main conclusions also transfer beyond SALT, and we will include a more detailed analysis in the revision.
>
> > *”I am not fully convinced that conclusions such as “perplexity is best for calibration” should be interpreted as robust calibration findings rather than findings under a particular score transformation.”*
>
> >*”1. The calibration results depend on mapping non-probabilistic confidence scores via z-score normalization and a sigmoid transform. Could the authors justify why this should be interpreted as calibration rather than score normalization, and report whether the main calibration conclusions hold under more standard post-hoc calibration methods?”*
>
> We agree that, in our setting, the more accurate framing is score normalization rather than post-hoc calibration, and we will revise the paper accordingly. Our goal was not to claim a dedicated calibration procedure, but to map different confidence scores into the probability range [0,1] for calibration evaluation. Appendix D.1 already describes this step as a normalization procedure, and we will make the main-text framing consistent.
>
> To verify that our conclusions are not an artifact of a specific normalization choice, we repeated the analysis with min-max normalization, isotonic post-processing, and z-score normalization followed by a sigmoid transform. Min-max performs worst overall, while z-score followed by sigmoid, performs slightly better than isotonic post-processing. Importantly, the qualitative conclusions remain unchanged. We will add this clarification and the corresponding results to Appendix D.1 in the revision.
>
> > *”some of the causal language is stronger than I am comfortable with. I would be more convinced if these results were framed as suggestive evidence of a directional relationship.”*
>
> Despite our efforts in the causal analysis [Appendix F.3], we agree that it is infeasible to rule out all potential confounders and make strong causal claims with high confidence. While we did aim to convey this nuance, we will further tone down the relevant statements and frame them as suggestive evidence of a directional relationship.
>
> >*”can the authors provide stronger evidence that the degradation in AUROC is attributable to reasoning itself rather than correlated differences in output distribution, verbosity, or other post-training changes between instruct and reasoning model variants?”*
>
> To reduce model-related confounding, we compare pairs of instruct and reasoning models that share the same pre-trained checkpoint and differ only in post-training on reasoning traces.
>
> We also tested Chain-of-Thought prompting as a softer reasoning regime that likewise induces verbose outputs. The two settings do not have the same effect on AUROC: internalized reasoning causes a larger drop in Micro-AUROC than CoT prompting, even though both increase verbosity.
> Stronger evidence could come from a controlled training study starting from one checkpoint and applying two otherwise identical post-training regimes, one with reasoning traces and one without. Such an experiment would still have limitations, but would make the comparison cleaner. Unfortunately, such a controlled experiment is beyond the current computational scope of this work.

---

> > ### Author Rebuttal · Reviewer_jNBU · 2026-04-03
> >
> > The rebuttal addressed my main concerns constructively and makes me more comfortable with acceptance, but some limitations remain, particularly regarding transfer beyond structured deterministic settings and the strength of certain methodological conclusions. I therefore remain at Weak Accept.

---

> > > ### Author Response · Authors · 2026-04-08
> > >
> > > We sincerely thank the reviewer for acknowledging our work and for the valuable discussion. We appreciate the time and effort spent reviewing our paper.

---

### Official Review · Reviewer_GRXz · 2026-03-13

**Soundness:** 4
**Presentation:** 2
**Significance:** 2
**Originality:** 3
**Overall Recommendation:** 4
**Confidence:** 4

**Summary:**

The paper introduces the SALT benchmark for evaluating LLM uncertainty in long contexts, consisting of 6 tasks with deterministic ground truth that does not rely on human experts. Analyzing 50+ LLMs on SALT finds that the relationship between calibration and accuracy flips from lower- to higher-performing models, and reasoning often comes in the form of a trade-off.

**Compliance With Llm Reviewing Policy:**

Affirmed.

**Final Justification:**

The rebuttal addressed most of my initial concerns, and I've decided to raise my score.

**Key Questions For Authors:**

N/A

**Limitations:**

yes

**Strengths And Weaknesses:**

Strengths:
- The paper introduces a novel benchmark that sits in the intersection of two important research questions, long-form generation and uncertainty estimation
- The authors provide detailed motivation for most key problem framing and benchmark design decisions

Weaknesses:
- The figures and tables are not very readable. All figures are very small and need to be zoomed in to read the content on them; Figure 4, which seems to present the most important results, is especially congested -- I can barely see what's going on; I like the intuition of using different sized logos, but in this case it is clearly not working on a visual level.
- The conclusion of Section 5.1 is not at all surprising. What is new to your findings beyond the well-established intuition that LLMs are more sensitive to errors in long contexts? Having some qualitative error analysis would really help, as I imagine that uncertainty estimation shed light on where or how LLMs make mistakes in long form generation.
- The presentation of Section 5.2 is not clear: instead of crowding everything into a numbered list, wouldn't it be better organized if the presentation of key results is disentangled from the more detailed analysis (in a separate "Discussion" section)?
- A lot of Section 2 could go into the appendix. For instance, the metrics (precision, recall, and ECE) in Section 2.3 do not seem to be custom implementations, so we don't really need the detailed formalizations in the main text (it can feel a bit like math for math's sake). Section 2.1 could also be better consolidated.
- Is Section 3 really necessary? You've already surveyed a lot of relevant prior work in Sections 1 & 2
- I think the clarity of the paper would really benefit from a classic "Figure 1" flow chart on page 1 that walks through the structure of the benchmark.

Overall, this is a work targeting an interesting problem, but the paper could be written much better: the manuscript is overburdened by technical details, and I still struggle to grasp the significance of the findings. The paper reads very rushed, which often happens when you just finish the experiments and can't get in the headspace to think more about writing & presentation. It'll be helpful to slow down and take some time to think about how to best organize the paper and communicate your key results.

---

> ### Author Rebuttal · Authors · 2026-03-31
>
> Thank you for the thoughtful review, especially for the concrete suggestions on presentation and organization.
>
> > *"The figures and tables are not very readable. Figure 4, which seems to present the most important results, is especially congested."*
>
> Thanks for pointing this out. We are redesigning all figures for readability, including an [improved Figure 4](https://anonymous.4open.science/r/SALT-Rebuttal-74DD/Revised_Figure4.pdf). We also intend to supply a plotly HTML version of the figures in the supplementary, so readers can inspect the plots more clearly.
> > *"The conclusion of Section 5.1 is not at all surprising. What is new to your findings beyond the well-established intuition that LLMs are more sensitive to errors in long contexts?"*
>
> We agree that the original section required further depth. Although degradation with longer context and degradation following prefix errors may appear intuitively related, the relationship between them is less clear: do they reflect the same underlying phenomenon, or distinct sources of degradation? And if they are distinct, which of the two is more impactful?
> Leveraging the unique properties of SALT, we therefore added a deeper error analysis that decouples the effects of *context correctness* from *context length*. This is particularly enabled by SALT, because its deterministic unit-level supervision makes such prefix manipulations precise and quantitatively interpretable. Specifically, we use the DNA task, where each output atom is determined directly by the input sequence, so the correctness of preceding generated atoms does not inherently dictate the validity of subsequent ones. This deeper analysis yields two concrete findings:
> - **Impact of Context Correctness**: By holding generation length constant and isolating the correctness of the context, we directly measured how context correctness affects accuracy. We observe a 15% improvement in accuracy when the context is strictly correct. Notably, this is non-trivial in the DNA task, where each target atom is determined strictly by the input prompt and not by the correctness of prior generated atoms.
> - **Context Enhancement Experiment**: Motivated by this result, we then asked whether improving context quality at the expense of longer context could still improve accuracy, thereby helping distinguish which factor is more dominant. We introduced additional synthetic correct atoms into the prefix, creating two opposing pressures:
>
>     1) **Positive**: Increasing the overall correctness ratio in the context.
>     2) **Negative**: Increasing the total context length.
> Our findings revealed that despite improving the context quality, we observe an accuracy degradation of up to 6%, suggesting that context length poses a greater challenge to model stability than the ratio of correct-to-incorrect prior atoms.
>
> > *"The presentation of Section 5.2 is not clear: instead of crowding everything into a numbered list, wouldn't it be better organized if the presentation of key results is disentangled from the more detailed analysis?"*
>
> This is an excellent suggestion for the revision. Following your recommendation, we plan to restructure Section 5.2 by opening with a short overview of the key findings (e.g., confidence-function variation, granularity sensitivity, the precision–calibration relationship, and the reasoning trade-off), and then turning to the detailed analyses.
>
> > *"A lot of Section 2 could go into the appendix. Section 2.3 do not seem to be custom implementations, so we don't really need the detailed formalizations in the main text. Section 2.1 could also be better consolidated."*
>
> We believe Section 2 can be streamlined. In the revision, we plan to move Section 2.3, along with some of the more detailed formalism, to the appendix.
>
> > *"Is Section 3 really necessary? You've already surveyed a lot of relevant prior work in Sections 1 & 2"*
>
> We had also reflected on whether Section 3 is necessary in the main paper, given the amount of related work discussion already covered in Sections 1 and 2. Following your feedback, we are now inclined to move it to the appendix in the revision, as we agree this would likely improve the paper’s flow and reduce redundancy.
>
> > *"the paper would really benefit from a classic Figure 1"*
>
> We fully agree. We are adding a classic Figure 1 that gives an overview of SALT through a concrete example, showing how a single generation is evaluated at multiple granularities. We believe this makes the benchmark much easier to grasp from the start. We attach [the Figure](https://anonymous.4open.science/r/SALT-Rebuttal-74DD/Figure%201.png) in response to your suggestion.

---

> > ### Author Rebuttal · Reviewer_GRXz · 2026-04-06
> >
> > Thank you for your detailed rebuttal. You have addressed my concerns, and I will increase my score to weak accept.

---

> > > ### Author Response · Authors · 2026-04-08
> > >
> > > We sincerely thank the reviewer for acknowledging our work and for raising the score. We appreciate the time and effort spent reviewing our paper.

---

### Decision · Program_Chairs · 2026-04-30

**Decision:**

Accept (regular)

**Comment:**

The paper introduces the SALT benchmark for evaluating LLM uncertainty in long contexts, consisting of 6 tasks with deterministic ground truth that does not rely on human experts. Analyzing 50+ LLMs on SALT finds that the relationship between calibration and accuracy flips from lower- to higher-performing models, and reasoning often comes in the form of a trade-off.

Reviewers found the problem highly relevant: as LLMs are increasingly used for long-form generation, uncertainty needs to be measured at a finer granularity than the full response. The benchmark, SALT, is seen as a meaningful contribution because it brings together long-form generation and uncertainty estimation in a coherent way. Reviewers also highlighted findings such as the decline in correctness over the course of long generations, the dependence of uncertainty ranking quality on evaluation granularity, and the reasoning-related tradeoffs observed in the experiments.

Overall, the meta-reviewer recommends acceptance of the paper.